# Amazon forest resistance to drought is increased by diversity in hydraulic traits

Liam Langan [1] ✉, Simon Scheiter[1], Thomas Hickler [1,2,4] & Steven I. Higgins [3,4]

The unique biodiversity and vast carbon stocks of the Amazon rainforests are essential to the Earth System but are threatened by future water balance changes. Empirical evidence suggests that species and trait diversity may mediate forest drought responses, yet little evidence exists for tropical forest responses. In this simulation study, we identify key axes of trait variation and quantify the extent to which functional trait diversity increases tropical forests' drought resistance. Using a vegetation model capable of simulating observed tropical forest drought responses and trait diversity, we identify emergent trade-offs between water-related traits (hereafter hydraulic traits) as a key axis of variation. Our simulations reveal that higher functional trait diversity reduces site-scale biomass loss during sudden catastrophic drought, i.e., a 50% precipitation reduction for four and seven years, by 17% and 32%, respectively, and continental-scale biomass loss due to severe chronic climate change-associated precipitation reductions, i.e., RCP8.5, constant $CO_2$ at 380 ppm, and a 50% precipitation reduction over 100 years, by 34%. Additionally, we find that functional trait diversity-mediated biomass resistance is stronger under more severe drought conditions. These findings quantify the essential role of hydraulic-trait diversity in enhancing tropical forest drought resistance and highlight the critical linkages between biodiversity conservation and climate change mitigation.

The world's tropical forests, particularly the Amazon, are globally important reservoirs of carbon and biodiversity. These reservoirs are however threatened by climate change-induced reductions in precipitation and increases in evaporative demand[1]. It has been proposed that these changes will have cascading negative impacts on Earth system functioning, ecosystem service provision, and human well-being[2]. Indeed, a growing literature convincingly documents widespread forest dieback and links this to ongoing climatic changes[3,4]. This evidence and reasoning suggest that the Amazon forest is particularly at risk[5]. However, substantial contradictory evidence from theoretical ecology and experimental ecology[6–11] suggests that high-diversity ecosystems, such as the Amazon, should be amongst the most resistant ecosystems on the planet, due to their high species richness and

functional trait diversity (note: functional trait diversity and functional diversity are synonymous), which enhances ecosystem resilience[12–14].

Early studies that used dynamic vegetation models to forecast the impacts of a warming climate system on Amazonian forests predicted dramatic diebacks and cascading impacts on the Earth system[15]. In contrast, modelling studies commonly simulate drought-induced diebacks which are lower than observed diebacks[16,17]. A further problem with this previous work is that the models used did not consider how hydraulic traits can mediate plant water status and how diversity in these traits may influence how forests respond to drought[18]. Therefore, these analyses could not evaluate whether hypotheses from theoretical and experimental ecology, that diversity increases the resistance of ecosystems, apply to hyper-diverse tropical forests.

[1]Senckenberg Biodiversity and Climate Research Centre, Frankfurt am Main, Germany. [2]Department of Physical Geography, Geosciences, Goethe University, Frankfurt am Main, Germany. [3]Plant Ecology, University of Bayreuth, Bayreuth, Germany. [4]These authors jointly supervised this work: Thomas Hickler, Steven I. Higgins. ✉e-mail: liam.langan@senckenberg.de

Predicting the effects of functional trait diversity on the functioning of ecosystems has been deemed the 'Holy Grail' of trait-based ecology[19]. However, the related challenges of vegetation models' failure to reproduce observed biomass decreases in tropical forest drought experiments[16,17,20,21] and the lack of plant traits within these models, which mediate plant hydraulic performance[18], impair our ability to predict whether diversity in traits mediates tropical forest responses to long-term and sudden reductions in water availability[22].

We pursue two avenues that simultaneously address these barriers. First, we include in our model, advances in disciplinary understanding of plant hydraulics (Supplementary Notes "Plant Hydraulic Strategies—safety versus efficiency trade-offs")[23,24]. Typically, in dynamic vegetation models (DVMs), plant water stress is governed by factors external to a plant, such as precipitation and soil moisture content[18]. The inclusion of plant hydraulics in aDGMV2 allows that, in addition to external factors, plant hydraulic performance is governed by properties intrinsic to a plant, i.e., root, stem, and leaf traits and phenotypes. Thus, this implementation allows the action of natural selection across an individual's root, stem, and leaf traits, which simultaneously affect hydraulic performance through both abiotic and biotic components of the soil-plant-atmosphere continuum (Fig. S16). Trade-offs between maximum xylem and leaf conductivity and cavitation[24] allow alternative strategies to emerge in the model. Phenological traits, i.e., whether an individual is evergreen or deciduous, determine whether individuals endure or avoid water shortages. Below-ground traits determine the amount of carbon allocated to roots, root shape, and plant rooting depth. These traits affect the water availability of individuals in the model and thereby competition for water (Fig. S16). Second, by creating an eco-evolutionary framework through which trait diversity emerges, we can consider how functional trait diversity impacts the response of ecosystems to sudden short-term catastrophic drought and chronic long-term climate change-associated precipitation reductions.

We use a dynamic vegetation model (DVM) with an improved representation of plant hydraulics to achieve this. The model can accommodate high levels of functional trait diversity, allowing us to evaluate whether diversity can buffer the carbon sequestration functioning of Amazonian forests under warming and drying climates. The model (aDGVM2[25,26], see Fig. S1 and Supplementary Notes) combines standard DVM methodology with eco-evolutionary processes. In previous work, aDGVM2 has been shown to accurately simulate the observed distribution of the Amazonian forest and the surrounding Cerrado savannas[25]. It has also been benchmarked against biomass, tree cover, and tree height for the study area[25]. It is individual- and trait-based. In the model, individuals have adaptive traits (see Table S10 for a full list), which influence their performance in competing for light and water. The traits of an individual have values within predefined ranges (Table S10). The benefits and costs of an individual's traits influence its phenotype (e.g., stem biomass, leaf area, height, number of seeds produced) (outer ring Fig. S1) in interaction with the specific abiotic (climate and soils) and biotic (competition between individuals) context. Feedbacks between the traits, phenotype, and changing abiotic and biotic conditions determine an individual's relative fitness through time via reproductive success[27]. Differential relative fitnesses through time drive the processes of community assembly[28] and adaptive trait evolution[29].

Here we summarise how aDGVM2 simulates community assembly and adaptive trait evolution; the appendices provide more detail. The model simulates individuals (trees and grasses); every individual can have a unique set of trait values. Trait values are inherited and can change through generations. Parent individuals belonging to a particular species (species is a generic label applied to individuals at initialisation to allow reproductive isolation between groups) exchange trait values via crossover with a fixed probability to produce seeds (see Supplementary Notes "Reproduction, inheritance, mutation, and

crossover", Fig. S19). The trait values of each seed are further modified by mutation with a fixed probability. Each year during recruitment, seeds are randomly drawn from the seed bank (see Supplementary Notes "Plant recruitment"). These recruits can then grow into saplings and small trees. The number of seeds an individual produces is determined by the amount of carbon it assimilates, the proportion of carbon allocated to reproduction (see Supplementary Notes "Carbon allocation", Fig. S18), and seed weight. The proportion of carbon allocated to reproduction and seed weight is an adaptive trait (see Supplementary Notes "Plant recruitment", Table S9). Individuals producing more seeds have a higher probability that their seeds will be drawn from the seed bank, while heavier seeds have a higher probability of germination and establishment. Thus, trait inheritance, crossover, and mutation allow new, potentially novel trait combinations to emerge via seeds. This combination of abiotic and biotic (competition) filtering and evolution allows functional trait diversity to emerge in modelled vegetation stands and thereby provides new opportunities for exploring how functional trait diversity impacts ecosystem services (inner ring Fig. S1) such as carbon sequestration.

Global biodiversity is being lost at an alarming rate. Yet, the extent to which functional trait diversity loss may reduce the resistance of tropical forests to anticipated changes in climate remains unknown. In this simulation study, we aim to fill this knowledge gap by exploring how functional trait diversity mediates tropical forest responses to two types of drought: catastrophic (sudden and severe reductions in precipitation) and chronic (long-term climate change-induced precipitation reductions). Site-scale catastrophic-drought simulations allow comparisons with empirical observations. Demonstrating realistic catastrophic drought responses is requisite for the examination of functional trait diversity-mediated responses and supports the investigation of continental-scale chronic drought responses. Site-scale diversity manipulations inform continental-scale manipulations essential for assessing the broader implications for tropical forests across the Amazonian region to the end of the 21st century.

Given the limited understanding of diversity-mediated resistance of tropical forest ecosystems, the underlying mechanisms, and the axes of trait variation potentially involved, our approach is intentionally exploratory[30,31]. Using an iterative experimental framework, we integrate site- and continental-scale analyses, where findings from one scale inform the next. By conducting a series of exploratory sensitivity analyses, we aim to identify key axes of trait variation, uncover mechanisms underlying drought resistance, and assess the magnitude of diversity-mediated drought resistance. Our aim with this approach is to answer the following key research questions (RQs):

1.1 Catastrophic drought: site-scale analyses

 1.1 Does functional trait diversity mediate biomass responses to catastrophic drought at the site scale, and what is the magnitude of this effect?

 1.2 What key axes of trait variation mediate diversity effects during catastrophic drought, and what mechanisms are associated with these axes?

2. Chronic drought and climate-change simulations: continental-scale analyses

 2.1 Do critical axes of trait variation at the continental scale align with site-scale axes, and are there indications of future shifts in the dominance of plant strategies?

 2.2 Does functional trait diversity mediate tropical forest biomass responses to chronic climate change-induced precipitation reductions at the continental scale, and what is the magnitude of this effect?

In this work, we clarify the role of functional trait diversity in enhancing the resistance of tropical forest ecosystems to climate change. We show that higher simulated functional trait diversity associated with plant traits that influence plant water status strongly

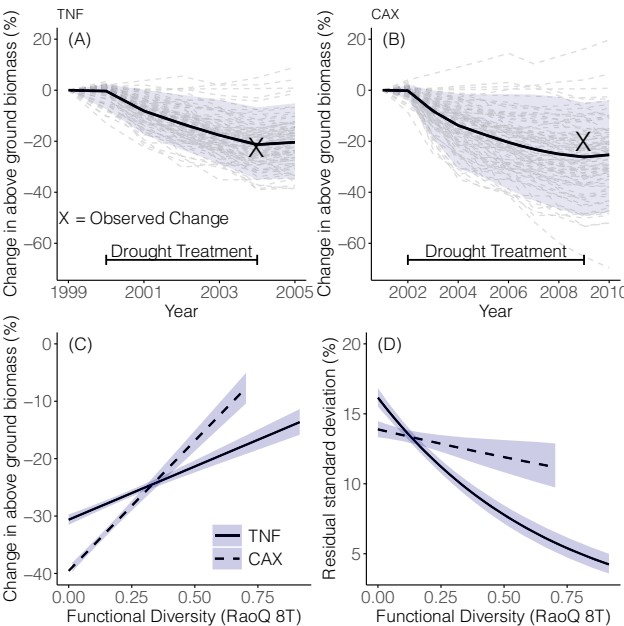

**Fig. 1 | Functional trait diversity-mediated drought responses.** Simulated drought experiments for TNF (**A**) and CAX (**B**). Shown is the change in above-ground biomass during drought using the default model setup with no constraints placed on diversity. Each line represents one of 96 replicate simulations. The bold line represents the mean over the replicate simulations. Crosses indicate the observed biomass reduction and the year these reductions were observed. Horizontal bars indicate the start and end of simulated and real-world drought treatments. For (**C** and **D**), simulated drought experiments were repeated, and the diversity level was manipulated by initialising the model with varying numbers of unique trait combinations (species). The model was initialised with the following 18 different numbers of species (1, 2, ..., 12, 16, 32, 48, 64, 80, and 96). No two species had identical trait values. 96 replicates were run for each species number treatment (see Fig. S1 and "Methods"). **C** Mean predicted post-drought biomass reductions vs. simulated functional trait diversity. **D** Residual standard deviation about mean biomass reductions. **C**, **D** The results of a Bayesian linear regression analysis (see "Examining relationships between functional trait diversity and changes in above-ground biomass" and Tables S2, S3). Shading indicates the 95% credible interval around mean posterior predictions. Rao's quadratic entropy, i.e., the mean functional dissimilarity between two randomly selected individuals[82], was used as a measure of functional trait diversity in (**C**) and (**D**). 8T indicates that RaoQ was calculated using the 8 traits displayed in Figs. 2, 3. The data underlying this figure are provided in figshare (https://doi.org/10.6084/m9.figshare.26232395).

reduces biomass loss during drought. For two tropical forest sites, we show that higher functional diversity reduces biomass loss during sudden catastrophic drought by up to 17% and 32% respectively. At the continental scale, our results indicate that higher functional diversity can reduce biomass loss associated with severe chronic climate change-induced precipitation reductions by up to 34% and that the increased drought resistance afforded by higher functional diversity increases with drought severity.

## Results and discussion
### Catastrophic drought: site-scale analyses of simulated vs observed biomass responses
We first use aDGVM2 to mimic drought experiments run at Tapajos (TNF)[20] and Caxiuana (CAX)[21] national forest sites (Fig. S3) by simulating catastrophic drought (i.e., 50% reduction in daily precipitation for 4 years at TNF and 7 years at CAX). We compare mean model responses of replicate simulations to observations as the model includes a range of stochastic processes. Simulations using the standard model parameterisation showed that the mean percentage of

biomass lost and the timing of these losses were reproduced at both sites (Fig. 1A, B).

### Catastrophic drought: does functional trait diversity mediate biomass responses to catastrophic drought? (RQ 1.1)
We then investigate whether functional trait diversity, i.e., diversity in traits which are impacted by, and impact, biotic and abiotic conditions[13], affects biomass responses to drought and conduct simulations initialised with different numbers of functionally unique species (see Supplementary Notes and Table S9 for a list of traits and their potential range, Table S1 for experimental setup, and "Methods" section "Examining relationships between functional trait diversity and changes in above-ground biomass" and Tables S2, S3 for methods and output). This simulation experiment shows that communities exhibiting lower functional trait diversity exhibit higher biomass reductions following drought (Fig. 1C). This analysis answers our RQ 1.1; we show that for the TNF and CAX sites, low-diversity communities lose ca. 17% and 32% more biomass than high-diversity communities. Additionally, theoretical and empirical studies of diversity-stability relationships highlight that higher diversity can stabilise various components of ecosystem function[32–34]. Our analysis of the relationship between the residual variance about mean responses and functional diversity (Fig. 1D) finds significant interactions. We show that higher functional trait diversity reduces the variability around predicted mean responses.

### Catastrophic drought: key axes of trait variation mediating diversity effects (RQ 1.2)
To investigate potential axes of trait variation and identify trait compositions that impart diverse communities a higher resistance to drought, we performed a principal component analysis (PCA). The PCA reveals that the functional composition of modelled communities is clearly organised in a two-dimensional strategy space (Fig. 2A, B) with trade-offs consistent with commonly described plant strategies[35]. These emergent axes of variation are a productive vs. conservative[36] dimension on the first axis[37,38], and whole-plant hydraulic strategies[39,40] on the second axis via coordination (selection-driven trait covariance[40]) between plant hydraulic and phenological traits (Fig. 2A, B).

   Productive vs conservative plant strategies differentiate across carbon allocation traits. Productive strategies allocate more of the acquired carbon to the immediate growth of roots, stems and leaves (see Fig. S18), which trades off against allocation to storage, which a plant can subsequently use to flush leaves when phenological triggers are activated or recover from fire-associated top-kill. Along this axis, deciduous plant phenologies tend towards the conservative end of the spectrum while evergreen plant phenologies tend towards the more productive axis. Indeed, recent evidence indicates that trade-offs between allocation to growth vs storage represent an important axis of variation within the spectrum of plant life-history strategies[36].

   The second axis shows that selection leads to the emergence of whole-plant hydraulic strategies via coordination between multiple plant traits, which mediate water relations and phenological traits[41] that closely match observed plant strategies across Amazonia[39]. In the model, there are four phenological strategies (see Supplementary Notes "Leaf phenology"). Evergreen and deciduous plants use two triggers for leaf flush or abscission—a water trigger, which responds to soil water content (Deciduous Water, Evergreen Water) and a light trigger, which responds to changes in light availability (Deciduous Light, Evergreen Light). Classifying species by leaf phenology reveals considerable overlap and distinct differences along both axes of variation (Fig. 2C, D). Oliveira et al.[39] highlighted two key trade-off axes for tropical forests—the avoidance-resistance and slow-safe vs fast-risky trade-offs, the latter is implicit in aDGVM2. Here, xylem cavitation

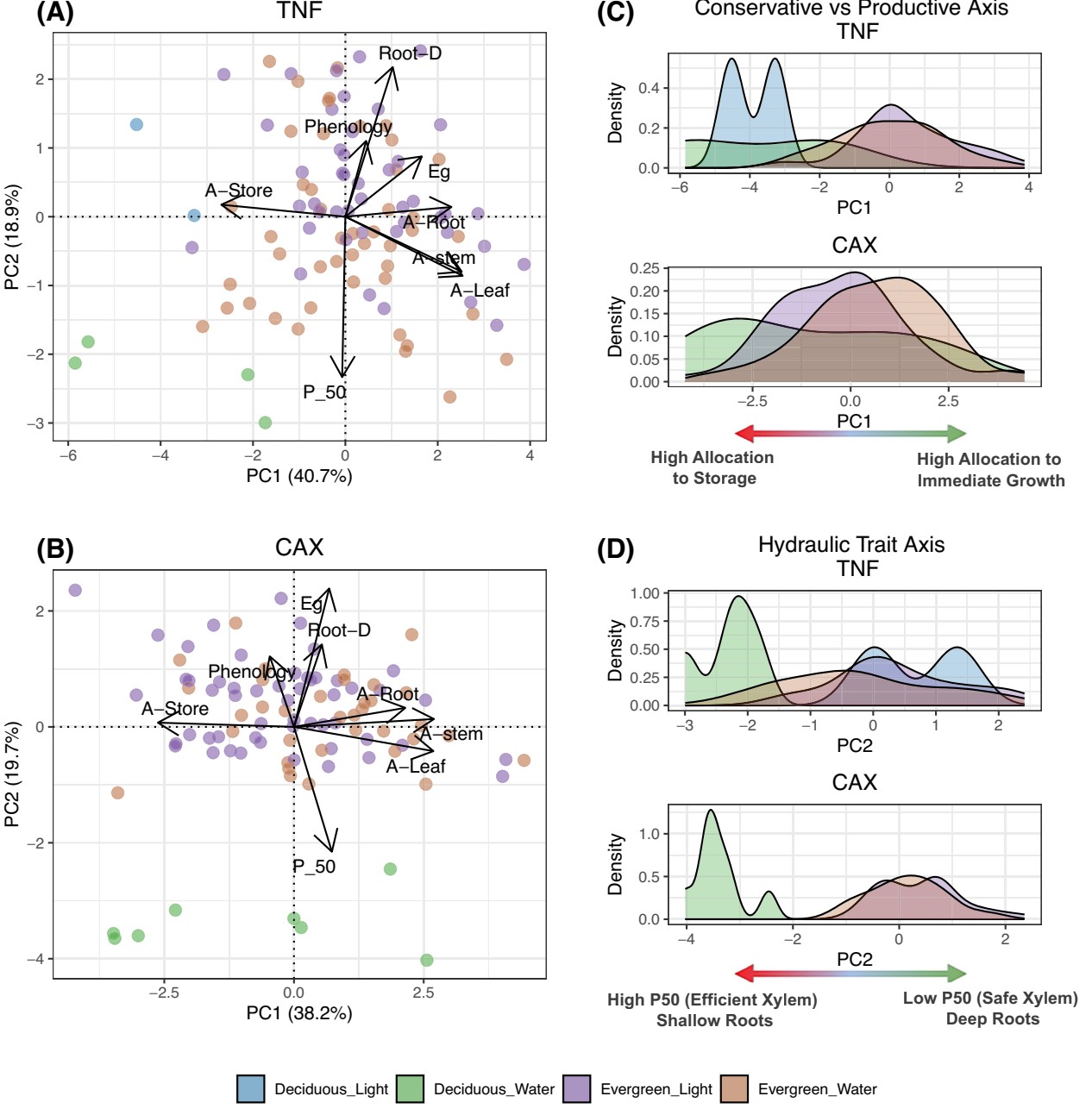

**Fig. 2 | Emergent axes of trait variation.** Principal component analysis of the trait combinations (species) used to initialise simulated diversity experiments (Fig. 1). PCA for (**A**) TNF and (**B**) CAX; points represent the position of each of the 96 species in trait-space. Density plots for the conservative vs productive axis (PC1) for the TNF and CAX sites (**C**). Density plots for the hydraulic trait axis (PC2) for the TNF and CAX sites (**D**). Coloration of points and density curves in (**A–D**) based on the four leaf phenological combinations of plant strategies. All traits are adaptive, see Supplementary Notes "Reproduction, inheritance, mutation, and crossover", Table S9. Conservative vs productive axis (PC1, **A–C**): arrows point toward a higher allocation of carbon to grow roots (A-Root), leaves (A-Leaf) and stems (A-Stem) vs deferred growth via higher allocation to storage (A-Store). Hydraulic-trait axis (PC2, **A**, **B**, **D**): arrows point toward less negative water potential at 50% loss of conductance (P50), an evergreen leaf phenology (Eg) (as opposed to deciduous), light-triggered leaf flush (Phenology) (as opposed to water-triggered), increased maximum rooting depth (Root-D). Note: significant conclusions cannot be drawn from the two Deciduous_Light species. The data underlying this figure are provided in figshare (https://doi.org/10.6084/m9.figshare.26232395).

resistance is defined by the adaptive trait P50. A plant's P50 value is related to other traits (Figs. S1, S16) via hard-coded empirically defined trade-offs (see Supplementary Notes "Plant Hydraulic Strategies—safety versus efficiency trade-offs") and defines specific leaf area (SLA), leaf longevity (Fig. S17), stem and leaf conductivity, and wood density. Our simulations show that the hydraulic trait axis (Fig. 2D) is defined by the trade-off between drought avoidance and resistance: at the vulnerable end, shallow roots and high P50 (less negative) are associated

with drought deciduousness (Deciduous_Water) which allows these strategies to reduce the consequences of drought, while the resistant end with safe xylem (more negative P50) and deep rooting is predominantly occupied by evergreens that flush leaves in response to soil water and light availability (Evergreen_Water, Evergreen_Light). Thus, along this axis, classically hydraulic and phenological traits together mediate plant water status; we, therefore, refer to these traits collectively as hydraulic traits.

Brum et al.[42] highlight the importance of the avoidance-resistance trade-off axis in a seasonal Amazon forest. In this forest (TNF), they associated avoiding strategies with deeper roots, greater water access, and higher P50, while tolerating strategies had shallower roots, exposing them to more regular water deficits, and more negative P50. In a global synthesis, Laughlin et al.[43] demonstrated that different combinations of rooting depth and P50 depended on specific abiotic factors such as aridity, precipitation seasonality, and water table depth. Our simulations reflect patterns expected for TNF (humid, deep water table, and a prolonged dry period[42]). Within this context, aDGVM2 captures the two dominant patterns predicted by Laughlin et al.: vulnerable strategies with high P50 and shallow roots, and resistant strategies with low P50 and deep roots.

Our results suggest that aDGVM2 effectively captures important axes of tropical forest trait variation. Further, while variability within each leaf phenological strategy is large, differences along the hydraulic trait axis highlight that leaf phenology accounts for a substantial portion of the diversity along this axis. This provides a practical means to test the effects of reduced diversity, as removing leaf phenologies removes significant variation along the hydraulic trait axis.

Niche differentiation, a well-established ecological principle, promotes species co-existence and increases the resilience of more diverse communities to perturbations[44]. In this study, differentiation of plant strategies across the hydraulic axis of trait variation is the most likely candidate to explain the simulated lower biomass loss in more functionally diverse communities. Catastrophic drought, which causes sudden and severe changes to the water balance, interacts with hydraulic traits which govern plant water availability. In the model, hydraulic traits mediate plant water balance by influencing temporal and spatial access to water. Specifically, leaf phenological traits regulate the timing of leaf flush or abscission, determining the timing of peak water demand for plant strategies (see Supplementary Notes "Leaf phenology"), while variability in these traits between plant strategies can reduce the temporal overlap of their peak water demands. Root traits affect spatial dynamics by determining a plant's rooting profile and access to soil water; variability in root traits is thought to be able to facilitate resistance to water stress[42]. Additionally, P50, which defines xylem resistance to cavitation and transport efficiency, covaries in a consistent manner with both leaf phenological and root traits (Fig. 2). These findings address RQ 1.2; they point towards leaf phenological and rooting traits as being the primary candidates responsible for simulated drought resistance. Variation in these traits can promote temporal and spatial niche separation of water use, potentially reducing peak ecosystem water stress and associated mortality. These results suggest that further investigation of leaf phenological and rooting traits could provide deeper insights into the mechanisms by which diversity mediates drought resistance.

## Catastrophic drought: mechanisms of diversity-mediated responses to catastrophic drought (RQ 1.2)

Following the identification of the hydraulic trait axis of variation, we next aimed to explore potential mechanisms underlying diversity-mediated responses to catastrophic drought. Niche differentiation can promote increased resistance via two non-mutually exclusive mechanisms. First, it can increase total water resource availability via more complete soil exploration[45]. Second, temporal and spatial niche differentiation may buffer the effects of drought by increasing the stability of water use[42,46].

To better understand potential mechanisms and relationships between modelled responses and ecosystem fluxes, we examine simulated transpiration as this represents a main indicator of water use dynamics. Our analysis reveals that simulated mean transpiration did not increase with functional diversity (Fig. S4B). However, we find that stability in transpiration increases with functional diversity (Fig. S4A) via reductions in the standard deviation (Fig. S4C). This result links

diversity-mediated drought responses with stability in transpiration and provides further evidence in support of temporal (mediated by leaf phenology) or spatial (mediated by differences in root distributions) niche differentiation in water use as being the main mechanism underpinning simulated diversity effects.

Experimental support for relationships between the stability of ecosystem properties and functional diversity in tropical or subtropical forests is rare. An exception[46], identified mechanistic linkages between diversity in P50, a key trait composing the hydraulic-trait axis above (Fig. 2), and stability in productivity. The authors suggest that other hydraulically important traits, such as those associated with leaf phenology, below-ground traits, and below-ground niche differentiation in rooting depth, may additionally play a role in mediating forest stability[46]. The alignment of these additional traits along the hydraulic-strategy axis (Fig. 2) supports this suggestion and highlights that variability in one of these traits (i.e., diversity) in the model, and empirically[39,47,48], can be linked to variability in the others.

To further investigate mechanisms underlying diversity effects, we focus on root and leaf phenological traits. These traits are chosen because they: (1) align along the hydraulic-trait axis (Fig. 2), (2) have empirical support for their influence in promoting spatial and temporal niche differentiation, and (3) could be experimentally manipulated at appropriate spatial scales. At the site scale, we test the role of root niche differentiation in mediating diversity effects. This scale allows us to use replicate simulations to identify and re-run cases where all plant strategies are present with relatively even abundances (see "Methods"), a setup supported by portfolio and evenness effect theory[49]. Testing this at the continental scale is not computationally feasible. We show that plant strategies' rooting niches are differentiated (Figs. 2, S5). We then remove this differentiation before the application of drought by setting all below-ground traits to the same values. This treatment causes additional drought-induced biomass loss (20% at TNF and 37% at CAX) (Fig. S6). Thus, these results address RQ 1.2, whereby they show that niche differentiation of below-ground traits is a key mechanism regulating tropical forest responses to drought.

The improvement of the linkages between above-ground and below-ground plant trait-suites which govern plant performance has been suggested as a promising way to improve vegetation model performance and guide empirical research, particularly for tropical forests[50]. Indeed, the above- and below-ground trait coordination and niche differentiation emerging from our simulations suggest that a holistic approach to whole-plant strategies and increased focus on below-ground traits represents a plausible and fruitful avenue for further empirical research and vegetation model development.

## Chronic drought and climate-change simulations: continental-scale analyses

Scaling from the site to continental scale (Fig. S3) and from sudden catastrophic drought to chronic climate change-associated precipitation reductions (Fig. S2B, C), simulations are conducted using RCP4.5 and 8.5 climate forcing[51]. As atmospheric $CO_2$ increases, the $CO_2$ fertilisation effect, i.e., increased uptake of carbon by plants via photosynthesis with increasing $CO_2$[52], increases modelled leaf-scale photosynthesis and water use efficiency[53] via changes to the ratio of photosynthesis to photorespiration and reduced stomatal conductance. However, the best evidence suggests that confidence in the magnitude of $CO_2$ fertilisation of growth is low[53]. Therefore, to explore potential bounds on vegetation responses with respect to $CO_2$ fertilisation, we run simulations both with and without a $CO_2$ impact on plants (Fig. S2G). For simulations without a $CO_2$ impact, we hold $CO_2$ levels constant at 380 ppm (Fig. S2G). Holding $CO_2$ constant allows the examination of responses in the absence of $CO_2$-fertilised plant growth. For simulations with a $CO_2$ impact, we increase $CO_2$ as prescribed by the RCP scenario (see "Methods": "Model forcing and

simulation setup", "Amazonian climate change scenario simulations–Full diversity", Fig. S2G).

Across all future change scenarios there is considerable variability in biomass changes across the study area; biomass increases (Fig. S7) are associated with areas where precipitation remained stable or increased (Fig. S2B, C) while biomass decreases are associated with areas where precipitation is reduced. In simulations where $CO_2$ and climate vary according to RCP scenarios, total biomass increases by 16% with RCP4.5 and 23% with RCP8.5 by 2100 (Figs. S8A, S7A, C). In simulations where $CO_2$ is held constant at ambient levels, total biomass stored is reduced by 20% with RCP4.5 and 44% with RCP8.5 (Figs. S8A, S7B, D). Without $CO_2$-fertilised growth, forest biomass loss is widespread (Fig. S7B, D).

While $CO_2$ fertilisation has been predicted to mitigate the risk of Amazon forest dieback[54,55], our results reveal that, even with increasing $CO_2$, there are large areas of Amazonia where biomass is reduced by up to 40% (Fig. S7A, C). Empirical studies show that the Amazon's ability to sequester carbon peaked in the 1990s[56] and has been decreasing[4]. This declining sink strength is generally at odds with model predictions[56]. For our study area (Fig. S3), we find that peak biomass accumulation occurs in the 1990–2000 decade in all of our future scenarios (Fig. S8B). Additionally, in agreement with previous work[4], which predicts that the Amazon sink strength will reach zero between 2011 and 2089, for our continental-scale simulations, we find that this threshold is crossed earliest (2000–2010) in scenarios without increasing $CO_2$ concentrations (RCP4.5 Clim, RCP8.5 Clim) (Fig. S8B). In scenarios with increasing $CO_2$, this threshold is approached by 2100 in RCP8.5 Clim+$CO_2$ and crossed between 2070 and 2080 for RCP4.5 Clim+$CO_2$ (Fig. S8B).

These results demonstrate that a model that exhibits a response to catastrophic drought that is broadly in line with observed responses, also exhibits biomass reductions at the continental scale in response to chronic, climate change-induced precipitation reductions. Additionally, we simulate that Amazon's ability to sequester carbon aligns well with empirical data. Taken together these results: (1) provide evidence that improvements in the sensitivity of vegetation to catastrophic drought may represent a path toward improved modelling of future carbon dynamics in both DGVMs and Earth System Models (ESMs), and (2) increase our confidence in model performance; this supports further evaluation of how functional trait diversity mediates vegetation responses at the continental scale.

### Chronic drought: critical axes of trait variation and climate change-induced shifts in functional dominance (RQ 2.1)

The continental-scale simulations also reveal that natural selection-driven community assembly within the model produces mean SLA values across the study area that closely match a number of published SLA maps (Figs. S9, S10), providing evidence that the model is capable of capturing variability of this key trait across the study region. At this spatial scale, we find that the main axes of trait variation (Fig. 3) closely match those found at the site scale. At the continental scale, we find that the hydraulic-trait and conservative vs productive axes of variation exchange places. Here, the hydraulic-trait axis aligns along the first principal component axis. However, these two axes remain the two main sources of variance, explaining 49–60% depending on location and scale. The consistency of the hydraulic-trait axis of variation across spatial scales again supports its relevance for testing diversity-mediated vegetation responses at the continental scale.

Future change simulations revealed shifts in the functional dominance of plant strategies across trait space (Fig. 3). Changes in the frequency of individuals across trait space show that, while there are marginal differences between simulations where $CO_2$ varies according to RCP scenarios (Fig. 3A, C) and simulations where $CO_2$ is held constant at ambient levels (Fig. 3B, D), all scenarios show increases in frequency in areas of trait-space associated with productive strategies

and deciduous water-triggered strategies (Deciduous_water). Such increases in productive strategies can be associated with both vegetation responses to drought-induced mortality events[57] and increasing $CO_2$[58], while increases in the dominance of deciduous species in tropical forests have been documented following long-term precipitation reductions[59] and drought[60].

The alignment of simulated SLA with published products, the consistency of the hydraulic-trait axes across spatial scales, and the shifts in functional composition–broadly in line with theorised and observed compositional changes owing to $CO_2$ and precipitation changes–suggest that the model effectively captures both contemporary and future drivers of community assembly. Together, these findings highlight the importance of diversity in hydraulic traits for the future adaptive potential of Amazon rainforests and underscore the importance of ascertaining the extent to which this diversity may mediate forest futures.

### Chronic drought: testing leaf phenology and quantifying diversity effects (RQ 2.2)

At the site scale, we have shown statistically (Fig. 1) and mechanistically (Fig. S6) that functional trait diversity increases the biomass resistance of tropical forests to catastrophic drought. While the site-scale analysis confirms the role of root trait variability in mediating responses to catastrophic drought (Fig. S6), it also highlights the potential importance of variability in leaf phenological traits for regulating the timing of peak water demand among plant strategies. Thus, at the continental scale, we aim to test whether variability in traits associated with leaf phenology mediates tropical forest responses to chronic drought. This experimental focus is supported by continental-scale climate change simulations, which predict that future conditions induce changes in leaf phenology and favour water-triggered plant strategies (Fig. 3).

Thus, to explore the consequences of a reduction in diversity associated with leaf phenology, we re-run climate change scenarios by: (1) removing deciduous water-triggered leaf phenologies, and (2) additionally removing evergreen water-triggered leaf phenologies. The binary nature of leaf phenological traits makes simulations computationally tractable at the continental scale, requiring only two additional runs of each climate change scenario. Functional trait diversity is then calculated using the remaining hydraulic traits, P50 and maximum rooting depth, as these are not directly manipulated. These simulations show that functional trait diversity is unevenly distributed across the study area (Fig. S11A) and that reducing variability in leaf phenology leads to a spatially patterned reduction in functional trait diversity (Fig. S11B, C). These results highlight the linkages between above- and below-ground traits. Specifically, while variability in below-ground traits and resulting niche differentiation (Fig. S6), along with variability in P50[22,46], can buffer the impact of drought on tropical forests, our findings show that reducing variability in leaf phenological traits causes a reduction in diversity associated with rooting depth and P50.

The amount of standing biomass in tropical forests has been linked to species diversity[61]. Our simulations echo these empirical findings, whereby we show that functional diversity loss causes lower contemporary and future biomass storage across the study area (Fig. S12). To initially assess the effect of reducing diversity in leaf phenological traits, we filter data to only include sites of comparable biomass. This reveals that: (1) the average biomass in 2100 is generally higher when diversity is not constrained (Fig. S13B–F), and (2) the biomass differences between simulations run with the standard model parameterisation, and those run where diversity in leaf phenology is reduced, are influenced by the magnitude of precipitation reduction. These differences are highest when drought is highest (RCP4.5 23% and 28%) (Fig. S13E) and in the absence of elevated atmospheric $CO_2$ (RCP4.5 18% and 20%) (Fig. S13F). However, the effects of functional trait diversity on vegetation's response to precipitation changes

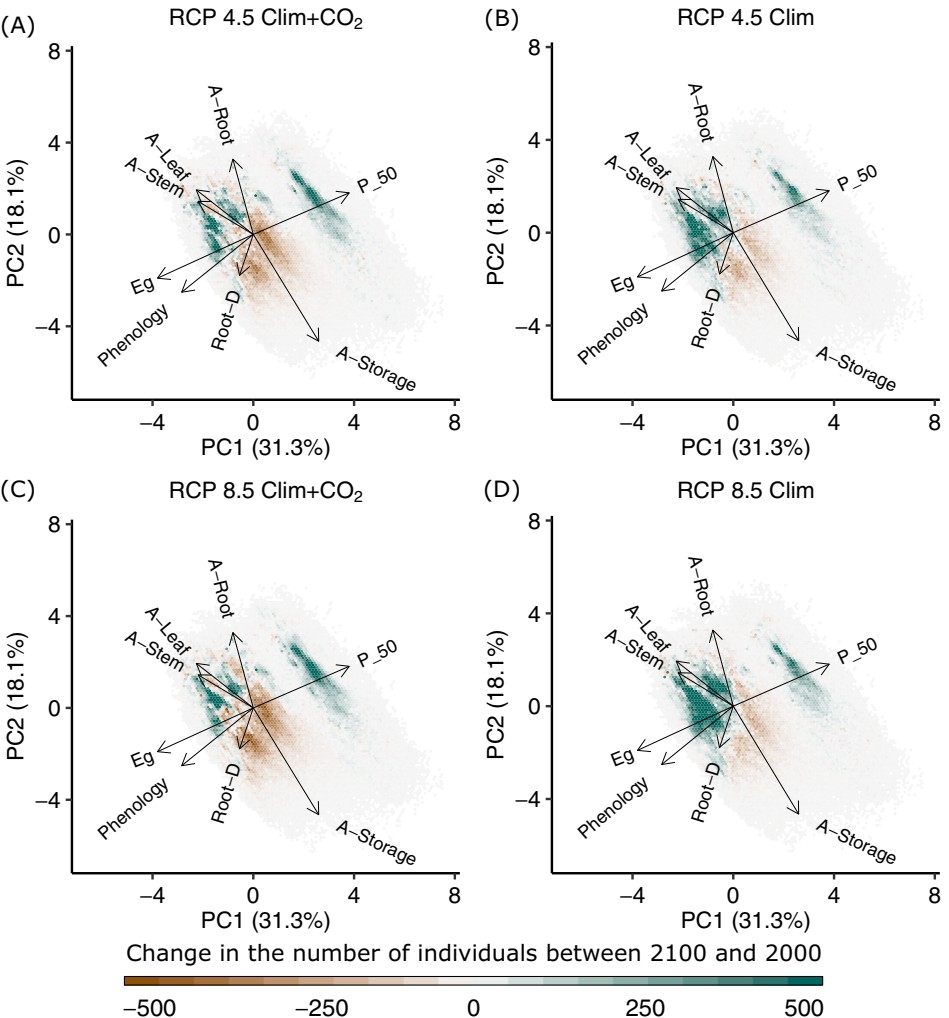

**Fig. 3 | Change in Amazonian functional composition across trait-space.** Principal component analysis for the continental-scale simulation area (Fig. S3) and each climate change scenario: **A** RCP4.5 Clim+CO$_2$, **B** RCP4.5 Clim, **C** RCP8.5 Clim +CO$_2$, and **D** RCP8.5 Clim. All trees in the study region were included in the calculations. Trait space was divided into hexagons. Each hexagon was then overlaid with an empirical 2D kernel density estimate of the change in the number of individuals across trait-space between 2000 and 2100. Traits used as in Fig. 2. Hydraulic-strategy traits align predominantly along PC1, with productive vs.

conservative strategy traits along PC2. For clarity, arrow length was tripled, and changes greater or less than ±500 were set to ±500. Arrows point toward: a higher allocation of carbon to grow roots (A-Root), leaves (A-Leaf) and stems (A-Stem) vs deferred growth via higher allocation to storage (A-Store), less negative water potential at 50% loss of conductance (P50), an evergreen leaf phenology (Eg) (as opposed to deciduous), light-triggered leaf flush (Phenology) (as opposed to water-triggered), increased maximum rooting depth (Root-D). The data underlying this figure are provided in figshare (https://doi.org/10.6084/m9.figshare.26232395).

cannot be fully isolated by this analysis. This limitation is due to differences in initial biomass (Fig. S13), differences in the spatial extent and intensity of future precipitation and temperature changes (Fig. S2B, C, E, F), and differences in the spatial extent and magnitudes of reductions in functional trait diversity caused by the removal of plant strategies (Fig. S11B, C).

Therefore, to answer RQ 2.2, we statistically isolate the effect of functional trait diversity on the climate change-associated reduction in biomass in 2100 (Fig. 4, see "Methods": "Amazonian climate change scenario simulations−Plant strategy removal", Table S6). In keeping with Fig. S13 results, biomass reductions increase with increasing precipitation reductions (Fig. 4). In the absence of increasing CO$_2$, changes in above-ground biomass are more negative (Fig. 4C, D). Irrespective of precipitation reduction or CO$_2$ treatment, functional trait diversity strongly mediates the change in above-ground biomass. To make plausible predictions of diversity effects for the Amazon, we hold precipitation at the approximate mean for area (2500 mm/year) and vary precipitation loss from the approximate mean reduction (750 mm, 30%) up to a reduction comparable with the simulated drought

experiments (Fig. 1) (1250 mm, 50%). Predicted magnitudes of diversity effects thus range from 10% (Fig. 4B with a 30% precipitation reduction) to 34% (Fig. 4D with a 50% precipitation reduction). Empirical evidence indicates that species richness can increase forest resistance to drought and that diversity effects increase with drought intensity[62]. Our simulations reveal similar relationships. We predict that the positive effect of higher functional diversity on ecosystem resistance becomes stronger as drought conditions become more severe (Fig. 4).

**Cross-scale synthesis: drought, climate change, and functional diversity**

Site-scale simulations reveal drought responses in line with observations[16,17] (Fig. 1), however, such extreme multiyear events are unlikely. At the continental scale the model predicts that the carbon sequestration functioning of the Amazon is broadly in line with observed and predicted trajectories[4,56] (Fig. S8). However, the chosen climate forcing data predicts large precipitation reductions[63]. The 50% precipitation reduction (Fig. 4) demonstrates the effect of functional diversity under extreme drought. Such a reduction is not predicted for

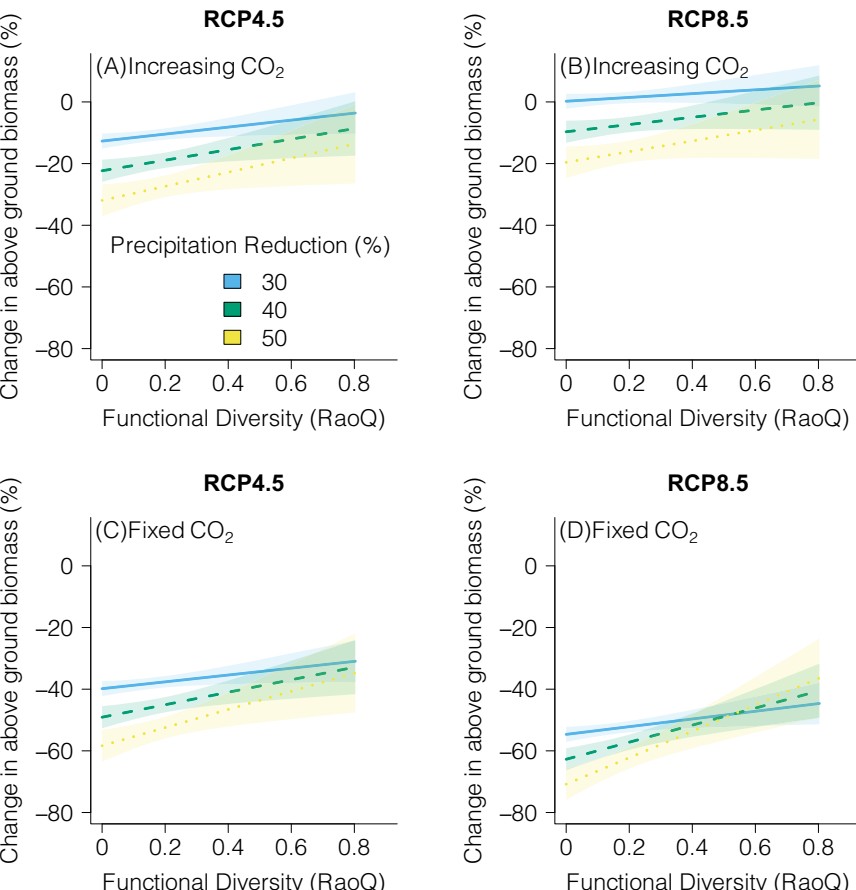

**Fig. 4 | Functional trait diversity-mediated changes in future above-ground biomass.** Predicted change in above-ground biomass between 1990 and 2100 for RCP4.5 with and without increasing $CO_2$. **A**, **B** Predicted percentage change in biomass as mediated by functional trait diversity (Rao's quadratic entropy). Changes are shown at varying levels of precipitation reduction by 2100. **A**, **B** Predictions with increasing $CO_2$. **C**, **D** As in (**A**) and (**B**) but with $CO_2$ held constant at 1990 levels. Predictions are based on a linear mixed effects model (Table S6). Lines are predicted mean responses for the effect of functional diversity

on change in above-ground biomass at varying levels of precipitation reduction. Shading indicates a 95% confidence interval. The range of values for precipitation reductions was based on the approximate mean reduction for the study area of 30% (Fig. S2B, C) up to a value of 50% consistent with site-level drought experiments. See "Amazonian climate change scenario simulations–Plant strategy removal" and Table S6 for further details. 2T indicates that RaoQ was calculated using 2 traits (P50 and Root-D). The data underlying this figure are provided in figshare (https://doi.org/10.6084/m9.figshare.26232395).

Northern South America under CMIP6 SSP5-8.5[64,65]. For SSP5-8.5, CMIP6 models predict conflicting precipitation changes (median − 14.7%, 5 − 95% quantile range − 31.9% to 4.9%). However, Brazilian ESM experiments demonstrated that the interactive effects of Amazon rainforest cover change (savannization) and climate change resulted in a 44% precipitation reproduction across the Amazon basin, showing that vegetation changes can have profound cascading effects on climate. ESMs generally fail to capture vegetation responses to drought[66]. Thus, while significant progress is being made[67], DGVMs and ESMs, our primary tools for predicting future vegetation carbon dynamics and climate change, still require further refinement to capture physiologically and ecologically mediated future dynamics[68], particularly those related to catastrophic and chronic drought.

We demonstrate that trait-based vegetation models often represent underutilised, yet ideal, vessels within which unified theories of community ecology[69] and predictive trait-based ecology[19] can be operationalised to allow improved quantification of the crucial role functional trait diversity plays in mediating ecosystem and Earth System functions[70]. Our results highlight the potential value of an improved integration of the biosphere in ESMs. Historically, in ESMs, the focus on improving the representation of ecological processes has lagged significantly behind advancements in geochemical processes. Our results demonstrate a substantial effect size of functional trait

diversity comparable to the effect of nutrient dynamics on vegetation productivity and carbon storage[71]. Thus, our findings strongly support recent calls to redress this imbalance and prioritise ecological realism in ESMs[68]; in the face of climate change, such improvements are essential to better understand and predict the feedbacks and interactions between the biosphere and atmosphere.

Crucially, we demonstrate the intrinsic linkages between climate change and biodiversity loss. We show that biomass storage in functionally diverse forests is more resistant (up to 34%) to both sudden catastrophic drought and chronic climate change-associated precipitation reductions. Thus, our results quantify the potential benefits of holistic conservation approaches that incorporate the conservation of functional trait diversity[72], particularly traits that mediate plant water status[22]. The conservation of diversity in plant functional traits can ensure continued resistance of tropical forests to drought, increase long-term carbon storage, and help mitigate cascading effects on climate.

## Methods
### Modelling concept and forcing data
**aDGVM2 model description.** aDGVM2 is an individual and trait-based vegetation model which has been fully described in the main text and supplement of Langan et al.[25] (the full model description is also

included here as Supplementary Notes). There are currently four versions of aDGVM2 being used[25,73–75]. In the model, every plant can have a potentially unique set of trait values, and trade-offs between traits constrain individual performance (see Supplementary Notes "Introduction and modelling concepts"). The model behaves like an iterative optimisation algorithm and thus mimics natural selection (see Supplementary Notes "Reproduction, inheritance, mutation, and crossover", Fig. S19). In the model, variability of trait values between individuals in a community is permitted, the set of trait values an individual possesses affects its relative reproductive success, and trait values are heritable. Individuals with poorly performing sets of traits produce fewer offspring. This modelled process of natural selection results in a filtering of the community of individuals and their trait values through time. Additionally, individuals are assigned a species label to ensure reproductive isolation between species. Each year, individuals with the same species label reproduce, whereby trait values can be crossed between individuals and trait values can mutate with fixed probabilities. This iterative process of reproduction, crossover, and mutation through time (see Supplementary Notes "Reproduction, inheritance, mutation, and crossover" in Appendix) allows modelled communities to continuously explore trait space and find trait combinations which may improve relative reproductive success (Fig. S1).

**Model forcing and simulation setup.** aDGVM2 simulates vegetation in 1 ha stands with daily time steps. Therefore, daily climate data are required to force the model. In order to conduct replicate simulations in different model experiments and ensure that daily time series are not identical in all replicates but differ between replicates, we decided not to use available daily time series but create time series based on a reference climatology and anomalies of precipitation and temperature relative to the climatology. As reference climatology, we used the Climate Research Unit (CRU)[76] dataset, representing climate for the 20th century. This dataset provides monthly mean data for temperature, temperature range, precipitation (monthly mean and coefficient of variation), number of rain days, sunshine hours per day, humidity and number of frost days averaged for the period 1961 to 1990 (that is, 12 monthly values per variable).

Creation of anomalies for precipitation and temperature was performed as in Moncrieff et al.[77]; first, we merged monthly time series of these variables for the period 1850 to 2100. For the historical data between 1850 and 2005, we used MPI-ESM-LR ESM[63]. For the future period between 2006 and 2100, we used representative concentration pathways (RCPs) 4.5 and 8.5 using data from the MPI-ESM-LR ESM[63]. These climate simulations were generated for the Coupled Model Intercomparison phase 5 (CMIP5) and were available at 1.9° spatial resolution. For both precipitation and temperature, we fitted quadratic functions to estimate the anomalies over the entire period. Specifically, we fitted functions

$$T_m^f = a + bT_m^2 \quad \text{and} \quad P_m^f = c + dP_m^2, \qquad (1)$$

where $T_m$ and $P_m$ are temperature and precipitation in month $m$ of the climate data, $T_m^f$ and $P_m^f$ are the respective fitted values, and $a$, $b$, $c$, $d$ are parameters describing the quadratic functions. We used numerical optimisation to determine the parameters $a$, $b$, $c$, $d$ such that the sum of squares error between climate data and fitted values was minimised, i.e.,

$$\min_{a,b} \sum_m \left( T_m^f - T_m \right)^2 \quad \text{and} \quad \min_{c,d} \sum_m \left( P_m^f - P_m \right)^2. \qquad (2)$$

Temperature anomalies were calculated in °C and scaled such that the mean anomaly in the reference period 2001–2010 is zero. Precipitation anomalies were calculated as proportional change and scaled such that the mean anomaly in the reference period 2001–2010 is one.

To derive monthly precipitation and temperature time series for each simulated grid-cell, we scaled the monthly values from the CRU climatology (Fig. S2A, D) with the respective time series of the fitted anomalies. As grid cells of the anomalies have a higher spatial resolutions than the grid cells of the simulation, all simulated grid cells within the coarser grid-cell of the anomalies had the same anomalies.

Temperature anomalies (Fig. S2E, F) were represented in °C, and they were added to values of the CRU climatology. Precipitation anomalies (Fig. S2B, C) were represented as proportional change, and they were multiplied with the values of the CRU climatology.

To generate a time series of daily precipitation for each simulation year, we use the stochastic rainfall algorithm provided in ref. 76. This stochastic rainfall algorithm assumes that monthly rainfall is a gamma-distributed random variable where the mean amount of precipitation in an event is the monthly rainfall amount $r_m$, divided by the mean number of rain days in a month $w_f$. The monthly parameter values for the gamma distribution and the mean number of rain days per month are provided in ref. 76. This approach to producing daily rainfall events has been used previously in aDGVM simulations[77,78].

Simulations are run for 1000 years for each grid cell in the study region (Fig. S3), and each scenario and replicate. Our study region is focused on tropical forests in South America. The majority of the study area is centred on the Amazon rainforest. Some areas of tropical forest included in the analysis fall outside the strict boundaries of the Amazon biome. For simplicity, and because of the global recognition of the Amazon as a key tropical forest, we refer to the study region collectively as Amazonian forests throughout the text. To make the number of simulations and time required manageable, tropical forest areas were subset based on national boundaries. Future simulations will take account of the full extent of the Amazon forest that extends beyond the currently subset national boundaries. We conduct a spin-up of 760 years with $CO_2$ concentrations fixed at a pre-industrial (1860) level of 286 p.p.m. to allow vegetation to reach equilibrium. From simulation year 760, $CO_2$ increases following observed increases up to simulation year 900, which corresponds to year 2000 values. Spinning up the model using pre-industrial $CO_2$ concentrations excludes the possibly confounding effects of $CO_2$ fertilisation on emergent vegetation. The effects of chronic, climate change-induced changes to abiotic conditions on the future functioning of the Amazonian rainforests remain uncertain. To assess such chronic changes in this study (Tables S1, S5), we apply RCP4.5 and 8.5 climate forcing[63] for the last 100 simulation years with increasing $CO_2$ as prescribed by the RCP scenario[51]. We additionally run experiments with fixed $CO_2$ to explore the effect of $CO_2$ fertilisation of plant growth on future vegetation of the study area and to disentangle $CO_2$ effects and climate change effects. The specific anomalies and $CO_2$ treatment applied in each experiment are specified in their respective sections and Tables S1, S5.

### Site-level experimental setup and analysis
**Catastrophic drought—simulating through-fall exclusion.** For drought experiments, we followed the individual site protocols given in ref. 20 for the Tapajós national forest (TNF) site and ref. 21 for the Caxiuanã national forest (CAX) site. TNF is located in the state of Pará (2.897° S, 54.952° W), Brazil (Fig. S3). On average, this forest receives 2000 mm a year of precipitation, with a range between 600 and 3000 mm[79], and experiences a prolonged dry season between July and December, where monthly precipitation is generally below 100 mm[20]. CAX is located in the eastern Brazilian Amazon (1.43° S, 51.27° W) (Fig. S3). This *terra firme*, mostly undisturbed, forest receives a mean annual rainfall between 2000 and 2500 mm with a dry season between June and November[21]. At both tropical forest sites, large-scale manipulative experiments were performed whereby plots were isolated from surrounding forest via soil trenches, and precipitation that passed through the canopy was partially intercepted[21,79]. At both sites, repeated censuses were used to quantify changes in above-ground

biomass. Both sites were simulated with a soil depth of 10m[80]. For all site-scale simulations, only trees were simulated. In order to simulate the through-fall exclusion experiments, daily precipitation generated by the stochastic rainfall generator (see section "Model forcing and simulation setup" above) was reduced by a fixed amount. For TNF, precipitation was reduced by 50%[20] for 4 years, while for CAX, precipitation was reduced by 50%[21] for 7 years, with the reductions applied in the simulation year corresponding to the year the through-fall experiments were conducted at each site. Forcing data were scaled using anomalies generated for RCP4.5, following ref. 51 with $CO_2$ increasing in line with this scenario.

**aDGVM2 full diversity simulations.** For each site, the drought experiment was replicated 96 times using different random initialisations of the model (Fig. 1A, B). For these simulations, the trait values of individuals in the initial population were randomly drawn from a uniform distribution for each trait and mutation and crossover were allowed.

**aDGVM2 constrained diversity simulations.** To test the effect diversity has on the response of vegetation to drought, we ran simulations where the model was initialised with the following 18 different numbers of species (1, 2, …., 12, 16, 32, 48, 64, 80 and 96). To initialise species trait values for this experiment, we created a species-trait-pool containing 96 species for each site by identifying the most abundant species in each "non-constrained diversity" replicate and took the mean value for each trait. For each set of simulations with different numbers of species, the abundance of each species in the initial population was approximately equal. The choice of 96 species might appear arbitrary. This number was however chosen to use existing computational resources most efficiently (24 processors per core, using 4 cores).

Each species number treatment was replicated 96 times, giving a total of 1728 simulation runs for each site (Fig. 1C, D). For this experiment, each species had fixed, but different, trait values. Mutation and crossover were turned off (see above and Supplementary Notes "Reproduction, inheritance, mutation, and crossover", Fig. S19). For each of the 96 replicates for each species number treatment, species were randomly drawn without replacement from the species-trait pool.

**Calculating functional trait diversity from site-scale drought simulations.** The species-trait-pool for each site was the one used to initialise the diversity experiment. In simulations that were initialised with more than one species, the number of simulated extant species present in a community before the drought treatment was applied could be less than the number at initialisation (Fig. 1C, D). Thus, for each simulation run, the pre-drought frequency of each extant species from the species-trait-pool was counted. These species-trait-pool and abundance data were used to calculate functional diversity (FD), Rao's quadratic entropy (RaoQ), using the 'dbFD' function in the R package 'FD'[81]. RaoQ represents the mean functional dissimilarity between two randomly selected individuals and was chosen as a measure of functional diversity as it can take relative abundances into account[82,83] and fulfils the criteria of suitability for functional diversity metrics[84].

The following eight traits we included in the calculations of FD: (1) proportion of carbon gained allocated to storage (A-Storage), (2) proportion of carbon gained allocated to stem growth (A-Stem), (3) proportion of carbon gained allocated to leaf growth (A-Leaf), (4) proportion of carbon gained allocated to root growth (A-Root), (5) the water potential at 50% loss of conductance (P_50), (6) a trait which defines whether an individual is evergreen or deciduous (Eg), (7) a trait which defines whether plant phenology is triggered by changes in light or changes in water availability (Phenology), (8) and a trait which defines the maximum depth to which a plant can root (Root-D). For the analysis of functional diversity, Eg and Phenology were considered as asymmetric binomial variables following ref. 85. These eight traits were selected following a principal component analysis[86] based on their

loadings on the first two axes[87]. Relationships between functional diversity and biomass change calculated using these traits are presented in Fig. 1C, D. Principal component analyses using these traits are presented in Fig. 2.

**Examining relationships between functional trait diversity and changes in above-ground biomass.** To test whether functional diversity affects the change in above-ground biomass during drought, and determine the size of the effect, we follow the guidelines for conducting a Bayesian analysis put forward in ref. 88. All models were run with 4 chains and 2000 iterations per chain, with a warm-up (burn-in) period of 1000 iterations. Our hypothesis (H1) is that modelled above-ground biomass change (BM_loss) for each site varies as a function of functional diversity (RAOQ), with both the mean and residual variance of BM_loss varying with RAOQ.

The models (H1) were specified as follows:

$$BM\_loss_i \sim \mathcal{N}(\mu_i, \sigma_i^2)$$

where:

$$\mu_i = \alpha + \beta_{RAOQ} \cdot RAOQ_i$$
$$\log(\sigma_i) = \gamma + \delta_{RAOQ} \cdot RAOQ_i$$

Here:
- $\mu_i$ is the predicted mean for observation $i$, and it varies linearly with RAOQ.
- $\sigma_i$ (the standard deviation of the residuals) also varies with RAOQ and is modelled on the log scale.

The chosen priors for each parameter were weakly informative:
- $\alpha \sim \mathcal{N}(0, 10)$: A normal prior for the mean intercept with a mean of 0 and a standard deviation of 10.
- $\beta_{RAOQ} \sim \mathcal{N}(0, 10)$: A normal prior for the slope of the mean model with respect to RAOQ, with a mean of 0 and a standard deviation of 10.
- $\gamma \sim \mathcal{N}(0, 1)$: A normal prior for the intercept of the variance model on the log scale, with a mean of 0 and a standard deviation of 1.
- $\delta_{RAOQ} \sim \mathcal{N}(0, 1)$: A normal prior for the slope of the variance model with respect to RAOQ on the log scale, with a mean of 0 and a standard deviation of 1.

To assess the importance of RAOQ on both the mean and variance, we compared the model above with a null model (H0) where both $\mu$ and $\log(\sigma)$ are intercept-only models. We used the Bayes factor to compare the predictive performance of our hypothesis (H1) with the null model (H0)[88]. The Bayes factor (H0 vs H1) for the TNF site indicated strong evidence for H1 (bf = 5.445588e+60). The Bayes factor (H0 vs H1) for the CAX site indicated strong evidence for H1 (bf = 1.648138e+74). Thus, our analysis at both sites provides strong evidence of an effect of RAOQ on BM_loss.

We then investigated the effect size of functional diversity on the change in above-ground biomass. At the TNF site, RAOQ reduced biomass loss by ca. 17% (Table S2, Fig. 1C) while also reducing residual standard deviation by ca. 12% (Table S2, Fig. 1D). At the CAX site, RAOQ reduced biomass loss by ca. 32% (Table S3, Fig. 1C) while also reducing residual standard deviation by ca. 3% (Table S3, Fig. 1D).

**Key axes of trait variation, emergent plant strategies, and the removal of rooting niche differentiation.** In aDGVM2, there are four potential phenological combinations. Owing to the emergent trait associations between phenological and hydraulic traits (Fig. 2), we refer to these as plant strategies. These were:
- Evergreen with light-triggered leaf flush
- Evergreen with water-triggered leaf flush

- Deciduous with light-triggered leaf abscission and flush
- Deciduous with water-triggered leaf abscission and flush

For the site-scale simulations, the proportion of these plant strategies was calculated. From the "aDGVM2 constrained diversity simulations" that were run for TNF and CAX sites, we tested whether the removal of below-ground rooting niche differentiation affected simulated forest responses to drought. We:

1. Plotted the maximum rooting depth for each plant strategy (Fig. S5) using the species-trait-pool to examine rooting niches of strategies.
2. Portfolio and evenness effect theory[49] predict that both the diversity of species, strategies, traits, or phenotypes within a community, as well as their relative abundances, affect the stability of communities. We therefore identified replicates in the "aDGVM2 constrained diversity simulations" where all plant strategies were present, and the biomass abundance of the least abundant strategy was at least 20%.
3. Identified replicates, one for each site, that best matched these criteria, i.e., these replicates had the most even biomass abundance of strategies.
4. We then re-ran drought experiments and removed below-ground rooting niche differentiation by setting the values of the traits which define root shape and depth (see Supplementary Notes "Root architecture and water competition", Table S9, and Fig. S6) to the mean values of the deepest rooting strategy at the CAX site (egwt, Fig. S6). The deepest strategy was chosen as constricting rooting to shallower depths may artificially restrict plant available water. To aid comparability of treatments, the same trait values were used for CAX and TNF sites. Trait values were changed on the first day of the year in which drought treatments were applied.
5. We additionally ran a control simulation run where below-ground rooting niche differentiation was removed, but no drought was applied.

Drought treatments were identical to those described in "Simulating through-fall exclusion". The solid black line in Fig. S6 displays the response of chosen replicates from the "aDGVM2 constrained diversity simulations", where no trait manipulation was applied. The dashed grey line displays the simulated responses to drought when rooting niche differentiation was removed and drought was applied. The dotted grey line displays the simulated responses to the removal of rooting niche differentiation when no drought was applied.

### Entire study region experimental setup and analysis

**Amazonian climate change scenario simulations—full diversity.** Simulations were run for 1000 years using the standard model parameterisation. This parameterisation initialises the model with 50% trees and 50% grasses to avoid making prior assumptions about initial vegetation distributions[25]. A spin-up of 760 years was run with $CO_2$ concentrations fixed at a pre-industrial (1860) level of 286 p.p.m. to allow vegetation to reach equilibrium. RCP4.5 and 8.5 climate forcing (precipitation and temperature anomalies (Fig. S2B, C, E, F)) was used both with and without a $CO_2$ impact on plants (Fig. S2G). For simulations without a $CO_2$ impact on plants, $CO_2$ levels were held constant at 380 ppm; for simulations with a $CO_2$ impact, we increased $CO_2$ as prescribed by the RCP scenario following ref. 51.

**Community trait re-assembly under climate change scenarios.** To assess how climate change alters the occupancy of trait-space a principal component analysis was conducted using a customised version of the code from package 'ggbiplot'[89] with changes in density calculated using the package 'Hexbin'[90]. Every simulated tree in the study area was included in the analysis. Figure 3A–D shows an empirical 2D kernel density estimate of the change in the number of individuals across trait-space per hexagon between 2000 and 2100 for the entire study region.

**Amazonian climate change scenario simulations—plant strategy removal.** Site-scale analysis of relationships between functional trait diversity and precipitation reductions indicated significant interactions (Fig. 1). To explicitly test these relationships at the continental scale (Fig. S3), we constructed a set of experiments in which we manipulated functional trait diversity (Figs. S11, S12). The RCP climate change scenarios 4.5 and 8.5 were re-run twice with both increasing and fixed $CO_2$. We reduced the maximum potential volume of trait-space by: (1) removing deciduous water-triggered strategies; and (2) additionally removing evergreen water-triggered strategies. Removing the possibility for plants to have trait values that define these strategies produced a spatially patterned reduction in functional trait diversity (Fig. S11). Functional diversity was calculated as Rao's quadratic entropy using the traits which define a plant's maximum rooting depth, and the xylem water potential at which 50% of conductance is lost (See Supplementary Notes and Table S9 for a full model description and list of traits). These traits were chosen as they align along the hydraulic traits axis and were not directly manipulated by steps (1) and (2) above.

In order to initially examine the response of reduced diversity simulations to precipitation reductions (Fig. S13), sites were selected across all scenarios where the amount of above-ground live biomass was between 150 and 200 t/Ha in the year 1990 and precipitation in the year 2100 was at least 250 mm lower than the 1990 level. To examine whether the level of precipitation reduction affected responses, data were further subset into loss of precipitation intervals; these intervals were chosen as the approximate quantiles of precipitation reduction in RCP4.5 (Table S4).

To isolate the interactive effect of functional trait diversity on the percentage change in biomass between 1990 and 2100, we fit a linear mixed effects model[91] (Table S6, Fig. 4). We followed the methodology for fitting this model described by Zurr et al.[92]. Percentage change in biomass between the two time points was the dependent variable. The independent variables were: (1) 'Precipitation 1990' which was the annual sum of precipitation in each simulated grid-cell in 1990; (2) 'RaoQ' which is functional trait diversity described below; (3) 'Precipitation Change' which is the reduction in precipitation in mm by 2100; (4) 'RCP' is a factor which defines whether the climate forcing applied corresponded to RCP4.5 or 8.5; (5) '$CO_2$' was treateated as a factor which defines whether simulations were run with increasing or fixed $CO_2$. 'Site' represents each 0.5° grid-cell and was added as a random effect. All independent variables were scaled prior to model fitting. Model selection was done by first fitting a saturated model containing all interaction terms and removing them based on the AIC value[92]. No interaction terms were removed following this protocol. The fitted model produced adequate marginal and conditional $R^2$ values of 0.56 and 0.67. Residual values appeared homogeneous and normally distributed. Deviating from the recommended model fitting methodology and fitting a simpler model containing only first-order interactions between functional diversity 'RaoQ' and the other independent variables yielded a similar effect size for 'RaoQ' as that presented in Fig. 4. Fitting a Bayesian model also yielded a similar effect size for 'RaoQ' as that presented in Fig. 4. We have chosen to present the most frequent version for simplicity. To present plausible predictions of the differences between high and low functional diversity (Fig. 4) we used approximate mean values for annual precipitation in the 1961–1990 time period (2500 mm, mean in data 2456 mm, Fig. S2A) and precipitation reductions in 2100 (Fig. S2B, C) close to the mean reduction in the data (750 mm, mean in data 713 mm). A reduction of 30% corresponds to a 750 mm reduction, a reduction of 40% corresponds to a 1000 mm reduction, and a reduction of 50% corresponds to a 1250 mm reduction (Fig. 4).

**Amazonian climate change scenario simulations−calculation of functional trait diversity**

To calculate functional trait diversity for the entire study area (Fig. S11), it was necessary to create a species trait-pool and abundance list for use with the 'dbFD' function[81]. RaoQ was chosen as a measure of functional diversity as it can take relative abundances into account[82,83] and fulfils the criteria of suitability for functional diversity metrics[84]. It represents the mean functional dissimilarity between two randomly selected individuals. In the model, species are defined by a simple index, which is used to ensure reproductive isolation; these indices are not consistent across sites. To generate a species trait-pool, a dataset was compiled that contained every tree individual in the study area for each RCP scenario and treatment (see "Amazonian climate change scenario simulations−Plant strategy removal"). This dataset contained all trees from the simulation year 2000 for RCP4.5 and 8.5 with increasing and fixed $CO_2$ at three diversity levels each (Full model, deciduous water-triggered leaf phenology removed, deciduous and evergreen water-triggered leaf phenology removed). The function 'Clara' in package 'Cluster'[93] was then used to draw 400 clusters. The mean trait values of each returned cluster were used as a species-trait-pool, and the frequency of individuals per grid-cell, RCP scenario, $CO_2$ treatment, and diversity level in each cluster was returned to generate abundances to calculate functional diversity. Two hydraulically associated traits were used to calculate functional diversity across the study area. These traits were: (1) the water potential at 50% loss of conductance (P_50) and (2) a trait which defines the maximum depth to which a plant can root (Root-D). Trait selection was hypothesis-driven[19] as these traits influence plant hydraulic performance and compose part of the axis of variation emerging from the model (Figs. 2, 3) associated with hydraulic traits. The leaf phenological traits which were directly manipulated were not included in the calculations of functional diversity. Thus, Fig. S11 highlights again above and below-ground coordination of traits as the displayed reductions in functional diversity are caused by the manipulation of leaf phenological traits, which are not a priori linked to rooting depth or P_50. Community trait means were also returned from the 'dbFD' function (Figs. S9, S10), whereby P_50 values were used to convert to specific leaf area (SLA) values.

**Statistics and reproducibility.** No statistical methods were used to predetermine sample size. No data were excluded from the analyses. The experiments were not randomised. The investigators were not blinded to allocation during experiments and outcome assessments. All analysis was performed using R Statistical Software[94] version 4.4.2.

**Reporting summary**

Further information on research design is available in the Nature Portfolio Reporting Summary linked to this article.

## Data availability

The processed data underlying all figures and analyses are provided in figshare (https://doi.org/10.6084/m9.figshare.26232395). Savanna distribution data can be requested from the authors of Lehmann et al.[95]. Non-aDGVM2 specific leaf area data can be downloaded from the supplementary data of Dong et al.[96].

## Code availability

aDGVM2 (v1.0-LL) code available here (https://doi.org/10.5281/zenodo.16265490)[26].

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

## Acknowledgements

We thank Mirjam Pfeiffer, Camille Gaillard, Dushyant Kumar, Timo Conradi, and Jeremy Lichstein. Carola Martens spotted the mismatch in soil texture parameters. Senckenberg Biodiversity and Climate Research Centre directly fund L.L.

## Author contributions

Conceptualisation: L.L., S.H., and S.S. Methodology: L.L. designed the study with contributions from S.S., S.H., and T.H. Software: L.L., S.S., and S.H. developed the model. L.L. and S.H. developed plant hydraulics sub-modules. Validation: L.L. Formal analysis: L.L. Investigation: L.L. Resources: Senckenberg Biodiversity and Climate Research Centre provided computational resources. Data curation: L.L. Writing—original draft preparation: L.L. and S.H. drafted the manuscript. Writing—review and editing: L.L., S.H., S.S., T.H. all critically reviewed, commented and revised the manuscript. Visualisation: L.L. prepared all visualisations with contributions from S.S., S.H., and T.H.

## Funding

## Competing interests

The authors declare no competing interests.
