## [Peer Review file · Nature Communications]

Amazon forest resistance to drought is increased by diversity in hydraulic traits

Corresponding Author: Dr Liam Langan

Version 0:

Reviewer comments:

Reviewer #1

(Remarks to the Author)

The manuscript "Hydraulic-trait diversity increases forest resistance to water deficits" brings the discussion to a very important subject that is associated to the forests response to predicted climate changes, focusing on biomass impacts. Moreover, it includes an aspect of functional diversity on forest response to drought.

The biomass loss over time, in addition to biodiversity loss, involves CO₂ release in a world where everyone is trying to sequester CO₂. So this results point to an area of global interest, and brings a recent approach of vegetation dynamic models. This seems to be very promising to improve vegetation response to climate change and could be replicated to predict other vegetation response across many tropical ecosystems.

Thus, I believe the manuscript would have a good impact in advancing studies of vegetation modelling in a changing world and has the potential to be published in Nature Communication. However, it needs to improve in many aspects, as commented below, including some text organizing and more clarity on methods and results, to make it more readable and understandable for the broader public of this journal.

Major comments (I listed the topics below but they are better explained together with the minor comments along the manuscript sections):

1. Text clarity. There are many parts in the text that are not very clear enough, and maybe it would need more writing to exactly explain what the authors mean. Sometimes it looks like is very clear to the authors what they are talking about but for who is reading it is not.
2. Methods. The methods need improvement with more details about what was going on in each step of this paper (i.e. traits involved, how the trait vs. species association, climatic changes scenario)
3. The results for the Amazon. Needs some checking if it was only data for Amazon or included data from other biomes in Latin America.
4. I think the discussion needs more biological association with hydraulic strategies, not only too descriptive.

Minor comments:

Main text

Summary paragraph – It is a bit confusing, it needs some rewriting.

Lines 11-12. The authors use the term "Biomass resistance", maybe better change for the forest resistance to biomass loss up to 25%. Here the author means the less diverse forests lost 25% more biomass, isn't it?

In many parts of the text is not very clear the way the authors present the results. They do not say exactly what was observed, using other terms.

Line 16 – The authors call the removing phenological groups (triggered by water) as hydraulic traits. The deciduousness is a strategy to avoid water loss, but may use the term of phenology strategy instead of hydraulic trait. When they say hydraulic traits, the readers would expect other traits as well. I think they should point out exactly the trait they removed (i.e. the water associated phenology strategies).

Lines 29-32 – I would use more forests studies as reference here instead of grassland based ones. Although the ecological aspects can be the same their response and time of response may be very different. Some forests references could be:

Sakschewski, B., Bloh, W., Boit, A., Poorter, L., Peña-Claros, M., Heinke, J., Joshi, J. & Thonicke, K. (2016) Resilience of Amazon forests emerges from plant trait diversity. *Nature Climate Change*, 6, 1032-1036.

Xu, X., Medvigy, D., Powers, J.S., Becknell, J.M. and Guan, K. (2016), Diversity in plant hydraulic traits explains seasonal and inter-annual variations of vegetation dynamics in seasonally dry tropical forests. *New Phytol*, 212: 80-95.
<https://doi.org/10.1111/nph.14009>

Lines 36-39. Long and confusing sentence. Needs some rewriting or punctuation.

Lines 40 – 45. Needs some rewriting. (Lines 40-41. What diversity are you talking about: functional diversity or diversity of species? The may be correlated but it is good to keep the same term along the text. Also, what major barriers do the authors refer to? Lines 42-43. Symptomatic of the problem - What the authors mean with this.)

Lines 47. Although I know the connection, the hydraulic traits are added without any link of it with plant response to drought. Maybe needs a sentence linking the different hydraulic strategies and how they can deal in different ways with drought. As this will be one of the main topics of the work needs more discussion about it. The first 3 three paragraphs can be summarised to include more details about it.

Lines 52-54 it is not clear which parameters are considered genotype and phenotype. When you say parameters that code for how individuals respond to environment I did not get it before reading all the paper. This should be linked better to the methods explanation and better explained here, as it needs to be clear enough in the main text at least for a brief understanding.

At the end, I think all the traits are selected based on the phenotype expressed because it will 'determine' their response to the environmental changes simulations. So, even the phenology, if it is been triggered by water or light, will also reflect on some physiological and molecular changes to receive signals from the environment. So you add this information in the model with will be associated to a genetic definition, but at the end, the response will be determined by the phenotype. All this needs to be more clear in the manuscript.

It is also important to make it clear that although the model is based on individual, the selection is made at species level, right? (see some other comments about this part of the methods below).

Line 55 – need some explanation of this term that you are using: Genotype-traits. May say for now on we will use the term Genotype-traits to refer to a phenological trait (deciduous or evergreen), which is triggered by water or light.

Line 62-63 – Was the distribution of the biomes based on the biomass estimates? Or phenotypes? This sentence is a bit without connection here. Maybe introduce more the model and how it can be used to estimate the vegetation response to some catastrophic events and then link this with the first analysis of this work, which was to explore the drought experiments in model and compare the ground and modelling data.

Line 65 – level and rate? Does level refers to the absolute biomass loss? Maybe be more specific on what you mean.

Line 66-68 – the authors use the word diversity in many parts but it would be more clear if the refer to what diversity they are talking about. Traits diversity or diversity of species? At the end it would be the same because I think the species diversity here are linked to functional/trait diversity, or not? Maybe it would be good to have a sentence saying this at the beginning.

Line 67 – Maybe list here what traits were considered in this first analysis.

Line 70 – Isn't the species associated with traits diversity? Isn't this a model premise? The reduction of species was also a result from the drought?

Figure 1- Which year were the observations result from in both drought experiments. Maybe add more details on the figure 1 subtitle about the species number treatments (18 in total right?) It would be good if the reader did not need to go to the methods to understand it. The Y axis does not need to say reduction as the numbers already point to a negative value. Maybe use Change in Above Ground Biomass.

Lines 77-79 - When you are talking about root, stem and leaves allocation you are talking about root, stem and leaves allocation in biomass associated to growth, right? Because when you say allocation this can be for growth or for storage, but they you consider storage the other extreme off the trade-off.

If you are talking about more acquisitive strategies you may be talking about higher growth investment. This is not very clear in the text.

Also, about the allocation to roots, this may not be completely related to acquisitive strategies. I would be careful when assuming that.

Lines 80-81 – The authors use the term matric potential for plants, but I guess they mean to say water potential. The matric

potential is a component of the water potential. In functional ecology, the P50 is a measure of water potential, when the plant loses 50% of its water transport capacity, so the term water potential would be more appropriate.

Lines 81-83 – Please add references for this. This is not resultant from you analysis right?

In lines 80-91 – It is not clear if you include all these data in the model, or you are just mentioning traits that may be correlated to it. Try to make clear of what is your result and what is discussion.

Lines 96 – Add ‘different strategies’ or ‘higher diversity of strategies’ that influenced the community resistance to make the sentence clearer.

Lines 97-98 – The authors are saying the same thing as a before. The resistant communities showed higher-levels of functional diversity.

Figure 2 – It would be good to know in which direction the arrows for category Eg and Phenology are going to. It is confusing, maybe use the term phenology as a determinant of evergreen or deciduous strategy and use another term for saying if the trigger is water or light. I think it would make more sense, but it is just a suggestion.
Is each colour + symbol one different species?

About the sub-section division: Maybe the sections of the text could be specific about what spatial scale the authors are talking about. Also the traits are the same as before I imagine? Maybe mention that?

Lines 101 – it would be good to have a line explaining why the use of the different terms catastrophic and chronic and which parameters they have changed. Many things are not very easy to understand what the authors are talking about. I would suggest saying things like; we will refer to this xxxx from now on as xxxxx. Or just keep saying the same thing; even if it seems to be repetitive, it is better then making it not understandable.

Lines 102 - About the scenarios RCP 4.5 and 8.5 I suggest the authors to give a better explanation of what is each one in the methods, and what parameters are they changing in the model.

Lines 105 – Also about the CO2 increase, I think you need to add more information about how much it was considered. I did not find any additional information in the methods.

Figure 3 – Again, the authors should indicate the direction of the Eg and Phenology in the subtitles.

Lines 107-111 – Does the results from figure 3 have the same data of figures S3? It is not clear whether the authors included other biomes (non-forested). The maps on Fig S3 show a more broad analysis over the Latin America, with points in other biomes, including non-forested biomes.

Also, we can see on Figs S3B-C there are different responses for some scenarios across different biomes. Part of the Amazon reduces the biomass, despite an increase in biomass in other areas. How did the authors deal with these different responses?

Lines 112-113 – trend of what? It is not clear in each direction the authors are referring to.

Lines 111-115 – From the Fig 3, the results point to an increase in biomass in the scenarios with CO2 increase for the species with higher growth and higher P50. Aren't these traits associated with more acquisitive strategies? And the scenarios with fixed CO2 the phenology and root depth are associated with higher biomass loss. It is difficult to interpret the direction of these phenology and Eg traits as it is not very clear in the text neither in the legend what are their direction.

Maybe make the response directions more clear.

This is an interesting result as the CO2 increase may favour some species strategies. Even more hydraulic vulnerable species (which are usually associated with higher acquisitive strategies) could have been favour by the increment of CO2 as they can maximize their photosynthesis without needing to lose much water. In higher CO2 plants can increase their water use efficiency, counterbalancing the reduction in water availability. I miss some discussion about plant functioning and ecosystem response, as it is the purpose of the paper.

Figure S6 – What is absolute % change related to? Biomass?

Also the density distribution made for each scenario is associated to the number of model runs? Sorry, but I did not understand it very well. Needs more information in the subtitles about what is going on there.

According to all PCAs results there are an increase in the biomass and individual numbers in other strategies that are not associated with phenology. Maybe I got something wrong.

Lines 124-126 - This is related with the different responses that you have in different areas. Here you say that the biomass reduced in 40% in Amazon, but in lines 108-109 you say the biomass increases.

Maybe the authors need to mention this heterogeneity more clearly in the text, including the spatial information about the data they used for these results. The figure 3 is only for Amazon? Or it has all data? Maybe separate the analysis per biome.

Lines 132-137 – Not sure about this term 'collapsed'. It reads weird for me and not very clear. Maybe say you combined or not distinguished some traits. And just say that in the first step you removed one strategy or other, instead of preventing the emergency of a certain trait? Sometimes it is better to say straight what do you mean instead of used other words that can make it confusing.

Lines 151-152 – The authors discuss about niche differentiation. How much were the deciduous traits associated with the other strategies? Because the phenology triggered by water or light is probably associated to some plants strategies. The deciduousness is a strategy, but the factor that triggers it is more associated in this case with how is the traits that will receive the perception from outside (environmental trigger) and will reflect on the leaf shedding. So for example, the root depth, could be the main trait associated to get the response of how much water is in the soil (at the soil root surrounding the root), as when there is no water enough in the soil the plants lose their leaves. How did you deal with this in the paper? The niche in this case they would be more associated to the root that to the phenology indeed.

Also, I would not call only the phenology (which was the focus of this last part of the manuscript) as a hydraulic strategy. It can be part of it associated to the water demand, but hydraulic strategies are broader, and involve other levels of water uptake, water transport, and capacitance.

Methods

Model description section

It was nice the section about this model, especially as it is a new approach and apparently very promising. It is good to know the possibilities of the model, but it need to get clear if the authors used all the utilities in the current manuscript. I think the part of mutations and cross over were left aside, so maybe include it here.

It would be interesting to know exactly all the traits that were use as an input, as well as some clarification about how the species links with traits. As far I understood, each species has one functional trait, right? Or each species is a combination of traits?

Also it is not clear how exactly the selection operates and the probability of changing a trait? It would be nice to have more details about how the model really works with this evolution aspect. Is it at individual level, or species level?

Lines 304-305 – Here you say it is possible to have variability between individuals in a community. Is here the variability intra specific (within the same population)? Or the variability within the community is only the variability across species? So is the selection operating over species?

When you say trait values can mutate at fixed probabilities, in which proportion these mutations affect the phenotype? As the phenotype is where the selections are made, right?

Model forcing and simulation set up

Line 316 – write what CRU stands for.

Line 324 – write what RCP stand for.

In this section is good to punctuate what simulations are associated with each part of the study. Maybe at the beginning after the model explanation, it would be nice to have a brief resume of the parts of the study and what was analysed in each section.

The terms catastrophic and chronic could be used here to say what you are talking about and if the simulations only involved the scenarios or they had additional precipitation analysis, like the case of your last analysis. I assumed the precipitation was only manipulated in the last analysis, right?

Site level and experimental set up and analysis

It is good that you added a section about the experimental analysis, but maybe add more detail about the ground data that you used for the comparison.

Line 365 - Not clear the combinations you have used here - is this 1 to 12, and he following combinations? 18 combinations? Maybe say it first.

Line 370-371- It is not clear for me whether the species has one trait only or they have multiple combinations of traits. This can be sorted as my previous comment when you explain about the model. When the author use species-trait-pool, depending it would be better to keep only species-pool. I understood the species do not repeat, but I guess some traits may repeat, right?

Was the mutation off for all this manuscript, maybe make it clear when you are talking about the model.

Lines 382-391 – These are the model traits, no? Maybe it should be included in the model explanation. Were there any options of traits? It is good to know why the authors used these ones and not other traits. What was the criterion for that?

Lines 397-399 confusing

Lines 404-407 – A better explanation about this should be added at the model explanation as well. It would be good to know how these traits are associated in the model.

Lines 409 – Why these types did not occur? Were they excluded at initial runs of the model? Maybe discuss a bit more about this analysis as it was not very discussed in the main text.

Lines 425 and 427- Give more details about these scenarios. Maybe at the section about Model forcing and simulation set

up.

Lines 463-464 – Why this difference between the two scenarios?

Check in these analyses if all the points used were the ones in Amazon.

Line 465 – Maybe separate this removal of strategy part as another section, as apparently there was an additional change in precipitation, right?

Although there were many issues about more clarity in the manuscript it is a very interesting and important manuscript.

Reviewer #2

(Remarks to the Author)

General comments

The manuscript Langan et al. addresses the role of diversity for vegetation resistance to water deficits - a timely and important topic. The authors find, through comprehensive modelling and analyses, that hydraulic trait diversity plays a critical role for tropical forests to withstand droughts. If substantiated, the findings are novel and noteworthy. However, the paper fails to present the methods and results in a clear, understandable and comprehensive manner, which detrimentally affected my ability to assess the soundness of the research design and methodology, and the significance and scientific robustness of the conclusions. In particular, I recommend the authors to address ambiguities in definitions, unclear descriptions of model validation/evaluation and unclear descriptions of the comparability of the experimental sites and the large-scale simulations. Some additional explanation of the rationale and the results in view of previous literature, as well as additional basic plots of input data and study area would also be useful, as discussed below.

For example, given the focus on diversity, a precise definition is essential. However, a variety of terms regarding diversity are used in the manuscript, sometimes seemingly referring to different concepts and sometimes used interchangeably (e.g., diversity, functional diversity, trait diversity, hydraulic trait diversity, drought-diversity, phenological strategy diversity). “Constrained diversity” is also seemingly used to refer to different types of treatments (one for the experimental sites, and one for the large study area), which is confusing.

Langan et al first run simulations in the experimental sites to reproduce the level and rate of biomass loss, referring to Fig. 1a and 1b to state that this is achieved. It is unclear to me how Fig 1a and 1b shows this. How should the reader interpret the “horizontal dashed line” that is supposed to show observed reductions in terms of “rate of biomass loss”, given that it is fixed over the entire period? How comparable are the “96 replicates simulations” with the actual experiments at the site? How can the model results be trusted given the vast spread in model simulations (particularly for the CAX site in Fig 1b, where the min and max even seem to be cut off)? The model validation and evaluation step is critical for a model-based study - if the reader is to be convinced that the model outputs can be trusted, the model validation and evaluation needs to be robust with regard to the type of outputs of interest here. Furthermore, in view of the validity and robustness of the model simulations, the authors could also discuss why the biomass can increase in the model (Fig S1) when observation-based studies indicate that the net carbon sink has peaked in the Amazon already in the 1990s (see Hubau et al 2020, which Langan et al also cites).

With regard to the precipitation modelling component of the work, the authors used CRU data for the present and MPI precipitation output for the future, and used the gamma distribution from New et al (2002) to generate daily rainfall patterns. The validity of the gamma distribution assumption can be questioned, given that the character of precipitation is projected to change under climate change. Nevertheless, if the model responds to droughts at the monthly-to-seasonal scale (similar to in the real world), the choice of daily precipitation model should not critically affect the results. Please consider referring to work or comment on the sensitivity of the model to daily variations in precipitation. Please also consider providing plots and comparisons of the precipitation inputs used, as well as clarifying the input data in existing plots. It is for example unclear what precipitation data were used in Fig.1 (In Fig 1b, the years extends to 2010, but it is stated in the Methods that CRU is until 1990, the historical MPI output until 2005).

Specific comments

Recommend adding an overview table of all the experiments in the SI (including information such as data input, biodiversity model set-up, time period, drought length/timing, and study area). To would help the reader to grasp how the simulations for the “experimental sites” and the “future climate” differ.

Please consider providing an explanation of the use of 96 species.

Please consider adding information When did the droughts occur? (Fig. 1)

Fig. 4 What is constrained (strategy removal)? What is the baseline (without precipitation reductions)?

Please consider adding spatial map of functional diversity, including model initial conditions.

Please consider adding more cross-ref to Extended Data and add legend to figures. (Needed for example in Fig. 2)

Suggest adding a map of the location of the sites and the study area for the “future” simulations.

The studies referred to previous studies that link diversity and stability of ecosystems and drought and biomass reduction. However, the reasoning for why the two should be linked, i.e. between diversity and biomass reduction due to drought needs to be more elaborated. It seems that only Isbell et al., 2015 makes this connection. This lack of reference also makes it difficult to discuss the results, other than the validation. For instance, Powell et al., 2013 explores what might be missing from estimating biomass reduction due to drought. How would this study perform better in representing the plant hydraulics

compared to Powell et al., 2013.

Similarly, characterization of diversity in mediating responses of tropical forests to drought in comparison to grassland is not discussed.

Figure 2: four clusters to be mentioned in the figure directly, is this referring to ecological strategies on page 24.

What is referred to as biomass re-assembly here? And why is it mostly referred to biomass reduction in Figure 3?

Please consider using more commas and shorter sentences.

Reviewer #3

(Remarks to the Author)

Review for "Hydraulic-trait diversity increases forest resistance to water deficits"

This manuscript concludes that hydraulic trait diversity increases forest resistance to water deficits. To support this conclusion, the authors conducted a set of dynamic global vegetation model simulations (aDGVM2-Pi) for catastrophic drought and chronic drought response in tropical forests. To interpret the results from model simulations and investigate trait characteristics, the authors used hierarchical cluster analysis (PCA and HCPC as noted in their methods) for 8 traits.

This is an interesting topic. But I have the following concerns:

First, the results and conclusions are solely based on the model simulations. But the observational evidence is almost absent. There is only one observational data point to compare with the time series simulations for the biomass response to catastrophic drought in Fig 1. Incidentally, in which year was the AGB reduction observed? 2005? The rest figures are purely based on model predictions. Even if in Fig 1, the model simulations have multiple replicates, and their trajectories are widespread: from almost no drought response to dramatic drought response. I am not sure whether the mean agreeing with the observation is a coincidence, and the simulations at the site CAX seems to be much worse than the simulations at site TNF. In addition, what's the accuracy of the simulated aboveground biomass (I am not asking the accuracy of biomass change)?

Catastrophic drought should be some intense water shortage that happens suddenly. Not sure why the plants do not respond in the beginning years, but the lagged effects last for years. I am a little bit confused whether Fig 1 shows the short-term effect or the lagged effect of the catastrophic drought. Need to define what the catastrophic drought is.

Second, the model is a black box. Even if after reading the method parts, including the "aDGVM2-Pi model description", I was not sufficiently informed, not to mention to be convinced by the presented results in the main figures. The authors did not provide the information about the exact hydraulic mechanisms considered in the DGVM, but only referred all these mysterious to a previous paper.

In fig 1, there is also a need to explain Rao's quadratic entropy, which might be a standard metric in ecology but not a common metric in the broad readers for Nature Communications. Also, is the simulated species richness accurate?

In Fig 2, what is the meaning of color? How is the arrow related to the dots? Why don't the arrows point to the center of the cluster?

L79, "the second axis revealed hydraulic-strategies which closely match observed strategies across Amazonia", please explicitly show how closely they match.

L83-85: do not understand this sentence. "Here, coordinated..."

L94: What is the hydraulic-strategy niche differentiation? Too vague. Why is the drought resilience attributed to the richness of diversity but not the hydraulic traits of certain species, such as isohydricity? At least need to exclude these confounding effects other than diversity.

L101: According to the description, the CO₂ effect is only included atmospheric forcing from the MPI model's RCP simulations. The DGVM model (which simulates the land processes) is decoupled with those atmospheric forcings. In reality, there should be land-atmosphere interactions/feedbacks. I am not sure if the DGVM simulation has a changing CO₂ concentration. If CO₂ is only changing in atmospheric forcing from RCP simulations and the DGVM does not consider CO₂ changing, then the experiment only considers the radiative effect of CO₂ (mostly adverse effect: too hot in tropical regions). Only if the DGVM has a changing CO₂ can it consider the CO₂ physiological effect due to gas exchange (which will affect RuBisCo activity). But the question is, how does the DGVM model parameterize the CO₂ physiological effect on water and carbon cycling?

Fig 3. The colors and clusters make no sense to me. What's the connection between fig 2 and fig 3? Can you explain Dim vs PC? Why does P₅₀ follow dim2 in fig2 while it follows PC1 in fig 3? Could you also define what re-assembly is?

L124: CO₂ feralization on what? This is another arbitrary argument.

L139: "...the biomass differences between full and constrained diversity are influenced by the magnitude of reductions in precipitation and climate change scenario (Figs. 4, S9), and that differences were higher in simulations run without CO₂

fertilized plant growth (Figs. 4, S9, Tab. S3).” This is the only explanation for Fig.4. I wonder can we believe these simulations? That’s intrinsic mechanisms? Since this is a modeling study, the authors should take the advantage of the nature of the mechanistic-driven model in explaining these phenomena but should not making hand-waving implications/conclusions.

In summary, considering the issues I raised above, I do not recommend publishing this manuscript in Nature Communications.

Version 1:

Reviewer comments:

Reviewer #1

(Remarks to the Author)

Dear authors and Editor,

The manuscript "Hydraulic-trait diversity increases forest resistance to water deficits" had significant improvement. It is an interesting and important effort to include hydraulic traits in models, and it can bring a lot of advances in predicting vegetation response to climate change.

However, there are still points to be clearer and written more directly. I know the model is very complex, and it is a challenge to say things understandably and to be concise at the same time.

I miss the aim of the work (the hypothesis to test or the question to answer) to be more emphasised in the introduction. You mention the problems, but not exactly what you want to reach with your work. Is it just a method improvement, or there are ecological questions?

It is a big challenge to add and represent the diversity of traits and trade-offs that yet are not fully understood. One point that is not clear is how the trait data presented in Table A3 was determined. The authors refer to the Genetic Optimization Algorithm but do not explain it. Maybe I missed something. Strangely, the P50 range is the same for trees and grasses. I wonder what the references were to create this range of values, for all traits.

So savannas were removed from the model, were the grass traits removed as well?

There are many hydraulic traits published for Amazonia so far, including Tapajos and Caixuanã, and some root depth data. I was wondering if it would be possible to change these parameters with some real data.

Another point about the traits aspect that I did not get very well: the strategies categories were directly used in the model and the traits that compose these categories as indirect, right? But then, there is a part that they focus on root traits. Why not try other traits (i.e. P50)? If you are focusing on hydraulic traits maybe you could use only them. It depends on your question.

Regarding the traits trade-offs and the traits within each group, there are some contrasting data from Tapajos linking higher embolism resistance with lower root depth (Brum et al. 2018). Hydrological niche segregation defines forest structure and drought tolerance strategies in a seasonal Amazon forest - 10.1111/1365-2745.13022. This result seems to contradict the trade-offs represented in the model. Maybe it would be good to show in a diagram how traits co-evolve in the model to make it easy to understand.

It is difficult to represent traits in functional groups, as traits are multidimensional. Even the deciduous and evergreen groups have been challenging this segregation into functional groups. But I understand that it makes it easier to include in the model. Of course, I know it is impossible to represent all this multifunctionality in models, but it needs to be discussed.

In the cluster analysis, the main traits driving the groups were also included. I wonder if the clustering result would be the same removing the Phenology and the Eg and using them as colours and symbols and seeing how they relate with the traits.

One point that there was some confusion before were the terminologies used in the text. The points raised before are clearer now, but there are new terms, such as niche. When you talk about niche segregation, are you referring to the niche segregation caused by different root depths? It gets confusing because sometimes you say root niche differentiation and others just niche differentiation, and hydraulic-strategy niche. So, there are many terms here with different meanings. It is good to keep consistency.

The use of the terms experiment and simulation also confuses the reader. Maybe use only simulations?

In the conclusion, the first paragraph is that Amazon is changing to be a source of carbon, but again, this needs to be aligned with the main goal of the work. The title is related to the result of the hydraulic traits diversity being linked to forest resistance, and then this conclusion seems to be secondary in the text. Thus, I believe you need to decide on what to focus on and the main story to tell.

Minor comments – Main Text

In the introduction, lines 44 -46, maybe add a sentence with the link between biodiversity with the diversity of functional traits, and resistance, to explain the sentence about the Amazon. This link will help you to introduce the next paragraphs

where you talk about traits and ecosystem functioning.

Line 81 - Gleason et al 2016 (Weak trade-off between xylem safety and xylem-specific hydraulic efficiency across the world's woody plant species - 10.1111/nph.13646). Found this trade-off can be very weak.

Line 101 – leaf area is not a trait? It is important to make clear what are you calling a trait and what is a phenotype. I think phenotype is the whole combination of traits, and maybe this term is confusing.

Line 149 – remove dramatic?

170-174 – Maybe you don't need to show the equations here, just mention the details in the appendix.

Line 185-186- which niche segregation is you referring to here?

Line 185 – I'm not sure what line of evidence you are talking about.

Lines 210-211 – asynchrony and niche differentiation of what? Maybe better to refer to the trait that was evaluated.

Figure 2 – add what each colour means. 96 species?

Lines 251-252 - Maybe add here that the response was different across sites - further on you say it was different and in some places it reduced.

Lines 257 – which scenario is this?

258 – Which scenario do you mean? Maybe point to the letter of Figure 3. On the right side, there is no increase in biomass. Maybe on the left side, but with increased CO2 you can see more increase in biomass.

Also, you added productive and conservative here and did not mention it before. Better keep with the same terms.

Lines 264-265 – is this right? Seem contradictory.

Lines 279 – add the peak first and after say it is decreasing.

Lines 287 – only sites used in - which sites? The two sites or the whole Amazon?

Lines 313-314 – this sentence is confusing

Lines 316-317 – Isn't this assumed? The strategies are linked with a group of functional traits, if you remove the strategies, you remove that group of traits too.

Lines 325 - 326 – the highest when the drought was the most intense? (the worst drought scenario)?

Lines 329 – 334 - A very long sentence and it is confusing.

Lines 335 – How did you isolate the functional diversity effect? Don't need to say but add a reference to the methods section.

Supplementary - Methods

Lines 118 – 119 - On line 118 you say 96 times, and then on line 119 you 96 species. Is it 96 simulations and 96 species? 96 species each one of the 96 times.

Lines 128 and 129 - You mention the most abundant species in the simulations. Was this after the model ran? Were the initial abundances equal? Maybe add info about the initial species dominance/abundance.

Lines 153 – 155 – does this refer to the Figure 2? I got confused.

Lines 163-165. It is confusing. Is it four or two clusters? Maybe remove the water and light trigger, plus the phenology to check if the groups still will be the same. Looks like a cycling thing. It is also confusing whether you use 3 or 4 clusters.

Was the section on line 310 done for all steps (CAX and Tapajos and the whole of South America)? If so, maybe on methods move it up and say it was done for both.

The simulations of experimental diversity were also done for both spatial analyses. Maybe do the same with the methods sections that apply to both spatial scales.

Another point, you mention the whole analysis was for South America and you show the results in graphs. But you always just refer to this part as Amazonian. Other forests in South America are not in the Amazonian region. So maybe use the term of the biome tropical rainforests or just analyse the Amazonian region.

(Remarks on code availability)

I'm sorry but I do not work with models, so I don't think I'm the best person to review the code.

Reviewer #2

(Remarks to the Author)

The authors have done an enormous job revising the manuscript. In the first submission, the manuscript was so unclear that I found the study near impossible to assess. In this iteration, I could better follow what the authors have done, albeit still with considerable efforts. My main concern is that the research design appears unsystematic, and not clearly motivated and explained.

The focus of the study appears to be "to evaluate whether diversity can buffer the carbon sequestration functioning of Amazonian forests under warming and drying climate". The study starts with introducing the 'catastrophic' and the 'chronic' drought experiments, but readers will soon find out that the two cases are in fact not comparable. The experiments are for different sites, different time periods, and even different ways of manipulating diversity ("initiation with different number of species" in Fig.1; "mean values of the deepest rooting strategy at the CAX site" in Fig S6-catastrophic; "removal of Functional diversity based on Rao's quadratic entropy" in Fig S14; "removal of ecological strategies" in Fig S15... etc.). As such, the experimental setup appears ad hoc and disorganised. As reader, I wonder why the authors do not simply

systematically simulate combinations of different drought types and diversity reduction types? I.e., keeping everything else constant, and systematically vary combinations of drought type and diversity reduction type? If the authors have a rationale for their seemingly disorganised research design, it's not made clear in the manuscript.

The research design might also appear particularly disorganised, because of the lack of a good overview. I previously asked if the authors could provide an overview of their experimental setup flow to make their manuscript more reader-friendly. Table S2 seems to be the addition that's supposed to provide such an overview, but it still unfortunately still fails to provide enough clarity. For example, the key terms 'catastrophic' and 'chronic' used in the main manuscript do not appear in Table S2 at all (nor anywhere else in the Supplementary material, based on word search). I can also not match the types of diversity reduction experiments between the figures and this table. For example, the experiment for Fig S6 that used "mean values of the deepest rooting strategy at the CAX site" does not seem to appear in Table S2, and "the removal of ecological strategies" used in Fig S15 is also not listed in Table S2. Either the terminology is inconsistent or the table is incomplete? Anyway, it makes it very hard for the reviewers and readers to get a good grasp of the different combinations of diversity reductions and droughts that formed the basis for the conclusions.

To add to the confusion, it's not clear from the start which research questions the paper is addressing. In the beginning, it seems as if the paper is focused on addressing if and to what extent hydraulic trait diversity reduce biomass loss under catastrophic and chronic droughts. But the Results section also address the role of different types of traits somewhere between presenting the results from the catastrophic and chronic droughts experiments, which adds to the messy impression. I would suggest that the authors (1) add a paragraph towards the end of the introduction, that clearly details the scope and the research questions, and (2) structure of the results section so it mirrors the order of the research questions list, and make sure the research questions are correspondingly addressed in the abstract and conclusions section. To conclude, I think a more systematic approach, perhaps even with fewer experiments, might have delivered more convincing insights. 'Less is more' could apply in this case. However, while I find a much more systematic and structured approach to experimental setup to be necessary, I can not completely rule out the possibility that it's still a matter of clarity in the presentation. But in the current form, I can not recommend publication for the interdisciplinary readership of Nature Communications. I wish I could be more positive, as the topic investigated is both novel and important.

Some specific comments:

Abstract: Where does the 34 % come from? Based on RCP 8.5, 50% precipitation reduction? Can the text in the Results section be more explicit about how the number is calculated? A lot of context is needed to interpret this number, I'm not sure putting it in the abstract outside that context is a good idea, as it might lead to more misunderstanding than clarity.

L49-55: The authors suggest that the lack of hydraulic traits representation in models as a main reason for the underestimation of diebacks in model simulations, but ref. 8 does not actually say much about possible reasons for low dieback rates in models at all, and various other reasons are given the other refs? Perhaps consider reformulating the sentence(s)?

L217: Not sure how Fig. S7 supports the statement. Wrong cross-ref?

L374: "conservation of functional trait diversity" – what does that mean in a practical sense? Elaborate?

Conclusions/Discussion section: Discuss limitations and future research directions more?

Fig. 3: explain the unit 'tonnes per hexagon'.

Fig. 4: are the precipitation reductions occurring on top of precipitation reductions predicted for the different RCP scenarios 1990-2100? (L341 says 2500 mm/year is mean for the area – but for which period of time?)

Input data files (Dropbox link) returned error message (files not available).

Supplementary Material – Cross-refs only refer to heading names, which is fine in a digital format using the search function, but hard to navigate in a printed format. Maybe consider adding heading numbers and/or page number?

Fig. S6: "set to the mean values of the deepest rooting strategy at the CAX site (egwt, Fig. S6)." Does it refer to Fig. S5? Anyway, not entirely clear what is meant by "the mean values of the deepest rooting strategy". Do I understand it right that S5 A and B show the distribution 'before' removing root niche differentiation. For clarity, would it be possible to simply make a sub-plot C of the traits after removing root niche differentiation?

Fig. S15: The term 'ecological strategies' is not explained in the caption nor in the main manuscript. The term is used only once more in the caption of Fig. 2 (based on word search in the document), but does not provide an explanation for what was done in Fig S15. Elsewhere, the term 'plant strategy' is also used. For clarity, maybe stick to one of the two terms if they refer to the same thing (and make sure it is clearly defined).

(Remarks on code availability)

Dropbox link to input data files returned error message (files not available).

Reviewer #4

(Remarks to the Author)

General comments:

1. Coming into the middle of the review process, I can say the revised manuscript reads clearly. I am not experiencing the confusion that the original reviewers encountered. I feel the issues that led Reviewer 2 to suggest rejection have been addressed in the thorough revision of this manuscript.
2. The original reviewers also made requests to address issues of model validation and skill. These are crucial, but it is also apparent that such opportunities are limited. The authors' reference to their previous work (ref 26), which mainly included tropical savannah, are appreciated. But this appears to be next-generation ecological modeling work. So, the additional careful documentation of the model structure and theory, governing equations, experimental set-up and permutations explored, all are vital and appreciated for increasing confidence in the model and conclusions. After all, the standard for all peer-reviewed scientific publications is reproducibility, and it appears the original manuscript fell well short of providing adequate detail for any other research to duplicate these simulations and analysis. The current version is much better, and the changes make the paper more relevant as well as robust.
3. This is a model sensitivity study. I think if it were clearly presented as such (in the introduction and the conclusions) it would ameliorate some of the criticism and concern regarding the limited validation of the model. Combined with the careful description of how the traits and their selection are grounded in reasonable theory, it would strengthen the position of the paper.

Specific comments (line numbers refer to the mark-up version of the manuscript):

L 114: The text states the model has a fixed probability each year for seed production. I am not familiar with tropical species, but in midlatitudes many tree species have very uneven seed production, with periodic "mast years" of very high production as an enhanced survival strategy. Is mast seeding also a tropical plant trait, and if so, would that additional parameter affect the emergent properties of trait-based adaptation as well?

Fig 1 C-D: Could you also show spread/uncertainty for these curves as well?

Fig 2 and associated text: How do these principal components relate to commonly used (and more widely understood) metrics like WUE and LUE? Such metrics also convolve the species traits you have listed in fairly straightforward ways.

Eq S2: Why did you not use CDF matching to translate anomalies? It is non-parametric, thus more robust than MSE minimization.

Some references to figures in the supplement are misnumbered - both in new text (e.g., L 86) and in the original text (e.g., L 241, 243), which apparently wasn't checked for changes to the figures. Please carefully check that correct figure numbers and panels are referenced throughout the revised manuscript.

Supplement L 325: change "used use" to "used".

Fig S4: Please clarify - is what's being called "stability of transpiration" here and elsewhere actually calculated solely from precipitation (according to the caption)? That would presume no moisture stress, which seems contrary to the premise of the paper being focussed on drought response.

Eq A41: I am not familiar with this root distribution function. Please provide a reference for its origin.

Fig A4: This figure does a particularly nice job of summarizing the phase-space portrayed in Figs 2 and 3 in the main body of the manuscript, and is an aid in interpreting those figures. I suggest mapping into this figure the specific trait abbreviations listed in Figs 2 & 3, and moving this figure into the main text - either as an additional panel of Fig 2, or as Fig 2 itself and then combining current Figs 2 and 3 into one figure. For the broader journal readership, it would be very valuable to have this in the main text.

Reviewer 2 asked about coupled land-atmosphere modeling with such a trait-based model. Inclusion of a trait-based model in a DVM would be the "6th generation" of land models the "5th generation" being those including vegetation demographics (e.g., FATES). This is an exciting prospect - would such a scheme be ready for CMIP7, or will be waiting longer to see coupled climate simulations with trait-based vegetation models? Just asking for your speculation/opinion here, but you could dangle the potential for trait-based modules in Earth system models as a closing remark in the Conclusions.

(Remarks on code availability)

No code posted online. See the authors' code availability statement.

Data for figures are posted online.

Version 2:

Reviewer comments:

Reviewer #1

(Remarks to the Author)

Dear authors,

The manuscript 'Amazon forest resistance to drought is increased by diversity in hydraulic traits' is an important and novel work that can bring new insights in climate change impacts on vegetation. Congratulations for all the changes made in the manuscript. It has significantly improved since the first version. I liked the new analysis, the new figures, and all the re-writing. It now reads a lot easier than previous versions, and it is clear to understand what was done and their results. With all these changes I think I had a much better understanding of the manuscript.

I'm happy with the responses and changes, and although it had a significant improvement, I still have a few last points that the manuscript would benefit before publication.

I would like to state that the points below are from my understanding of the paper, and what I think could be done to give a last improvement to it. I also understand, if for the authors some of these suggestions does not make sense.

Best wishes

Comments/Suggestions

1. I liked the way you exposed the goals and hypothesis, but I think when you write about the two types of droughts, it is important to make the link with the scale of the effect as well (maybe after you write the hypothesis). Something like, to test the different drought effects we had to focus on different scale approaches, where the catastrophic drought was focused on local sites and the chronic drought effect, also related to the changes in climate (long term changes), was tested on the whole forest scale (see Methods SI for details).

Maybe then with this reference you can add an explanation somewhere in the SI why you change the scale when you change the drought type (something like your response 37 to reviewer #2).

I guess this would also address some of the points that the reviewer #2 made, about the changing in methods along the manuscript. It does get confusing as the methods and your approaches changes while you describe your results.

I understand it is because the nature of the investigative study, but it also needs to be clear for the reader.

2. About the questions, I think you are mainly looking at 2 questions, but at different perspectives (drought type).

So, you mainly want to understand:

- How functional diversity mediates biomass in both drought type/scale? (when I say How I think it already includes the magnitude of the effect).

- What are the key functional traits that mediate these responses, in drought type/scale?

I think if you do something like that it is already clear that you will have two sections and that you will investigate the same questions in each drought type/scale analysis.

3. For the results, I don't think it is necessary to be pointing out the hypothesis every time within brackets. I suggest splitting the sections in drought types and remember the reader the scale it was done, but the main points cited in your goals were to test the drought types, and not the scale. From what I understood the scale change was more a method reason.

Example:

Section 1 could be something like: Catastrophic drought simulations, a site scale analysis

Then, you could divide the section in two sub-sections (Biomass change mediated by trait diversity and the Key axes of trait variation mediating plant response).

Section 2 could be something like: Chronic drought and climate change simulations: Amazon analysis.

Then, again, you could divide it in two sub-sections. Here, can you change the order of the results and show figure 4 before figure 3? So, the section about Diversity effect on biomass would come first, and then, you can detail the key traits that have impacted more this change in plant dominance. It will make more sense in the logical order.

4. This manuscript is quite dense of information, and I wonder if there are parts that does not reflect to the main results could go to the Supplementary making the manuscript more objective.

5. Also, I have the impression that the conclusion is very long and has lots of information, looking like more to a discussion. My suggestion is to see what information is new and add to the discussion part/or SI and just leave as conclusion the most important message for the reader.

6. About the Amazon point, why did not the rest of the Amazon was included then? A large-scale study, you can choose by country, continent, or vegetation type and here you mixed all of them and I was just curious why. And the focus is Amazon forest, and it is a huge forest that goes beyond Brazil and Venezuela. So, I guess maybe add some more explanation on SI.

7. On the SI, figures SI 1 and A3, I did not see any information about the allocation in storage. It would be good to add this detail.

(Remarks on code availability)

Reviewer #2

(Remarks to the Author)

The authors have done a great job revising the manuscript! I only have a few comments.

The term 'functional diversity' is central to this paper, but there seems to be different ways of calculating it and it's not always clear which calculation the authors refer to when the term is used.

- "Functional diversity" and "Functional trait diversity" – Please use consistently or add at first use that they have the same meaning.

- Information about what goes into Functional diversity is quite essential, and should probably be in the main manuscript. E.g., consider mentioning the eight traits from L1 183 in the Supplementary in the main (text and/or relevant figure caption).

- So, how many different ways of calculating functional diversity are there in this paper? At L1314 Suppl.: "Functional diversity was calculated as Rao's...": Confusing. So is the 'functional diversity' here calculated from a different set of traits than the eight traits listed at L1 183? At L1366 Suppl.: "Two hydraulically associated traits were used to calculate functional diversity...". If the term 'functional diversity' is used to refer to different calculations, it should be made more clear, for example by use of subscripts (such as $FD_{8traits}$, $FD_{hydraulic}$ etc).

Also, the term 'hydraulic trait' is not clearly defined anywhere in the main text. The 'water-related traits' in abstract does not really explain it. If hydraulic traits simply refer to the two traits "plants maximum rooting depth, and the xylem water potential at which 50% of conductance is lost", it should be stated clearly early in the main manuscript, given how central the term is for this paper.

Abstract "We show that higher functional trait diversity reduces site-scale biomass loss during sudden catastrophic drought by 17-32% and continental-scale biomass loss due to chronic climate change-associated precipitation reductions by 10-34%." : I am of the view that numbers given in an abstract should be possible to understand roughly without having read the entire paper. In this case, the way the numbers given may be interpreted in many ways, for example, a straightforward way to interpret these numbers would be to think that they refer to an uncertainty range relevant for real-world droughts. This is, however, not at all the case. Catastrophic and chronic droughts are also terms that leave it up to the imagination of the reader. Without a better understanding of the context and background, these numbers are quite meaningless, and may spread in news and social media to cause more confusion than clarity. I would suggest either replacing the quantitative reporting with a qualitative one (for example your interesting finding that FD effect on ecosystem resistance is stronger under more severe drought conditions), or that you also provide the necessary background (e.g., ...by 34 % under severe chronic drought (i.e., 50 % precipitation reduction over X years)...).

You may also want to explain how the 30-50% precipitation experiments compare to historical observations or projections of precipitation levels for the Amazon, for helping the reader understand how comparable these theoretical experiments are to the real-world, having in mind the broad readership of Nature Communications.

Other than that, only minor comments:

- Fig. 1: Consider making the cross a bit more prominent (e.g. add legend). Caption says "Horizontal bars indicate the start and end of simulated and real-world drought treatments." – does this refer to the same horizontal bar annotated as "Simulated drought" in the A and B subplots? If so, consider revising the annotation to e.g. simply "drought treatment" or "simulated and real-world drought" for clarity. Also, is functional diversity only calculated for the model results, i.e., in C, real-world functional diversity can not be shown, is that right?

- L299 'results not shown': Better to add support of claims to the Supplementary? Actually, Nature Communications actively discourage this type of statements (see <https://www.nature.com/ncomms/submit/how-to-submit>): "We request that authors avoid "data not shown" statements and instead include data necessary to evaluate the claims of the paper as Supplementary Information."

- Fig. S1: part of the text annotations are up-and-down, better to flip them the right way to make it easier to read.

- Supplementary. Consider adding brief explanations and motivations for the choice of measures and methods (e.g., RaoQ) along with relevant references. Even if the choice may seem obvious to you, it might not be for readers, especially if you target readers also outside the immediate ecologist community.

(Remarks on code availability)

Reviewer #4

(Remarks to the Author)

General comments:

The new organization, with clearly stated research questions and the results section structured primarily around the questions (one section and principal figure per question), is a vast improvement in terms of clarity. The new structure of the text, showing clearly how each question has been addressed, is easy to read and follow. This distillation of the work ("less is more") greatly clarifies the results and makes the paper stronger. When one has done so much work, it is difficult to leave so much "on the cutting room floor", so to speak, but it is necessary to convey your story clearly. It also makes the case much more strongly that this model platform is useful for such explorations of trait diversity.

The care now taken to describe and justify the steps of the research and the decisions made, in the main paper, supplement and appendix, is also a great improvement, making the study and conclusions more sound.

I suggest only minor revisions are necessary before the paper is published (see specific comments below). I do not need to see the revised manuscript again.

Specific comments:

L 67: Start a new paragraph with "We pursue..."

L136, 140, 187, 369, 376, 412, 429-445, 590-644, perhaps more: Please use present or present-perfect tense for your own work and results, not past tense. Throughout the paper, present and past tense is mixed when describing your work and results, which is quite jarring.

L161: Use the model name "aDGVM2" instead of "the model", as this is the first instance in this section.

L222: The strategies do not "avoid" the drought – the drought occurs regardless. Better to say that the strategies moderate or reduce the consequences of drought.

L226-227: I believe the terms in parentheses are reversed from what they should be.

Figure 2 and associated text: It is now clear that there are only 2 species in the Deciduous-Light category, and they arise only at one site. Should they even be included in the analysis? It seems no significant conclusions can be drawn from such a small sample regarding where they sit in PC-space - this point should at least be mentioned. On the other hand, we see clear separation in PC-space for the Deciduous-Water category.

L404-406: This statement is somewhat incomplete, and thus inaccurate. PC1 and PC2 have now exchanged places – this should be stated clearly. However, these two axes remain the two main sources of variance, explaining 49-60% depending on the location and scale. That can be emphasized as well - currently there seems to be attention given only to the hydraulic trait axis.

Figure 3: The explained variance stated on each axis is identical (to the 3 significant digits shown) for each of the four cases. This seems improbable – is there perhaps an error in the plotting script? There are clear (albeit usually small) differences in the PDFs.

Figure S1: I still find this (an even more) useful conceptual diagram for the model framework – it is a shame it cannot fit into the main paper.

Figure S4: For clarity and to help make the author's point, could you add the $y=0$ line to the plot, and perhaps also a vertical line showing the x -value (they are very close) where the curves cross this line? This is a key threshold in the study.

Figure S6: The abbreviations for the curves are not intuitive compared to the terminology used elsewhere. I presume "Red" means "reduction" but in the caption and elsewhere, you say diversity ("Div") is "removed". Please choose consistent labeling.

Figure S12: Likewise, the curve labels are quite cryptic. Please expand as in the Fig. S11 caption, or map them clearly to what is shown in Fig. S11.

Figure A2: The figure is not "below" – the first five words can be deleted.

(Remarks on code availability)

Dear Reviewers,

We greatly appreciate the effort taken to review our manuscript and provide such detailed and helpful commentary. Below please find a summary of the main changes and detailed responses to each of your comments (blue text).

The main changes include:

- As per editorial and reviewer requests, the entire manuscript has been extended and edited for clarity. Substantial additional text has been added to the main text at positions suggested by the reviewers. Much of this additional text is used to introduce the reader to the vegetation model used. However, this additional text may be better suited to methods or supplementary sections. We have chosen to leave this text in positions suggested by the reviewers and welcome editorial and reviewer guidance on the best positioning of these additional descriptions.
- An overview table with all model experiments in the supplement (Tab. S2).
- A full model description from an earlier paper has been added as an appendix.
- The order of residual soil water content (rc) values in Tab. A1 has been changed to correct a mis-specification. All model simulations have been re-run with an updated model version of aDGVM2. Reviewer three expressed doubt as to whether our initial results were coincidence (comment 96). Our re-run of all simulation experiments with the corrected soil water content values clearly demonstrates the reproducibility of our results.
- We have updated the literature list in order to consider the most recent related studies and provide additional context to our findings.
- We have taken advantage of suggestions by reviewer two to conduct additional analyses which allow better comparison to be drawn between our future change scenario predictions and published literature (Figs. S10 and S11).
- We have conducted an additional simulation experiment to identify a mechanism which can lead to simulated biodiversity effects as per a request by reviewer three. These simulations reveal that the amount of biomass lost during severe drought is mediated by below-ground rooting niche differentiation (Figs. S5 and S6).
- Source data can be downloaded using the following link: <https://figshare.com/s/fa75add00e8445b6fc6a>. A link is also provided in the main text.
- Source code has been updated and uploaded.

REVIEWER COMMENTS

Reviewer #1 (Remarks to the Author):

The manuscript "Hydraulic-trait diversity increases forest resistance to water deficits" brings the discussion to a very important subject that is associated to the forests response to predicted climate changes, focusing on biomass impacts. Moreover, it includes an aspect of functional diversity on forest response to drought.

The biomass loss over time, in addition to biodiversity loss, involves CO₂ release in a world where everyone is trying to sequester CO₂. So this results point to an area of global interest, and brings a recent approach of vegetation dynamic models. This seems to be very promising to improve vegetation response to climate change and could be replicated to predict other vegetation response across many tropical ecosystems.

Thus, I believe the manuscript would have a good impact in advancing studies of vegetation modelling in a changing world and has the potential to be published in Nature Communication. However, it needs to improve in many aspects, as commented below, including some text organizing and more clarity on methods and results, to make it more readable and understandable for the broader public of this journal.

RESPONSE: We thank the reviewer for these encouraging introductory lines.

Major comments (I listed the topics below but they are better explained together with the minor comments along the manuscript sections):

1. Text clarity. There are many parts in the text that are not very clear enough, and maybe it would need more writing to exactly explain what the authors mean. Sometimes it looks like is very clear to the authors what they are talking about but for who is reading it is not.

Thank you for your constructive comments. The text has been comprehensively revised to improve clarity of writing, methods, and results. We detail these revisions below.

2. Methods. The methods need improvement with more details about what was going on in each step of this paper (i.e. traits involved, how the trait vs. species association, climatic changes scenario)

The methods have been improved as follows:

- 1) A table (Tab. S2) has been added. The table gives an overview of all simulation experiments, the forcing data used, manipulations of forcing data, the time period of any manipulations, and trait-diversity setup.
- 2) A full model description has been added as an appendix. The main text has been revised to provide relevant references to sections and equations in the appendix in order to assist the reader through what is admittedly a complicated model.
- 3) The descriptions of climate forcing and climate change data used has been significantly revised and extended in the supplement.
- 4) To provide a more transparent overview of climate data used to force the model we have included a new figure (Fig. S2). This figure shows climate forcing, climate change anomalies, and CO2 treatments.

3. The results for the Amazon. Needs some checking if it was only data for Amazon or included data from other biomes in Latin America.

The previously presented data included tropical forest and savanna sites. For clarity, savanna sites have now been removed entirely.

4. I think the discussion needs more biological association with hydraulic strategies, not only too descriptive.

Thank you for your comment. In response to this comment and comment 71 by reviewer 2 we have significantly extended the current text and added many comparisons between our results and existing research on hydraulic traits and diversity. We have added references to recent research on hydraulic strategies (Oliveira et al., 2021) to point the reader to the very strong parallels between the strategies emerging from the model and the “state of the art” thinking on hydraulic strategies.

Additionally, in response to this comment and your comment no. 9 we have:

- 1) Clarified the importance of hydraulic traits with respect to relevant research in the summary paragraph.
- 2) Expanded the introduction of the main text (lines 69-78)(with appropriate links to a full model description)
- 3) Improved the discussion by including additional literature to frame and compare our simulation results with empirical evidence. We use these additional linkages to draw parallels between our simulation results and evidence in tropical forests e.g.:

Lines 180 - 189: “Experimental support for diversity-stability relationships in tropical or subtropical forests is rare. An exception,³⁸ was designed to address this shortcoming. Within this project, mechanistic linkages between diversity in P₅₀, a key trait composing the hydraulic-strategy axis above (Fig. 2), and stability in productivity were identified.⁹

The authors suggest that other hydraulically important traits, such as those associated with leaf phenology, below ground traits, and below ground niche differentiation in rooting depth may additionally play a role in mediating stability.⁹ The alignment of these additional traits along the hydraulic-strategy axis (Fig. 2) supports this suggestion and highlights that variability in one of these traits (i.e. diversity) in the model, and empirically,^{7, 39, 40} can be linked to variability in the others”

- 4) Conducted additional analysis to link simulated relationships with observed linkages between transpiration fluxes and diversity observed at BEF-China (Fig. S4). We additionally identified below-ground rooting niche differentiation as one mechanism leading to higher drought resistance in diverse communities (Fig. S5, S6), a mechanism proposed by Schnabel et al. (2021) as part of BEF-China, and many other authors. Taken together the additional discussion, analysis, and experiment, situate our simulation results solidly within observed and theorised mechanisms.
- 5) Lines 274-286 contextualise our future change biomass results by improving comparisons to empirical evidence.
- 6) We have also improved the description of linkages between the results presented in the new Fig. 4 and existing research.

Minor comments:

Main text

Summary paragraph – It is a bit confusing, it needs some rewriting.

1. Thank you for your helpful comment. We agree. The summary paragraph has been rewritten with the following goals:

- a) to improve clarity
- b) to immediately introduce the reader to specific terminology used throughout the manuscript, i.e. our definition of hydraulic traits with appropriate literature reference (lines 10-13). We have chosen to group leaf phenological and root traits under the term “hydraulic traits” following the conceptual structure presented by Oliviera et al. These authors argue that all of these traits covary (phenological, root, and classically hydraulic traits) and together mediate plant water status. They refer to diversity in all of these traits together as “hydraulic diversity”.
- c) as per your comments below we have added more references to literature related to forests, drought, species diversity, and plant traits. We chose to add these additional references in the summary paragraph to improve the immediate framing of the central focus of the manuscript.

Oliveira, R. S. et al. Linking plant hydraulics and the fast–slow continuum to understand resilience to drought in tropical ecosystems. *New Phytologist* 230, 904–923(2021).

Lines 11-12. The authors use the term “Biomass resistance”, maybe better change for the forest resistance to biomass loss up to 25%.

2. Thank you. We have changed line 21 to: “...we show that higher diversity in hydraulic traits reduces biomass loss during sudden catastrophic drought and chronic climate change-associated precipitation reductions by up to 34%.”

Here the author means the less diverse forests lost 25% more biomass, isn't it?

3. Now it means less diverse forests lost 25% (34%) more biomass. It is the difference in percentage reduction between high and low diversity, i.e. high diversity communities incur a 20% biomass reduction and low diversity communities incur a 54% biomass reduction. The difference between the two is then 34%.

Previously this calculation was different and was not conveyed clearly enough. We have updated Fig. 4 and associated analyses to improve clarity; we additionally explicitly calculate functional trait diversity at the continental scale in each simulated treatment, this allowed us to quantify the effect size of functional trait diversity on biomass responses. The differences in responses owing to the level of functional diversity are now hopefully both clear and more intuitive.

In many parts of the text is not very clear the way the authors present the results. They do not say exactly what was observed, using other terms.

4. The text has now been considerably revised for clarity. We have made efforts to rephrase our findings in clear and simple terminology and have reworked some of the analyses (Fig. 4) to aid clarity.

Line 16 – The authors call the removing phenological groups (triggered by water) as hydraulic traits. The deciduousness is a strategy to avoid water loss, but may use the term of phenology strategy instead of hydraulic trait. When they say hydraulic traits, the readers would expect other traits as well. I think they should point out exactly the trait they removed (i.e. the water associated phenology strategies).

5. We agree that the grouping of phenological and hydraulic traits could be confusing but would prefer to leave the terminology as is. We have chosen to

group leaf phenological and root traits under the term “hydraulic traits” following the conceptual structure presented by Oliviera et al. (2021). These authors argue that all of these traits covary (phenological, root, and classically hydraulic traits) and together mediate plant water status. They refer to diversity in all of these traits together as “hydraulic diversity”. We refer to these groupings of traits which covary in consistent ways as hydraulic strategies and diversity in these traits as hydraulic trait diversity. Reference to hydraulic traits and trait diversity in this way is also in keeping with the Xu et al. (2016) reference you provided. Here, the authors embedded hydraulic architecture and traits within leaf phenological modules again highlighting that leaf phenological and hydraulic traits group together and form strategies. They refer to the diversity in the combination of phenological and hydraulic traits in the title of their manuscript as hydraulic trait diversity. At the very beginning of the manuscript (line 11) we clarify our definition of hydraulic traits to be traits which affect plant water status. Between lines 74-86 we add additional details of traits involved in mediating plant water status and provide links to the model description which provides full details of plant hydraulic and phenological modules. We hope that together these changes will improve clarity.

The sentence in question has also been changed to:

“Further, by manipulating functional diversity, we show that more biomass is lost in low diversity climate change simulations, thus clearly illustrating that diversity in hydraulic traits buffers the impacts of water balance changes.”

In the main text we follow your recommendation to be specific about what we have removed. On line 299-302 we now specify that we remove deciduous and evergreen strategies. The above sentence (initial line 16) is now rewritten in functional diversity terms as subsequent analyses now use calculations of functional diversity for the study area (as per reviewer 2’s request) for the full and reduced diversity simulations. This has the benefit of being able to assess the initial distribution of functional trait diversity across the study area (Fig. S14 A), assess whether removing strategies affected functional diversity (Fig. S14 B, C), and isolate the effect of functional trait diversity statistically (Fig. 4, Tab. S3).

Lines 29-32 – I would use more forests studies as reference here instead of grassland based ones. Although the ecological aspects can be the same their response and time of response may be very different. Some forests references could be:

Sakschewski, B., Bloh, W., Boit, A., Poorter, L., Peña-Claros, M., Heinke, J., Joshi, J. & Thonicke, K. (2016) Resilience of Amazon forests emerges from plant trait diversity. *Nature Climate Change*, 6, 1032-1036.

Xu, X., Medvigy, D., Powers, J.S., Becknell, J.M. and Guan, K. (2016), Diversity in plant hydraulic traits explains seasonal and inter-annual variations of vegetation dynamics in seasonally dry tropical forests. *New Phytol*, 212: 80-95. <https://doi.org/10.1111/nph.14009>

6. We fully agree and thank you. We have added a reference to Sackschewski here. We have added a reference to Xu into the revised summary paragraph (line 13). As per reviewer two's request, we have added further references to relevant studies which focus on forests, diversity, traits, or drought, i.e. Aguirre et al. 2022, Oliveira et al. 2021, Schnabel et al. 2021, Anderegg et al. (2018).

The Sakschewski et al (2016) reference has been added to line 44.

Line 51 has been changed to include the Xu et al (2016) reference and now reads:

“A problem with this previous work is that the models used did not consider how hydraulic traits can mediate plant water status and how diversity in these traits may influence how forests respond to drought.⁸ Therefore, these analyses could not evaluate whether hypotheses from theoretical and experimental ecology, that diversity increases the resistance of ecosystems, apply to hyper-diverse tropical forests.”

Lines 36-39. Long and confusing sentence. Needs some rewriting or punctuation.

7. Hopefully the changes made to lines 36-39 above suffice to improve clarity.

Lines 40 – 45. Needs some rewriting. (Lines 40-41. What diversity are you talking about: functional diversity or diversity of species? The may be correlated but it is good to keep the same term along the text. Also, what major barriers do the authors refer to? Lines 42-43. Symptomatic of the problem - What the authors mean with this.)

8. Thank you for highlighting the unclear text. We mean species, phylogenetic, and functional trait diversity but as the model we applied has been designed to represent functional trait diversity, we now only refer to functional trait diversity here for clarity. We have specified the two major barriers we tackle in the manuscript. “Symptomatic of the problem” has been removed and the text in this paragraph has been substantially revised and now reads:

“Predicting the effects of functional trait diversity on the functioning of ecosystems has been deemed the 'Holy Grail' of trait-based ecology.²⁶ However, the related challenges of vegetation models' failure to reproduce observed biomass decreases in tropical forest drought experiments^{19, 20, 22, 23} and the lack of

plant traits within these models which mediate plant hydraulic performance,⁸ impair our ability to predict whether diversity in traits mediates tropical forest responses to long-term and sudden reductions in water availability.¹⁰ We pursue two avenues which simultaneously address these barriers. First, we include in our model, advances in disciplinary understanding of plant hydraulics (see “Plant Hydraulic Strategies” in Appendix).^{24,25} Typically in DVMs, plant water stress is governed by factors external to a plant such as precipitation and soil moisture content.⁸ The inclusion of plant hydraulics in aDGMV2 allowed that, in addition to external factors, plant hydraulic performance is governed by properties intrinsic to a plant, i.e. root, stem, and leaf traits and phenotypes. Thus, this implementation allows the action of natural selection across an individual's root, stem, and leaf traits which simultaneously affect hydraulic performance through both abiotic and biotic components of the soil-plant-atmosphere continuum (Fig. A2). Trade-offs between maximum xylem and leaf conductivity and cavitation²⁵ allow alternative strategies to emerge in the model. Phenological traits, i.e. whether an individual is evergreen or deciduous, determine whether individuals endure or avoid water shortages. Below-ground traits determine the amount of carbon allocated to roots, root shape, and plant rooting depth. These traits affect water availability of individuals in the model and thereby competition for water (Fig. A2). Second, by creating an eco-evolutionary framework through which trait diversity emerges, we can consider how functional trait diversity impacts the response of ecosystems to sudden short-term catastrophic drought and chronic long-term climate change-associated precipitation reductions.“

Lines 47. Although I now the connection, the hydraulic traits are added without any link of it with plant response to drought. Maybe needs a sentence linking the different hydraulic strategies and how they can deal in different ways with drought. As this will be one of the main topics of the work needs more discussion about it. The first 3 three paragraphs can be summarised to include more details about it.

9. The paragraph which ended at line 47 has been extensively rewritten to explain the linkages between drought responses, hydraulic traits, diversity. A full model description has now been included and reference to relevant sections has been made in relation to hydraulic strategies, leaf phenology, and differential responses to water shortages. See new paragraph text in previous comment.

Lines 52-54 it is not clear which parameters are considered genotype and phenotype. When you say parameters that code for how individuals respond to environment I did not get it before reading all the paper. This should be linked

better to the methods explanation and better explained here, as it needs to be clear enough in the main text at least for a brief understanding.

10. The paragraph which included lines 52-54 has been re-written to improve linkages with the additional text in the preceding paragraph, to improve linkages with the methods section, and to clarify the intended meaning of genotype and phenotype within this context. Referring to an individual's suite of traits as a genotype may not aid understanding, we have replaced “genotype” with “suite of traits” or “traits” and added a full model description which includes a full list of traits (Tab. A3). The text now reads as follows:

“We use a dynamic vegetation model (DVM) with an improved representation of plant hydraulics to achieve this. The model can accommodate high levels of functional trait diversity, allowing us to evaluate whether diversity can buffer the carbon sequestration functioning of Amazonian forests under warming and drying climates. The model we use (aDGVM2,²⁷ Fig. S1, Supplement, Appendix) combines standard DVM methodology with eco-evolutionary processes. It is individual- and trait-based. In the model, individuals have traits (see Tab. A3 for a full list of adaptive traits), which influence their performance in competing for light and water. The traits of an individual have particular values within predefined ranges (Tab. A3). The benefits and costs of an individual's traits influence its phenotype (e.g. stem biomass, leaf area, height, number of seeds produced) (outer ring Fig. S1) in interaction with the specific abiotic (climate and soils) and biotic (competition between individuals) context. Feedbacks between the traits, phenotype, and changing abiotic and biotic conditions determine an individual's relative fitness through time via reproductive success.²⁸ Differential relative fitnesses through time drives the processes of community assembly²⁹ and adaptive trait evolution.³⁰”

Here we provide a summary of how community assembly and adaptive trait evolution are simulated by the model; the appendices provide more detail. Parent individuals belonging to a particular species (species is a generic label applied to individuals at initialisation to allow reproductive isolation between groups²⁶) exchange trait values with a fixed probability to produce seeds (see “Reproduction, inheritance, mutation, and cross-over”, Fig. A5 in Appendix). The trait values of each seed are further modified by mutation with a fixed probability. Each year during recruitment, seeds are randomly drawn from the seed-bank (see “Plant recruitment” in Appendix). These recruits can then grow into saplings and small trees. The number of seeds an individual produces is determined by the amount of carbon it assimilated, the proportion of carbon allocated to reproduction (see “Carbon allocation” and Fig. A4 in Appendix), and seed weight. The proportion of carbon allocated to reproduction and seed weight are adaptive traits (see Tab. A3 and “Plant recruitment” in Appendix). Individuals producing more seeds have a higher probability their seeds will be drawn from the seed-bank while heavier seeds have a higher probability of

germination and establishment (Eq. A77). Thus, trait inheritance, cross-over, and mutation allow new, potentially novel trait combinations to emerge via seeds. This combination of abiotic and biotic (competition) filtering and evolution allows functional trait diversity to emerge in modelled vegetation stands and thereby provides new opportunities for exploring how functional trait diversity impacts ecosystem services (inner ring Fig. S1) such as carbon sequestration.”

At the end, I think all the traits are selected based on the phenotype expressed because it will ‘determine’ their response to the environmental changes simulations. So, even the phenology, if it is been triggered by water or light, will also reflect on some physiological and molecular changes to receive signals from the environment. So you add this information in the model with will be associated to a genetic definition, but at the end, the response will be determined by the phenotype. All this needs to be more clear in the manuscript.

11. Apologies, explaining the details of how the model works in a concise fashion is difficult. Therefore, we now include a full model description as an appendix for more details, which is a common procedure for rather complex process-based models. However, we very much appreciate the reviewer comments, which helped us to better explain how the model works and which processes are exactly covered and how. The text alterations in response to this and your previous comment will hopefully clarify the interactions between traits, the phenotype, and the abiotic and biotic conditions. All references to a “genotype” have been removed and replaced by “traits”/”trait-suite”.

It is also important to make it clear that although the model is based on individual, the selection is made at species level, right? (see some other comments about this part of the methods below).

12. Apologies for the confusion. The edited text in response to your previous comment about the paragraph, which included lines 52-54, will hopefully clarify the level at which selection acts. We have added the sentence “This allows the action of natural selection across an individuals root, stem, and leaf traits which simultaneously affect hydraulic performance through the soil-plant-atmosphere continuum (Fig. A2).” to indicate that selection happens at the level of the individual, as influenced by interactions between the values of traits an individual possesses, the abiotic, and biotic conditions. We have also included text to make it clearer that relative fitness through time is driven at the level of the individual via differentials in mortality and birth (see “seed production and germination” in Appendix).

Line 55 – need some explanation of this term that you are using: Genotype-traits. May say for now on we will use the term Genotype-traits to refer to a phenological trait (deciduous or evergreen), which is triggered by water or light.

13. This confusion has hopefully been clarified and the term “genotype-traits” has been removed.

Line 62-63 – Was the distribution of the biomes based on the biomass estimates? Or phenotypes? This sentence is a bit without connection here. Maybe introduce more the model and how it can be used to estimate the vegetation response to some catastrophic events and then link this with the first analysis of this work, which was to explore the drought experiments in model and compare the ground and modelling data.

14. The biome classification scheme applied by Langan et al. (2017) used a combination of tree basal area and grass biomass to classify simulated vegetation patches into either savanna or forest. Given the main focus of the manuscript is on tropical forests and to avoid confusion, all savanna areas have been removed from the results. We used data on the distribution of tropical savannas provided by Lehmann et al. (2011) to remove areas designated as savanna from our simulations.

Lehmann, C. E. R., Archibald, S. A., Hoffmann, W. A. & Bond, W. J. Deciphering the distribution of the savanna biome. *New Phytologist* 191, 197–209 (2011).

Line 65 – level and rate? Does level refers to the absolute biomass loss? Maybe be more specific on what you mean.

15. Thank you, level and rate have been replaced in the text to improve clarity as follows: “Simulations showed that the percentage of biomass lost and the timing of these losses were reproduced at both sites (Fig. 1 A, B)”. Figure 1 A, B has also been altered to show the period during which field and simulated drought treatments were applied. The year in which field measurements were carried out to assess the effect of the drought in these forests is now also indicated in Fig. 1.

Line 66-68 – the authors use the word diversity in many parts but it would be more clear if the refer to what diversity they are talking about. Traits diversity or diversity of species? At the end it would be the same because I think the species diversity here are linked to functional/trait diversity, or not? Maybe it would be good to have a sentence saying this at the beginning.

16. We now always make clear which aspect of diversity we refer to in the text, mainly functional trait diversity. The trait diversity is also linked to species diversity, but high species diversity does not necessarily lead to high trait diversity (in reality and in the model). The species concept here is novel and not captured by similar trait-based DGVMs, such as LPJmL-Fit (Sakschewski et al. 2016). For aDGVM2, the species concept is fundamental as trait values are heritable and the addition of species (reproductively isolated groups of individuals) led to the emergence of multiple strategies rather than convergence toward a single solution (Scheiter et al. 2013). We have added additional details on what is meant by species in the model (line 102) as well as reference to the relevant section of a full model description (Appendix) which details the implementation.

The species diversity vs functional diversity query is hopefully adequately addressed in response (18) to your comment about line 70.

Line 67 – Maybe list here what traits were considered in this first analysis.

17. A table containing all traits which can evolve has been added to the appendix (Tab. A3). All of these traits were used in the first experiment as every trait required a trait value at initialisation. The table is now referred to in the text as follows: “We then investigated whether functional diversity affects biomass responses to drought by conducting simulations initialised with different numbers of functionally unique species (see Appendix and Tab. A3 for a list of traits and their potential ranges)”. This table is also now referred to in the section “aDGVM2 constrained diversity simulations” of supplement S1.

Line 70 – Isn't the species associated with traits diversity? Isn't this a model premise?

18. The relationship between species number and functional diversity is not predefined but rather an emergent property of model dynamics. In the model, the number of species is not necessarily associated with trait diversity. Communities could assemble or be created which contained a high number of species, and these species could theoretically all have identical trait values and, as such, exhibit zero functional diversity. Niche theory is commonly invoked to explain the diversity of forests (Falster et al, 2017), whereby niche differentiation allows the coexistence of a diversity of strategies. Indeed, Falster et al. (2017) showed that trait trade-offs allow niche differentiation and the coexistence of higher diversity of strategies. A novel feature of the emergent dynamics displayed between Figs. 1 C, D, and 2 is that, in higher species richness communities, functional diversity is also higher. This result implies that diversity of strategy, as defined by trait differences, allows a higher number of species to coexist. The

analysis conducted and presented in Fig. 2 suggests that two important axes of differentiation emerge, one linked to allocation trade-offs and one linked to hydraulic and phenological traits which mediate plant water relations.

As such, our results show that the triple conundrum of species coexistence, niche differentiation across trade-off axes and functional diversity, and the relationships between functional and species diversity and forest resistance are strongly related. We fully recognise the significance of these results with respect to the drivers of tropical forest diversity and the coexistence of strategies. However, we feel that, given the current biodiversity crisis, the most important focus of the present manuscript should lie on the relationships between functional diversity and tropical forest resistance to changes in biomass as affected by water availability and have constructed the manuscript as such.

The reduction of species was also a result from the drought?

19. The differences in the numbers of species were due to the model being initialised with varying numbers of species but also the extinction of species during the simulation spin-up prior to drought treatment being applied. In Fig. 1 D, only the number of species that were present in communities before the simulated drought treatments were applied, were identified. Whether drought resulted in species extinction or a reduction in functional diversity in Fig 1 simulations was not analysed here. This sentence has been rewritten to clarify as follows: “This simulation experiment showed that communities of lower functional diversity contained fewer coexisting species before the application of drought (Fig. 1 D) and exhibited more dramatic biomass reductions following drought (Fig. 1 C). ”.

Figure 1- Which year were the observations result from in both drought experiments. Maybe add more details on the figure 1 subtitle about the species number treatments (18 in total right?) It would be good if the reader did not need to go to the methods to understand it. The Y axis does not need to say reduction as the numbers already point to a negative value. Maybe use Change in Above Ground Biomass.

20. Fig.1 has been altered and now shows the year in which the effect of drought in terms of biomass reductions was measured at each site as well as the period during which real-world and simulated drought treatments were applied. The Y-axis label is now “Change in above ground biomass (%)”. More detail has been added to Fig.1 about the species number treatments. It is also now indicated that both the species number treatment level and extinction during the spin-up phase

can affect the species richness before the drought treatments were applied (Fig.1 D).

Lines 77-79 - When you are talking about root, stem and leaves allocation you are talking about root, stem and leaves allocation in biomass associated to growth, right? Because when you say allocation this can be for growth or for storage, but they you consider storage the other extreme off the trade-off. If you are talking about more acquisitive strategies you may be talking about higher growth investment. This is not very clear in the text.

21. Thank you. The clarity of this text section has been improved as per your suggestion. A sentence has been added that situates this trade-off axis within the context of observed allocation trade-offs (as requested by reviewer three). The text is now as follows:

“Productive vs. conservative strategies differentiated across carbon allocation traits with productive strategies allocating higher amounts of acquired carbon to the immediate growth of roots, stems and leaves. This immediate allocation to grow plant compartments traded off against allocation to storage which a plant can subsequently use to flush leaves when phenological triggers are activated or recover from fire-associated top-kill. Indeed, recent evidence indicates that such trade-offs between the active allocation of acquired carbon to growth vs. to storage represent an important axis of variation within the spectrum of plant life-history strategies³².”

Also, about the allocation to roots, this may not be completely related to acquisitive strategies. I would be careful when assuming that.

22. Yes, indeed, there are many terms used to describe life-history strategies that resemble this modelled axis of variation. To improve clarity and follow terminology which better describes this trade-off axis (Blumstein et al. (2022) (reference 32 in the main text)) we have changed “acquisitive vs safe” to “productive vs conservative” throughout the manuscript.

Lines 80-81 – The authors use the term matric potential for plants, but I guess they mean to say water potential. The matric potential is a component of the water potential. In functional ecology, the P50 is a measure of water potential, when the plant loses 50% of its water transport capacity, so the term water potential would be more appropriate.

23. Yes, thank you. Matric potential has been changed to water potential throughout the manuscript when referring to P50.

Lines 81-83 – Please add references for this. This is not resultant from your analysis right?

24. If you are referring to the trait covariance in the model which results in familiar plant strategies then yes, this is an emergent model property and is a novel result. However, it is not entirely clear what is being referred to here, and I assume this comment refers to the linkages between the water potential at which 50% of conductivity is lost (P50), specific leaf area (SLA), leaf longevity, stem and leaf conductivity, and wood density. In the model, P50 is an adaptive trait which natural selection can act upon. We have coded empirically derived relationships into the model between P50 and SLA, stem and leaf conductivity, and wood density. Leaf longevity is also a function of SLA and is, therefore, also derived from the P50 value an individual has. A full model description has been added as an appendix in order to allow easier clarification of model specifications. Cross-references between main text and relevant components and equations in the model description have been added throughout the text. The above sentence is now:

“These strategies were: (1) drought tolerant strategies exhibiting very negative water potentials at which 50% of conductivity is lost due to cavitation (P₅₀). This adaptive trait influences derived traits (Figs. S1, A2) via coded empirical trade-offs (see “Plant Hydraulic Strategies” in Appendix) and results in low specific leaf area (SLA, Eqns. A47-49), high leaf longevity (Fig. A3, Eq. A72), low stem and leaf conductivity (Eqns. A50, A52), high wood density (Eq. A46), and greater resistance to cavitation (Eqns. A68, A69).”

In lines 80-91 – It is not clear if you include all these data in the model, or you are just mentioning traits that may be correlated to it. Try to make clear of what is your result and what the discussion is about.

25. The text which began “The second axis revealed hydraulic-strategies which closely match observed strategies across Amazonia” has been changed to “The second axis of differentiation showed that selection within the model led to the emergence of whole-plant hydraulic strategies via coordination between multiple plant traits which mediate water relations and phenological traits³⁶ which closely match observed strategies across Amazonia³⁴” in order to make it clear that the trait-coordination and formation of strategies is a result which emerges from the model.

If you are referring to traits derived from the value of P50 we hope that our response to comment 24 clarifies sufficiently.

Lines 96 – Add ‘different strategies’ or ‘higher diversity of strategies’ that influenced the community resistance to make the sentence clearer.

26. Thank you. This comment and the one immediately below are addressed below.

Lines 97-98 – The authors are saying the same thing as before. The resistant communities showed higher-levels of functional diversity.

27. We agree that this is somewhat unnecessarily repetitive. As such, this text and analysis have been removed from the manuscript. Reviewer 3 correctly pushed for the identification of mechanisms. We have replaced the original analysis (old Fig. S3) with new analyses and simulations. We now show that more diverse communities exhibit higher stability in transpiration (Fig. S4). That below ground rooting niches are different for the different strategies present (Fig. S5). We then manipulated below ground rooting niches at both drought sites by setting all below ground traits to the same values immediately before drought. This additional experiment demonstrates mechanism, here we show that the removal of below ground rooting niche differentiation causes strong additional reductions in biomass loss at both sites (Fig. S6). We believe these additional analyses are novel, more strongly complement the focus of the manuscript, and address long standing questions regarding the relationships between diversity and resistance by showing that rooting niche complementarity/differentiation is indeed a mechanism which can lead to the modelled responses we present.

Figure 2 – It would be good to know in which direction the arrows for category Eg and Phenology are going to. It is confusing, maybe use the term phenology as a determinant of evergreen or deciduous strategy and use another term for saying if the trigger is water or light. I think it would make more sense, but it is just a suggestion.

28. Thank you for your comment. I have tried relabelling these traits. However, this did not improve the plot. I have instead changed the description for these traits in the figure legend of Fig. 2 and referred to this description in all PCAs to reduce confusion and address reviewer three's comment regarding the lack of clarity associated with the direction of the arrows.

Is each colour + symbol one different species?

29. Each colour and symbol together represents one of the four clusters to which species have been assigned by the clustering algorithm. Large symbols indicated the cluster centres, these have been removed to aid clarity. This has been clarified in the figure text.

About the sub-section division: Maybe the sections of the text could be specific about what spatial scale the authors are talking about.

30. The subsection divisions have been changed to reflect the relevant spatial scale and to make clear that these are simulation results as per the editor's request. Additionally, a table has been added to the supplementary materials detailing the spatial scale, forcing data, diversity set-up/manipulation, time period and simulation length, forcing variable manipulation, and the type of simulation experiment conducted.

Also the traits are the same as before I imagine? Maybe mention that?

31. The sentence which previously began on line 108 has been edited to make this clear. This now reads: "Simulations were run without imposing constraints on diversity (see Tab. S2 for experimental set-up), and the trait values of all individuals (Tab. A3) were randomly initialised at the start of a simulation. Following initialisation, new individuals inherited trait values which could be modified by mutation and cross-over (see "Reproduction, inheritance, mutation, and cross-over" in Appendix)."

Lines 101 – it would be good to have a line explaining why the use of the different terms catastrophic and chronic and which parameters they have changed. Many things are not very easy to understand what the authors are talking about.

32. The text at line 57 has been changed to improve clarity as this is the first mention of "catastrophic" and "chronic" in the manuscript. "sudden short-term" and "long term" has been added to the text.

The text reads, "Second, by creating an eco-evolutionary framework through which diversity can emerge, we can consider how functional diversity impacts the response of ecosystems to sudden short-term catastrophic drought and chronic long-term climate change-associated precipitation reductions."

Although sudden severe droughts and long-term gradual reductions in precipitation represent changes to the same resource, they are different types of perturbation which are often referred to as pulse and press perturbations. Rather than introduce extra jargon which does not aid general understanding, we chose to use the more generally understood terms "catastrophic" and "chronic" to highlight these differences in perturbation type. To further improve clarity, each instance of "catastrophic" and "chronic" in the manuscript is now written together with its associated precipitation change, i.e. "sudden catastrophic drought" and "chronic precipitation reductions".

I would suggest saying things like; we will refer to this xxxx from now on as xxxxx. Or just keep saying the same thing; even if it seems to be repetitive, it is better than making it not understandable.

33. Thank you for your comment. In response to the previous comment, we also address this comment and now clarify and repeat terminology throughout the manuscript.

Lines 102 - About the scenarios RCP 4.5 and 8.5 I suggest the authors to give a better explanation of what is each one in the methods, and what parameters are they changing in the model.

34. The explanation of changes has now been significantly revised in the supplement, a table detailing parameters which change for each experiment has been added (Tab. S2), and a figure has been added (Fig. S2) to display precipitation, temperature, RCP anomalies, and CO₂ forcing.

Lines 105 – Also about the CO₂ increase, I think you need to add more information about how much it was considered. I did not find any additional information in the methods.

35. The explanation of changes has been significantly revised. A table detailing changes made for each experiment has been added (Tab. S2). A figure (Fig. S2) shows precipitation and temperature input and anomalies (Fig. S2 A-F). Fig. S2 G shows the temporal changes in CO₂ concentration applied for each experiment.

Figure 3 – Again, the authors should indicate the direction of the Eg and Phenology in the subtitles.

36. We have indicated the direction of Eg and Phenology and improved the description in the legends of all PCAs.

Lines 107-111 – Does the results from figure 3 have the same data of figures S3? It is not clear whether the authors included other biomes (non-forested). The maps on Fig S3 show a more broad analysis over the Latin America, with points in other biomes, including non-forested biomes.

37. The original study area included savanna sites. These sites have been removed from the analysis using Lehmann et al. (2011) savanna distribution data. We have done this to: 1) to address your comment. Having multiple biomes (savanna and forest) in this manuscript makes it more cumbersome to clearly explain to the reader what the main results are without doing completely

separate analyses for each biome; 2) to reduce confusion as the main focus of the manuscript is on tropical forests and; 3) the response of savanna biomes to climate change may differ substantially to that of tropical forests. To analyse these dynamics appropriately will require a separate manuscript.

The data from Fig. 3 were for the same area as Fig. S3. This area did include areas of tropical savanna. As such, old Fig. 3, and old Figs. (S4, S5, S6) displayed trait data and changes for both tropical forest and tropical savanna sites. In Langan et al. (2017) the “study area comprises of sites in Brazil and Venezuela north of 23 S which are classified as either forest or savanna” and “exclude sites above 1000 m elevation”. In this revised manuscript, all savanna sites have been removed. This has been made clear in the methods.

For the purpose of this response, we re-plotted the above-listed figures using only data for forests which reveals that the removal of savanna sites does not change our initial predictions of trait-space changes (biomass in tonnes per hexagon, number of individuals, proportion of total biomass). We include a figure below which is an updated version of Fig. 3 from the main text with savanna sites removed from the analysis. The removal of individuals in savanna sites did not change the general tendency of shifts in functional dominance across trait-space.

Response Fig. 1 **Amazonian community biomass re-assembly - forest only**: principal component analysis for each climate change scenario using all trees in the study region overlaid with an empirical 2D kernel density estimate of the average biomass change across trait-space between 2000 and 2100. Traits used as in Fig. 2. Hydraulic-strategy traits align predominantly along PC1 with productive vs. conservative strategy traits along PC2. For clarity arrow length was tripled and changes greater or less than 150 were set to 150. Arrows point toward: higher allocation of carbon to grow roots (A-Root), leaves (A-Leaf) and stems (A-Stem), versus deferred growth via higher allocation to storage (A-Store), higher water potential at which 50% loss of conductance (P_50), an evergreen leaf phenology (Eg) (as opposed to deciduous), light triggered leaf flush (Phenology) (as opposed to water triggered), increased maximum rooting depth (Root-D).

Also, we can see on Figs S3B-C there are different responses for some scenarios across different biomes. Part of the Amazon reduces the biomass, despite an increase in biomass in other areas. How did the authors deal with these different responses?

38. Thank you for querying our biomass responses and the inclusion of both tropical forests and savannas in our study area. We have now removed savanna sites from our study area and analysis. As per a request from another reviewer we have added figures showing our climate forcing data (Fig. S2). We have indicated in the text that areas that incur biomass losses or gains are associated with precipitation reductions or gains. We have added the following sentence to the text.

“Across all scenarios, biomass increases (Fig. S10) are associated with areas where precipitation remained stable or increased (Fig. S2 B, D).”

Lines 112-113 – trend of what? It is not clear in each direction the authors are referring to.

39. Apologies, it is not entirely clear what you mean here. The sentence on lines 112-113 “biomass changes across trait-space revealed shifts in the functional dominance of strategies and a tendency toward increased water triggered evergreen and deciduous strategies (Fig. 3)” describes shifts in how biomass is distributed across trait space.

We have revised this sentence to improve clarity and account for your comment 40 below. The revised text is presented in the following response.

Lines 111-115 – From the Fig 3, the results point to an increase in biomass in the scenarios with CO₂ increase for the species with higher growth and higher P50. Aren't these traits associated with more acquisitive strategies? And the scenarios with fixed CO₂ the phenology and root depth are associated with higher biomass loss.

40. Yes. You are correct. In Fig. 3 A and C there is also an increase in areas of trait space associated with higher growth (acquisitive) strategies. The increase in the area of trait space associated with higher P50 also corresponds to the area of deciduous phenology as the opposite side of the Eg arrow in Fig. 3 indicates deciduousness (the figure legends have been adjusted to make this clearer). In Fig. 3 B, D, biomass is reduced across trait space. The motivation for the supplementary figures was to assess whether there were indeed functional shifts being masked by the biomass reductions in Fig. 3 B, D. These figures (old S4 and S5) revealed very similar patterns to Fig. 3 A, B, i.e. increases in acquisitive and deciduous (high P50) areas of trait space.

The main purpose of these figures was to show that we predict functional shifts across trait space which broadly favour plants which can respond well to changes in water availability or take advantage of mortality events (and canopy openings) or increasing CO₂. These results are supported by empirical findings, e.g. Fauset et al. (2012) and Aguirre-Gutiérrez et al. (2016) describe increases in deciduousness in tropical forests due to precipitation changes and drought. Bonal et al. (2016) discuss potential increases in acquisitive strategies following drought mortality. Korner's (2006) review discusses how increasing CO₂ would favour fast growing species, either acquisitive (as above) or low wood density

(high P50) as you point out. These references have been added to the text which now reads:

“Biomass changes across trait-space revealed shifts in the functional dominance of strategies and a tendency toward increases in fast-growing productive strategies and increases in water-triggered evergreen and deciduous strategies (Fig. 3). In simulations with increasing CO₂ (Fig. 3 A, C), biomass increased in areas of trait-space associated with productive strategies and water-triggered strategies (Dec Water, Eg Water), whereas with fixed CO₂, biomass loss dominated (Fig. 3 B, D). However, examination of frequency and biomass proportion changes across trait-space reveal functional dominance shifts towards increased dominance of productive and water-triggered phenologies with fixed CO₂ (Fig. S47, Fig. S58). Such increases in productive strategies can be associated with vegetation responses to drought-induced mortality events⁴⁴ and increasing CO₂⁴⁵, while increases in the dominance of deciduous species in tropical forests have been documented following long-term precipitation reductions⁴⁶ and drought.⁴⁷”

Beyond the purpose of displaying broad shifts across trait-space, we interpret the nuances of the PCA plots with caution. To assist we included Fig. S6 which showed that the community mean trait values which are changing the most over the 100 years between 2000 and 2100 are evergreen and deciduous, and phenology (light and water) and thus focused on the largest changes in traits apparent in both sets of plots. However, we have since removed this figure (old Fig. S6) to reduce complexity. These results remain the same and this plot can be re-added upon request. If the reviewer had any suggestions how we should modify these PCA plots aimed at showing that climate change favours strategies in some areas of trait-space, but not others, we would be grateful. We can also generate additional plots to assist the comprehension and reduce complexity.

It is difficult to interpret the direction of these phenology and Eg traits as it is not very clear in the text neither in the legend what are their direction. Maybe make the response directions more clear.

41. The following text has been added to all PCAs to clarify the direction of traits and arrows: “Arrows point toward: higher allocation of carbon to grow roots (A-Root), leaves (A-Leaf) and stems (A-Stem), versus deferred growth via higher allocation to storage (A-Store), less negative water potential at 50% loss of conductance (P_50), an evergreen leaf phenology (Eg) (as opposed to deciduous), light triggered leaf flush (Phenology) (as opposed to water triggered), increased maximum rooting depth (Root-D).”

This is an interesting result as the CO₂ increase may favour some species strategies. Even more hydraulic vulnerable species (which are usually associated with higher acquisitive strategies) could have been favour by the increment of

CO₂ as they can maximize their photosynthesis without needing to lose much water. In higher CO₂ plants can increase their water use efficiency, counterbalancing the reduction in water availability.

42. In response to your comment 41 and this one we have modified the text of this passage and added relevant references. We also added text to later sections (lines 319-322). We refrain from further discussion however as the text is already long and the main focus of the manuscript is to quantify diversity effects, identify traits responsible for these effects, and highlight linkages between ecosystem function, climate change mitigation, and the conservation of biodiversity.

Figure S6 – What is absolute % change related to? Biomass?

Also the density distribution made for each scenario is associated to the number of model runs? Sorry, but I did not understand it very well. Needs more information in the subtitles about what is going on there.

43. Apologies for the confusion. This Figure has now been removed as it was not useful. To generate this figure we:

1) returned the abundance weighted community mean value of each trait at each site in the study area for the years 2000 & 2100 using the R function “dffD” in the “FD” package.

2) calculated the percentage change between 2100 and 2000 for each trait at each site.

3) took the absolute value of this percentage difference for Fig. S 6.

This plot was intended to ease identification of traits which undergo changes between 2000 and 2100 and allow the identification of differences in changes between RCP scenarios, for this reason we used the absolute values of differences as we were not directly interested in the directionality of change here.

According to all PCAs results there are an increase in the biomass and individual numbers in other strategies that are not associated with phenology. Maybe I got something wrong.

44. You are correct. We have modified the text in response to this comment and your comment 40.

Lines 124-126 - This is related with the different responses that you have in different areas. Here you say that the biomass reduced in 40% in Amazon, but in lines 108-109 you say the biomass increases.

45. What we mean here is that, even with increasing CO₂, there are a large number of simulated gridcells in the north east of the simulated area where biomass is reduced by up to 40%.

Maybe the authors need to mention this heterogeneity more clearly in the text, including the spatial information about the data they used for these results. The figure 3 is only for Amazon? Or it has all data? Maybe separate the analysis per biome.

46. All savanna biome sites have been removed from the study area & analysis. See responses 23 & 36 above.

Lines 132-137 – Not sure about this term ‘collapsed’. It reads weird for me and not very clear. Maybe say you combined or not distinguished some traits.

47. Thank you. Collapsed has been removed. We now write: “we then reduced the maximum volume of trait space and re-ran scenarios by 1) removing deciduous water-triggered strategies; and 2) additionally removing evergreen water-triggered strategies.”

Also, in response to this comment and that of another reviewer, maps (Figs. S 15) have been added which show the extent to which 1 & 2 above reduce functional diversity across the study area.

And just say that in the first step you removed one strategy or other, instead of preventing the emergence of a certain trait? Sometimes it is better to say straight what do you mean instead of used other words that can make it confusing.

48. Thank you. “Preventing the emergence of” has been replaced with “removing”.

Lines 151-152 – The authors discuss about niche differentiation. How much were the deciduous traits associated with the other strategies? Because the phenology triggered by water or light is probably associated to some plants strategies. The deciduousness is a strategy, but the factor that triggers it is more associated in this case with how is the traits that will receive the perception from outside (environmental trigger) and will reflect on the leaf shedding. So for example, the root depth, could be the main trait associated to get the response of how much water is in the soil (at the soil root surrounding the root), as when

there is no water enough in the soil the plants lose their leaves. How did you deal with this in the paper? The niche in this case they would be more associated to the root than to the phenology indeed.

49. In reality, and in a process-based model like here, a niche hardly ever results from one trait only. Trait associations are shown in the PCAs (Figs. 3, S12, S13), for example, and some traits are more important for competition for water while others determine the competition for light. Disentangling the role of all individual traits would be beyond the scope of this study. We do however recognise the importance of disentangling the importance of teasing apart cause and effect traits. Thus, in response to this comment and a comment by reviewer three we have run additional simulation experiments. Below ground rooting niche differentiation is now plotted in Fig. S5. We explicitly tested the effect of removing rooting niche differentiation on the response of simulated forest stands to drought (Fig. S6). These results show that removing below-ground rooting niche differentiation strongly increases drought impacts. No other traits were changed for this experiment and all state variables (biomass, tree height, leaf area index, etc.) were identical at the start of the simulated drought. As such we can infer causation, at least within the model. We have also run a control simulation for this experiment where niche differentiation was removed but no drought was applied to demonstrate that the resulting additional biomass reductions are caused by the interaction of the removal of niche differentiation and drought.

We have not yet devised an appropriate experiment to properly test other traits (above ground traits) as changing, for example, P50, would also change specific leaf area, which would change leaf area index and thus affect photosynthesis and light competition. Likewise, changing P50 would change wood density which would affect the heights of stems and thus alter light competition and carbon gains. We will endeavour to devise appropriate experiments to test all traits in future studies.

Also, I would not call only the phenology (which was the focus of this last part of the manuscript) as a hydraulic strategy. It can be part of it associated to the water demand, but hydraulic strategies are broader, and involve other levels of water uptake, water transport, and capacitance.

50. Thank you, we agree to an extent. We hope that our response to your comment 5 addresses the reservations expressed in this comment sufficiently.

Methods

Model description section

It was nice the section about this model, especially as it is a new approach and

apparently very promising. It is good to know the possibilities of the model, but it needs to get clear if the authors used all the utilities in the current manuscript. I think the part of mutations and cross over were left aside, so maybe include it here.

It would be interesting to know exactly all the traits that were used as an input, as well as some clarification about how the species links with traits. As far as I understood, each species has one functional trait, right? Or each species is a combination of traits?

51. Thank you for your comment and interest. The main text has now been revised to include a concise description of model concepts necessary to follow the main text. The supplement has been extended with additional details, and a full model description has been added as an appendix with references in the main text to all relevant sections, tables, and equations.

Also it is not clear how exactly the selection operates and the probability of changing a trait? It would be nice to have more details about how the model really works with this evolution aspect. Is it at individual level, or species level?

52. It's individual level. We hope that the responses to your previous comment (12), additional clarification in the main text, full model description as an appendix, and references to appropriate sections, equations and tables in the model description will clarify how the model works.

Lines 304-305 – Here you say it is possible to have variability between individuals in a community. Is here the variability intra specific (within the same population)? Or the variability within the community is only the variability across species?

So is the selection operating over species?

53. Variability can be both inter and intra specific with variability in traits both within and between species. This applies for all simulations run with the exception of simulations run for Fig. 1 C, D. For Fig. 1 C, D different species had different trait values for each trait but every individual belonging to a particular species had identical trait values.

In the model every individual has a set of adaptive traits (a full list of traits is included in the appendix as Tab. A3). The ranges of possible trait values are defined in Tab. A3. The trait values of each individual can potentially be unique. An individual's trait values do not change during its lifetime. The values of an individual's traits are randomly chosen when the model run is initialised and

then inherited from a parent individual. Inherited traits are those of the parent individual but are modified by allowing trait values to be swapped out “crossover” with the values of another individual which has the same species label. The probability that trait values crossover is fixed in the model.

At model run initialization, every individual is randomly assigned a “species label”. This label ensures that only individuals which have the same species labels can exchange trait values. Offspring inherit their species label from their parents. In Scheiter et al. (2013) we found that reproductive isolation between groups of individuals, which we code for using the species label, promoted diversity and the emergence of distinct strategies in the model.

As such we can calculate variability between all individuals in a simulated vegetation stand. We can also aggregate to the “species” level using the species labels and can calculate variability at this level.

The summed costs and benefits of an individual's trait values under the particular abiotic and biotic conditions determine their relative reproductive success. As such, natural selection acts at the level of the individual.

When you say trait values can mutate at fixed probabilities, in which proportion these mutations affect the phenotype? As the phenotype is where the selections are made, right?

54. Yes, the selection is on the phenotype. Mutations are restricted to newly produced seeds and the extent to which a particular mutation affects the phenotype (i.e. change in a particular trait value for a particular trait) is a function of the abiotic and biotic conditions. In response to these comments and your previous comments we have re-written the introductory paragraph to provide a better explanation of how the model works. This text has been added as part of the response to your comment on Lines 52-54 of the original manuscript.

Model forcing and simulation set up

Line 316 – write what CRU stands for.

55. This has been done.

Line 324 – write what RCP stand for.

56. This has been done.

In this section is good to punctuate what simulations are associated with each

part of the study. Maybe at the beginning after the model explanation, it would be nice to have a brief resume of the parts of the study and what was analysed in each section.

57. A table (Tab. S2) has been added which details all simulation experiments run in this study.

The terms catastrophic and chronic could be used here to say what you are talking about and if the simulations only involved the scenarios or they had additional precipitation analysis, like the case of your last analysis. I assumed the precipitation was only manipulated in the last analysis, right?

58. Apologies for any confusion. For the site-level drought experiments, precipitation was reduced by 50% for a number of years that corresponded to the number of years rainfall exclusion experiments were conducted at these sites (4 & 7 years).

For the RCP scenarios, precipitation was reduced or increased according to the anomalies calculated for each respective scenario. Plots have been added to the supplement (Fig. S2) showing CRU-derived annual precipitation (Fig. S2 A) and the percentage change in precipitation for each RCP scenario in 2100 (Fig. S2 B, C). A table (Tab. S2) has also been added to the supplement. Tab. S2 details any forcing manipulations applied to each simulation experiment presented in the manuscript.

With respect to this and one of your previous comments, the terms catastrophic and chronic have been combined with qualifiers throughout the text to improve clarity. As per your recommendation, we now write “sudden catastrophic drought” and “chronic precipitation reductions” throughout the manuscript.

Site level and experimental set up and analysis

It is good that you added a section about the experimental analysis, but maybe add more detail about the ground data that you used for the comparison.

59. Thank you. Additional text for each tropical forest site has been added to the section “Site level experimental setup and analysis”.

Line 365 - Not clear the combinations you have used here - is this 1 to 12, and he following combinations? 18 combinations? Maybe say it first.

60. We have clarified this in the caption of Fig. 1. and in the supplement. This line is now:

“To test the effect diversity has on the response of vegetation to drought we ran simulations where the model was initialised with the following 18 different numbers of species (1, 2, ..., 12, 16, 32, 48, 64, 80 and 96).”

Line 370-371- It is not clear for me whether the species has one trait only or they have multiple combinations of traits. This can be sorted as my previous comment when you explain about the model.

61. Apologies for the lack of clear description. We have addressed this comment in response to your previous comment and added a full model description as an appendix.

When the author use species-trait-pool, depending it would be better to keep only species-pool. I understood the species do not repeat, but I guess some traits may repeat, right?

62. Apologies again, this should have been more clearly explained. All individuals possess all traits. In this experiment, only trees were simulated. We added a table (Tab. A3) which lists all traits and their ranges. The values of these traits can vary from individual to individual. The “species” that were created for this experiment had a single value for each trait listed in Tab. A3, as such, all traits repeat. The values for each trait differ between species. In order to choose the trait combinations which performed well in previous simulations which we could then use to initialise the diversity experiment in Fig. 1 C, D, we identified the species in each replicate for each site in Fig. 1 A, B that had the highest number of individuals. Taking all individuals belonging to this species label, we then calculated the mean value for each trait. For the diversity experiment, the value of each trait, for each of the 96 species in the species-trait-pool, was then set to this mean value. No two species have the exact same set of trait values, though some do cluster closely in trait space as evident in Fig. 2.

We would prefer to retain the species-trait-pool terminology as it accurately describes what it is, i.e. a list of species and their respective trait values. To calculate functional diversity we needed both a species-trait-pool (species ID and trait values) and abundance data (species ID and abundance in each of the 18 different treatment levels and replicate (96 for each treatment level)).

Was the mutation off for all this manuscript, maybe make it clear when you are talking about the model.

63. No. Mutation and crossover were only turned off for the drought-diversity experiment. This was done to prevent diversity from emerging via mutation and subsequent crossover. The output from these simulations are presented in Fig. 1 C, D. Mutation and crossover were turned on for all other simulations in the

manuscript. A table (Tab. S2) has been added to the supplement. Column three in this table details whether and how diversity was manipulated and specifies whether mutation and crossover were turned on or off.

Lines 382-391 – These are the model traits, no? Maybe it should be included in the model explanation. Were there any options of traits? It is good to know why the authors used these ones and not other traits. What was the criterion for that?

64. These were all model traits. The description of the model and traits has been improved in the main text and a full model description has been added as an appendix. A table listing all adaptive traits which can evolve through generations has now been added to the supplement (Tab. A3).

All traits (Tab. A3) were included in an initial PCA analysis. The traits were selected based on their loading on the first two principal component axes. We stated this in the original submission but omitted to include a reference to support this method. We have added a citation for a study by Zhu et al. (2017). The authors suggest multiple methods of selecting traits to calculate functional diversity, one of which being to choose “the traits that had the highest loadings in the first several ordination axes”. Further methods suggested by the authors, e.g. selecting “traits considered to be important dimensions for plant ecological strategies” would also have resulted in the selection of traits which mediate plant water status given our focus on changes to plant water availability but would have missed the emergent trade-off axis in allocation of carbon to immediate growth vs to storage.

Zhu, L., Fu, B., Zhu, H. *et al.* Trait choice profoundly affected the ecological conclusions drawn from functional diversity measures. *Sci Rep* 7, 3643 (2017). <https://doi.org/10.1038/s41598-017-03812-8>

Lines 397-399 confusing

65. This sentence has been simplified as follows: “Four clusters were chosen for the cluster analysis as this resulted in two separate clusters which delineated evergreen trees with different leaf-flush triggers (light/water).”

Lines 404-407 – A better explanation about this should be added at the model explanation as well. It would be good to know how these traits are associated in the model.

66. This analysis has been removed from the manuscript as you correctly pointed out that it essentially repeats what was already found in Fig. 1 C, D. It has been replaced with a simulation experiment which identifies below-ground

rooting niche differentiation as one mechanism responsible for simulated diversity effects. Additionally, the explanations of model functionality have been significantly revised throughout the manuscript and a full model description, with appropriate cross references in the main text, has been added as an appendix.

Lines 409 – Why these types did not occur? Were they excluded at initial runs of the model? Maybe discuss a bit more about this analysis as it was not very discussed in the main text.

67. It is difficult to answer the question why deciduous tree strategies that abscise and flush their leaves in response to changes in light did not occur in the study area, with certainty. The combination of a deciduous phenological trait value, and a phenological trigger trait value that makes a plant trigger leaf flush and abscission in response to changes in light, would be assigned randomly to individuals in the initial population of plants with a probability of 0.25 ($\text{prob_deciduous}=0.5$, $\text{prob_light_trigger}=0.5$). Therefore, we assume that this strategy is completely outcompeted across the study area because it does not perform well under these particular abiotic conditions when in competition with other strategies. Simulations run for the northern hemisphere temperate zone show that this strategy can emerge as a dominant one in the model given the right abiotic conditions. Whether the appropriate phenological trigger in this zone should be light or temperature is arguable. Light and temperature in the northern temperate zone are clearly highly correlated. Evidence suggests that it is likely a combination of both light (photoperiod) and temperature (Polgar & Primack, 2011).

Since this analysis has been removed from the manuscript we refrain from additional discussion in the main text. In future work we will however be focussing on large scale distributions of leaf phenological strategies.

Polgar, C.A. and Primack, R.B. (2011), Leaf-out phenology of temperate woody plants: from trees to ecosystems. *New Phytologist*, 191: 926-941.
<https://doi.org/10.1111/j.1469-8137.2011.03803.x>

Lines 425 and 427- Give more details about these scenarios. Maybe at the section about Model forcing and simulation set up.

68. Thank you. We have substantially revised the section “Model forcing and simulation setup” in the supplement to address this comment and comments by reviewer 2.

Lines 463-464 – Why this difference between the two scenarios?
Check in these analyses if all the points used were the ones in Amazon.

69. Owing to the additional analysis conducted for Fig. 4 we have removed old Fig. S7 as it is superfluous. The differences between RCP scenario anomalies and CO2 forcing is now graphically displayed in Fig. S2. Savanna sites have been removed from the manuscript as detailed in the responses to previous comments.

Line 465 – Maybe separate this removal of strategy part as another section, as apparently there was an additional change in precipitation, right?

70. The climate forcing applied to simulations described in sections “Amazonian climate change scenario simulations - non-constrained diversity” and “Amazonian climate change scenario simulations - Ecological strategy removal” (now “Amazonian climate change scenario simulations - Reducing functional trait diversity”) were identical. To improve clarity we have added a table (Tab. S2). The fifth column of the table “Forcing variable manipulation” describes changes to forcing variables applied for each simulation experiment.

Although there were many issues about more clarity in the manuscript it is a very interesting and important manuscript.

Sincere thanks for your positive feedback and great effort taken to assist us in improving the manuscript.

Reviewer #2 (Remarks to the Author):

General comments

The manuscript Langan et al. addresses the role of diversity for vegetation resistance to water deficits - a timely and important topic. The authors find, through comprehensive modelling and analyses, that hydraulic trait diversity plays a critical role for tropical forests to withstand droughts. If substantiated, the findings are novel and noteworthy.

However, the paper fails to present the methods and results in a clear, understandable and comprehensive manner, which detrimentally affected my ability to assess the soundness of the research design and methodology, and the significance and scientific robustness of the conclusions.

Dear reviewer, thank you for your positive feedback and detailed constructive commentary. The manuscript has been substantially revised to address comments. The methods section has been extended and revised to clarify methods. A full model description has been added as an appendix with references added throughout the main text to relevant sections. The main text has also been substantially revised to improve clarity.

In particular, I recommend the authors to address ambiguities in definitions, unclear descriptions of model validation/evaluation and unclear descriptions of the comparability of the experimental sites and the large-scale simulations.

Thank you for your comment. We have substantially revised the manuscript in light of your comments. We detail these revisions below.

Some additional explanation of the rationale and the results in view of previous literature, as well as additional basic plots of input data and study area would also be useful, as discussed below.

71. Thank you for your helpful and constructive comments. Basic plots of input data have been added (Fig. S2). Additional explanations of rationale and results in view of previous literature have been added throughout the main text (e.g. lines: 47, 64-69, 146, 180-188, 198, 205-210, 237-240, 249-260, 302, 311, 319, 324-326, 334-338).

For example, given the focus on diversity, a precise definition is essential. However, a variety of terms regarding diversity are used in the manuscript, sometimes seemingly referring to different concepts and sometimes used interchangeably (e.g., diversity, functional diversity, trait diversity, hydraulic trait diversity, drought-diversity, phenological strategy diversity). “Constrained diversity” is also seemingly used to refer to different types of treatments (one for the experimental sites, and one for the large study area), which is confusing.

72. In response to this comment and comment 16 from reviewer 1 we have revised the text to be more concise about the type of diversity we are mean.

We now define what we mean by hydraulic traits, i.e. traits that mediate plant water status, in the summary paragraph. The rationale for this definition is provided in response 5 to reviewer 1. Throughout the manuscript we now refer to functional trait diversity.

We have also removed the phenological strategy diversity analysis from the manuscript and replaced it with one where the effect of functional trait diversity is estimated (Fig. 4). This removes some additional jargon and addresses your comment 84, maps of functional diversity (Fig. S15) have now been added to the manuscript.

Langan et al first run simulations in the experimental sites to reproduce the level and rate of biomass loss, referring to Fig. 1a and 1b to state that this is achieved. It is unclear to me how Fig 1a and 1b shows this. How should the reader interpret the “horizontal dashed line” that is supposed to show observed reductions in terms of “rate of biomass loss”, given that it is fixed over the entire period?

73a. Thank you for your comment. In response to this comment and a comment from another reviewer Fig. 1 A, B has also been altered to show the duration of field and simulated drought treatments applied. The horizontal line has been replaced by an X to indicate both the year in which final field measurements were carried out to assess the effect of the drought in these forests and the percentage change in biomass that was observed.

We have removed the “rate of biomass loss”. The text now reads “Simulations showed that the percentage of biomass lost and the timing of these losses were reproduced at both sites (Fig. 1 A, B)”. The text in the figure legend has also been modified. We believe these changes improve clarity by making the duration of observed and simulated drought treatments, and the timing of observed biomass reductions, more obvious.

How comparable are the “96 replicates simulations” with the actual experiments at the site?

73b. Apologies, I’m not certain what you mean by “comparable with the actual experiments”. There was no replication of the experiments at each site due to costs and time.

Real-world vegetation stands exhibit considerable heterogeneity in their biomass, tree cover, demographic structure, composition, successional stage, and vegetation type under similar abiotic conditions. A common approach with vegetation models to capture this variability is to simulate partly stochastic processes at the level of a forest patch, such as a canopy tree dying leaving a gap for pioneer trees, with probabilistic functions. To characterise the whole forest, then the results of a number of such replicate patches are averaged (often 50 to 100, e.g. in the LPJ-GUESS DGVM). The approach in aDGVM2 is similar, as individual model simulation results are heavily influenced by random processes. Therefore, the observations should be compared to the average model results (Fig. 1 A, B), but for transparency, we show the results from all replicate model runs.

Regarding real world heterogeneity, within the Tapajos and Caxiua national forests there is considerable variation in both vegetation type and biomass. For

example, reported values of above ground biomass at forest sites within Tapajós National Forest (TNF) reserve varies between 146 - 399 t/Ha (see Table below). Vegetation types within TNF range from closed to open ombrophilous forest to open savanna (Bispo et al., 2016). Reported values for biomass at forest sites within Caxiuanã National Forest (CAX) range from 333-456 t/Ha (see Table below). However, there are also multiple vegetation types within CAX that range from closed forest to open savanna (Plano de Manejo Floresta Nacional de Caxiuanã, 2012). Both biomass within forest stands and vegetation type can vary considerably within both reserves, we thus expect that, should replicates of drought experiments be carried out, the amount of biomass lost would vary depending on species composition, functional diversity, biomass, vegetation type, soil type, depth to water table. However, given that the dominant vegetation in both reserves is closed tropical forest, we expect the drought responses observed at the TNF and CAX sites to be broadly representative of average responses.

The DGVM used here has previously been benchmarked (above ground biomass, tree cover, tree height, biome distribution) against satellite derived data for the entire study area (Langan et al., 2017). These comparisons demonstrate that the model performed well as indicated by high R^2 values.

Study	TNF (t biomass / Ha)	CAX (t biomass / Ha)
Hunter et al. 2013	146-350 (Tab. 2, Km 83) 267-398 (Tab. 2, Km 67)	-
Keller and Palace 2001	157-399	-
Brando et al. 2008	ca. 275 (Fig. 2 b)	-
Napstad et al. 2007	302.6 (control plot) 310.6 (drought plot)	-
Mahli et al. 2009	287-350 (Tab. 1)	382-424 (Tab. 1)
da Costa et al. 2010	-	399.4 - 456.2 (control plot) 374.8 - 427.6 (drought plot)
Mahli et al. 2006	300.06 - 375.9 (Tab. A1)	333-430 (Tab. A1)

Avitabile et al. 2015	203-358	304-348
Baccini et al. 2012	193-317	252-267
Saatchi et al. 2011	224-304	252-330
aDGVM2 Full Model	182, 50 (mean, sd)	203, 56 (mean, sd)
aDGVM2 96 Species Treatment	187, 46 (mean, sd)	205, 55 (mean, sd)

How can the model results be trusted given the vast spread in model simulations (particularly for the CAX site in Fig 1b, where the min and max even seem to be cut off)?

74. As we explained in our response to the comment above, the average model results are what should be compared to the observations. This is now also clarified in the text as “We compared mean model responses of replicate simulations to observations as the model includes a range of stochastic processes”.

There are now no values cut-off in Fig. 1 A/B. In response to your comment above we have detailed the large variability in observed biomass at forest sites in these national parks. The real-world drought experiments have not been replicated. We expect that such replication would also yield variable responses. We have no way at this time of ascertaining whether our simulated variability in drought response is accurate. However, it is a common practice in vegetation modelling to simulate replicates or patches to account for stochastic processes and present the mean modelled values (Sato et al. 2007). Many DGVMs and related models include rather small-scale processes, such as the establishment, growth and death of individual trees. Commonly, these processes are simulated for a number of forest patches, with the size of about 1000 m² (e.g. in the LPJ-GUESS model, Smith et al. 2001, Smith et al. 2014), an approach that has been adopted from forest gap models (Bugmann 2001). At this scale, processes such as the establishment and mortality of tree individuals are to a large extent stochastic, in reality and in the models. Therefore, it is common practice to average the results of a number of replicate patches to characterise the average forest condition in a region or model grid cell (e.g. Smith et al. 2014, Fisher et al. 2017, Pugh et al. 2020). We have chosen to display both the mean response

and the variability about this mean. While the model has been extensively benchmarked for the entire study area, the model can produce stands of vegetation that differ substantially in terms of simulated biomass. A main focus of our modelling approach is to attempt to understand variability in community assembly, emergent traits, responses to perturbations, state variables such as biomass etc. As such we believe that our mean modelled percentage biomass reduction that is extremely close to observations represents the best that can be achieved at this time.

Sato, H., Itoh, A., Kohyama, T.: SEIB–DGVM (2007): A new Dynamic Global Vegetation Model using a spatially explicit individual-based approach, *Ecological Modelling*, <https://doi.org/10.1016/j.ecolmodel.2006.09.006>.

Smith, B., Wårlind, D., Arneth, A., Hickler, T., Leadley, P., Siltberg, J., and Zaehle, S.: Implications of incorporating N cycling and N limitations on primary production in an individual-based dynamic vegetation model, *Biogeosciences*, 11, 2027–2054, <https://doi.org/10.5194/bg-11-2027-2014>, 2014.

Smith, B., Prentice, I.C. and Sykes, M.T. (2001), Representation of vegetation dynamics in the modelling of terrestrial ecosystems: comparing two contrasting approaches within European climate space. *Global Ecology and Biogeography*, 10: 621–637. <https://doi.org/10.1046/j.1466-822X.2001.t01-1-00256.x>

Fisher RA, Koven CD, Anderegg WRL, et al. Vegetation demographics in Earth System Models: A review of progress and priorities. *Glob Change Biol*. 2018; 24: 35–54. <https://doi.org/10.1111/gcb.13910>

Pugh, T. A. M., Rademacher, T., Shafer, S. L., Steinkamp, J., Barichivich, J., Beckage, B., Haverd, V., Harper, A., Heinke, J., Nishina, K., Rammig, A., Sato, H., Arneth, A., Hantson, S., Hickler, T., Kautz, M., Quesada, B., Smith, B., and Thonicke, K.: Understanding the uncertainty in global forest carbon turnover, *Biogeosciences*, 17, 3961–3989, <https://doi.org/10.5194/bg-17-3961-2020>, 2020.

Bugmann, H. A Review of Forest Gap Models. *Climatic Change* **51**, 259–305 (2001). <https://doi.org/10.1023/A:1012525626267>

The model validation and evaluation step is critical for a model-based study - if the reader is to be convinced that the model outputs can be trusted, the model validation and evaluation needs to be robust with regard to the type of outputs of interest here.

75. We fully agree. Therefore, we have emphasized the model evaluation at the two drought experiment sites. We now also added additional emphasis on previous benchmarking in the text (lines 132-134). To further address this comment we have performed additional analyses. Given key outputs of the

model are plant traits we have conducted comparisons of simulated community weighted mean specific leaf area with all currently published predictions of specific leaf area (SLA) for the study region (Figs. S7, S8). These comparisons indicate that aDGVM2 produces a mean SLA for the study area which is very similar (in one case there is no statistical difference) to the data-driven mean values in the majority of these products. The range of mean SLA values for the study area is also quite similar to many of these products.

Furthermore, in view of the validity and robustness of the model simulations, the authors could also discuss why the biomass can increase in the model (Fig S1) when observation-based studies indicate that the net carbon sink has peaked in the Amazon already in the 1990s (see Hubau et al 2020, which Langan et al also cites).

76. Thank you for your comment/query. In light of this comment and the comment of reviewer 1 regarding our study area extent, which contains both tropical forest and savanna sites, we have separated tropical forest and biomass sites and removed savanna sites from this study.

To make our simulated changes in biomass more comparable with both Brienen et al. (2015) and Hubau et. al. (2020) we have added a plot of net biomass change per decade to the new Fig. S9. The figure (Fig. S9 B) reveals that our simulated net changes in tropical forest biomass are in general agreement with both Brienen et al. (2015) and Hubau et al. (2020). Despite the discrepancies between our study area (new Fig. S10 A, Response Fig 2 below) and the tropical forest sites considered in Brienen and Hubau, we show that peak biomass accumulation occurs in the 1990-2000 decade in three of four future scenarios (RCP4.5 Clim+CO₂, RCP4.5 Clim, RCP8.5 Clim)(Fig. S9 B). Peak biomass accumulation for RCP8.5 Clim+CO₂ happens in the decade 2070-2080. Additionally, we find that biomass accumulation of tropical forests crosses zero in three out of four scenarios. This threshold between being a sink and a source is crossed earliest (2000-2010) in the scenarios which do not include increasing CO₂ concentrations (RCP4.5 Clim, RCP8.5 Clim). In scenarios with increasing CO₂ (RCP4.5 Clim+CO₂, RCP8.5 Clim+CO₂), this threshold is crossed between 2070-2080 for RCP4.5 Clim+CO₂ and closely approaches this threshold by 2100 for RCP8.5 Clim+CO₂. These results are surprisingly consistent with those of Hubau et al. (2020), who predict that the Amazon carbon sink strength will reach zero between (2011-2089).

Response Fig 2: Simulated future biomass changes for overlap of sites in study area and Brien & Hubau et al. Red “X” indicates Brien et al. & Hubau et al. study sites. Grey boxes indicate simulated grid cells. A) spatial distribution of simulated grid cells and forest plots included in

To further assess how comparable our results are with those of Brien and Hubau we repeated the analysis conducted for Fig. S9 using only simulated grid cells which overlap with the coordinates of tropical forest sites in the above studies (Fig. S10 A, Response Fig. 2). This subsection of simulated sites does not include all sites from Brien et al. and Hubau et al. (Response Fig 2 A). However, in comparison to our entire simulation area, subsetting shows that: 1) the percentage change in biomass through time (Fig. S10 B) is lower or more negative across all RCP scenarios; 2) that the net decadal change in biomass (Fig. S10 C) becomes negative (a carbon source) in all scenarios by 2100; 3) in agreement with the predictions of the earliest transition to a net carbon source by Hubau et al. (2020), even in the RCP scenarios with increasing CO₂, the net zero line is crossed on multiple occasions beginning in the decades 2010-2020 & 2000-2010 for RCP 4.5 & 8.5 and; 4) that, while the selection or subsetting of sites clearly influences the timing of predicted transitions to becoming a net carbon source, all of our simulations results, and the results of Hubau et al. (2020), point toward the Amazon becoming a carbon source by the end of the 21st century, at the latest.

We have re-written the paragraphs which began at line 108 and included the following text to incorporate the new figures and address your comment:

“Empirical studies show that the Amazon’s ability to sequester carbon is decreasing⁴ and peaked in the 1990s.³⁵ This declining sink strength is generally

at odds with model predictions.³⁵ For our study area (Fig. S3) we find that peak biomass accumulation occurs in the 1990-2000 decade in three of our four future scenarios (RCP4.5 Clim+CO2, RCP4.5 Clim, RCP8.5 Clim)(Fig. S9 B). Additionally, in agreement with⁴, who predict that the Amazon sink strength will reach zero between 2011-2089, for our entire study area we find that this threshold is crossed earliest (2000-2010) in scenarios without increasing CO2 concentrations (RCP4.5 Clim, RCP8.5 Clim). In scenarios with increasing CO2, this threshold is approached by 2100 in RCP8.5 Clim+CO2 and crossed between 2070-2080 for RCP4.5 Clim+CO2 (Fig. S9 B). Sub-setting our simulated data to include only sites used in^{4,35} (Fig. S10) we find that the sink-source threshold is crossed in all scenarios by 2090. Taken together these results add further evidence suggesting that the Amazon may become a carbon source by the end of the 21st century at the latest.“

With regard to the precipitation modelling component of the work, the authors used CRU data for the present and MPI precipitation output for the future, and used the gamma distribution from New et al (2002) to generate daily rainfall patterns. The validity of the gamma distribution assumption can be questioned, given that the character of precipitation is projected to change under climate change. Nevertheless, if the model responds to droughts at the monthly-to-seasonal scale (similar to in the real world), the choice of daily precipitation model should not critically affect the results. Please consider referring to work or comment on the sensitivity of the model to daily variations in precipitation.

77. The main focus of the current manuscript is to examine whether functional diversity affects the response of tropical forests to drought. As such, in the current manuscript, we are not overly concerned with how the character of precipitation is projected to change, though we recognise the importance of this. We have added text (lines 344-346) to indicate that a higher level of certainty in the future biomass storage potential of the Amazon could be achieved by using model ensembles of daily future change data. However, we have published a number of studies using aDGVM2 (e.g. Kumar et al. 2021) where the gamma-model has been replaced by downscaled ISIMIP daily precipitation data. This change from the gamma-model did not appear to affect the dynamics of simulated vegetation in tropical Asia and this model version performed well in meeting standard benchmarks. Further, the current model version is being used to take part in a tropical forest drought model intercomparison project (MIP). For this MIP the models are forced with observed daily precipitation values with drought treatments derived via methods similar to those presented in Longo et al. (2018). Though not published, aDGVM2 performs well in meeting standard benchmarks at the simulated sites and the response of vegetation to drought remains intact and reasonable. As such, we have no evidence which suggests

that the presented response of vegetation to drought is in any way an artefact of the precipitation model used.

Kumar, D., Pfeiffer, M., Gaillard, C., Langan, L., and Scheiter, S.: Climate change and elevated CO₂ favor forest over savanna under different future scenarios in South Asia, *Biogeosciences*, 18, 2957–2979, <https://doi.org/10.5194/bg-18-2957-2021>, 2021.

Longo, M., Knox, R.G., Levine, N.M., Alves, L.F., Bonal, D., Camargo, P.B., Fitzjarrald, D.R., Hayek, M.N., Restrepo-Coupe, N., Saleska, S.R., da Silva, R., Stark, S.C., Tapajós, R.P., Wiedemann, K.T., Zhang, K., Wofsy, S.C. and Moorcroft, P.R. (2018), Ecosystem heterogeneity and diversity mitigate Amazon forest resilience to frequent extreme droughts. *New Phytol*, 219: 914-931. <https://doi.org/10.1111/nph.15185>

Please also consider providing plots and comparisons of the precipitation inputs used, as well as clarifying the input data in existing plots.

78. Plots are now provided (Fig. S2) showing mean annual precipitation for the CRU forcing data (years 1961-1990), mean annual temperature for the CRU forcing data (years 1961-1990), and precipitation and temperature anomalies for RCP scenarios 4.5 and 8.5 in the year 2100.

It is for example unclear what precipitation data were used in Fig.1 (In Fig 1b, the years extends to 2010, but it is stated in the Methods that CRU is until 1990, the historical MPI output until 2005).

79. Apologies for the unclear description. We have significantly revised the section “Model forcing and simulation setup” to improve clarity. The text now reads: “As reference climatology, we used the Climate Research Unit (CRU)⁶⁴ data set, representing climate for the 20th century. This data set provides monthly mean data for temperature, temperature range, precipitation (monthly mean and coefficient of variation), number of rain days, sunshine hours per day, humidity and number of frost days averaged for the period 1961 to 1990 (that is, 12 monthly values per variable).”. Thus, these data provide 12 monthly values for precipitation (mean and coefficient of variation) which represent the average values for the period 1961-1990. For the last 100 simulation years, RCP precipitation anomalies are then applied to alter the mean monthly precipitation values; the coefficient of variation values remain unchanged. RCP anomalies for the last 100 simulation years correspond to the years 2000-2100. The description of how RCP anomalies were created has been extended in the section “Model forcing and simulation setup”. Briefly, the historical data is from 1850-2005. The future data is from 2006-2100. Historical and future data were combined and anomalies were created with a zero-anomaly period centred on the decade 2001-2010. We chose this decade as this represents the join of historical and

future data. Thus, for the drought experiments presented in Fig. 1, RCP 4.5 precipitation, temperature, and CO₂ anomalies are applied (Tab. S2). However, given that precipitation and temperature anomalies are centred on zero change for this decade (2001-2010), we do not expect that these anomalies have any discernible influence on simulated drought responses.

Specific comments

Recommend adding an overview table of all the experiments in the SI (including information such as data input, biodiversity model set-up, time period, drought length/timing, and study area). To would help the reader to grasp how the simulations for the “experimental sites” and the “future climate” differ.

80. Thank you for your suggestion. This has been done (Tab. S2).

Please consider providing an explanation of the use of 96 species.

81. The choice of 96 species is related to both the simulation of an appropriate number of replicates to have confidence that we are accurately representing mean responses, i.e. $n > 30$ the central limit theorem applies, and resource constraints at the time of running the simulations. On the high performance computing (HPC) cluster which was used the number of processors per core was 24. 24 divides evenly into 96 and as such occupies complete nodes and avoids wasting computing capacity. Scaling up computing resources for the diversity experiment (Fig. 1 C, D) 18 treatments (number of species used to initialise the model) and 96 replicates (each replicate drew a different species or different sample of species from the 96 total species) required 1728 separate simulations for each site. These 3456 separate model runs represented the approximate maximum computing resources which could be taken at the time which would deliver results within approximately one week (3456 runs, 3 nodes with 24 processors per node (72 processors), $3456/72 = 48$ runs per processor. 48 runs with ca. four hour simulation time per run = 8 days). To explain this, a sentence has been added to the Methods (line 798): “The choice of 96 species might appear arbitrary. This number was however chosen to use existing computational resources most efficiently (24 processors per core, using 4 cores).”

Please consider adding information When did the droughts occur? (Fig. 1)

82. The horizontal line which indicated the amount of biomass lost during drought at the throughfall experiments has been replaced with an “X” to indicate both the percentage change in biomass and the year in which this loss was

observed at these sites. A horizontal line has been added to indicate the duration of the throughfall experiments and the simulated drought in the model.

Fig. 4 What is constrained (strategy removal)? What is the baseline (without precipitation reductions)?

83. What was meant was that the possibility for plants to have trait values which defined a deciduous strategy which flushed and abscised its leaves in response to changes in soil water was removed. The possibility for plants to have trait values which defined an evergreen strategy which flushed its leaves in response to soil water was additionally removed. In response to some of review 1's comments and these comments we now state more clearly in the text what was done (lines 272-274).

In response to your previous comment we now calculate functional trait diversity for the study area in the year 2000 (Fig. S14 A) which represents the baseline simulated predictions (as suggested) of functional diversity. Baseline precipitation and temperature plots have been added to the supplement (Fig. S2). We also present results of changes in functional trait diversity owing to the removal of the deciduous and evergreen strategies (Fig. S14 B, C). Having generated these new calculations of functional trait diversity we availed of the opportunity to use them to isolate the effect of functional trait diversity on biomass loss statistically and present these results in a new Fig. 4.

Please consider adding spatial map of functional diversity, including model initial conditions.

84. We have now however added a spatial map as requested (Fig. S14). Doing so allowed us to additionally display how our applied treatments, i.e. the removal of evergreen and deciduous trees with a leaf phenological trigger that responds to water, reduces functional diversity across the study area (Fig. S14 B, C). Calculating functional diversity for these set of simulations allowed us to improve the statistical methods used for Fig. 4 as we could then use calculated functional diversity as a quantitative explanatory variable rather than simulation ID as a treatment with three levels. Thank you for the great suggestion. Baseline precipitation and temperature plots have also been added to the supplement (Fig. S2).

To address a previous comment of your regarding appropriate benchmarking for a given enquiry and to improve confidence that our model is producing community mean values of functional traits, we now include a map of simulated specific leaf area (SLA), compare this with a suite of existing predictions, and show that our simulated SLA corresponds well with data-driven remote-sensing

derived estimates of SLA. SLA is correlated with many other above and below ground traits (Bergmann et al. (2017)).

With regard to the initial conditions, the model initialization of traits is identical in every grid-cell. Following initialisation, varying abiotic and biotic conditions allow trait diversity to emerge through evolution and ecological filtering (competition). When we are running standard simulations in which we are not purposefully manipulating the area/volume of trait space, the trait values of each individual in the initial population is randomly chosen with values for each trait being drawn from a uniform distribution bounded by the minimum and maximum values for each trait (see Tab. A3). As such the initial populations of all simulated grid-cells are approximately equal and also approximately equal to the maximum volume of trait-space available in the model. This volume is filled with an approximately even density of individuals. We present the baseline level of functional diversity before climate change treatments are applied in Fig. S14 A and append this below.

Figure Response FD: Functional diversity and reductions caused by removing plant phenological strategies. A) Functional diversity for the study area calculated as Rao's quadratic entropy. B) Reductions in functional diversity when deciduous plants with a phenological trigger

Please consider adding more cross-ref to Extended Data and add legend to figures. (Needed for example in Fig. 2)

85. Additional cross-references have been added to the main text and supplement with extensive additional reference to relevant sections, equations,

and tables in the full model description which has been added as an appendix. A legend has not been added to Fig. 2. I made a version of these plots with legends but they took up too much of the plot/page space. I have removed them but improved the description in the text to make it clear that the different (colours+symbols) represent the clusters that species were assigned to by the clustering algorithm. The large symbols which represented cluster centres have been removed to further simplify these plots.

Suggest adding a map of the location of the sites and the study area for the “future” simulations.

86. A map indicating both the continental scale study area and the location of the CAX and TNF national forest sites has been added (Fig. S3).

The studies referred to previous studies that link diversity and stability of ecosystems and drought and biomass reduction. However, the reasoning for why the two should be linked, i.e. between diversity and biomass reduction due to drought needs to be more elaborated. It seems that only Isbell et al., 2015 makes this connection. This lack of reference also makes it difficult to discuss the results, other than the validation. For instance, Powell et al., 2013 explores what might be missing from estimating biomass reduction due to drought. How would this study perform better in representing the plant hydraulics compared to Powell et al., 2013.

87. We have reformulated the text emphasising that there is increasing evidence that hydraulic traits are very important for drought responses, but very few mechanistic models include tree hydraulic architecture and variable traits of competing individuals (instead of PFTs with more or less fixed trait values). This has hampered mechanistic understanding. We have also added text to discuss deficiencies in previous modelling approaches due to lack of drought responses and lack of traits which mediate plant responses to changes in water availability (see also Response no. 6).

Rather than simply elaborate on the linkages between diversity and biomass reductions we have conducted additional analyses and simulation experiments to identify differences in the stability of transpiration (Fig. S4) which may mediate drought responses. We then conducted simulation experiments where we identified below-ground rooting niche differentiation as one mechanism responsible for simulated linkages between functional trait diversity and higher resistance of diverse communities to biomass loss (S6). We have linked this in the text to multiple sources of empirical evidence related to tropical forest drought responses and diversity of hydraulic traits.

We feel that the space allocated to these additional references, text, analysis, and simulation experiments allow improved discussion potential beyond validation.

Similarly, characterization of diversity in mediating responses of tropical forests to drought in comparison to grassland is not discussed.

88. In response to a comment by reviewer 1 (response 6) we have removed some of the references to grasslands and added additional references for forests.

For example: line 10 now reads “Empirical evidence suggests that species and trait diversity mediate the response of forests to drought^{5,6} with water-related traits (hereafter hydraulic traits) playing a key role in mediating vegetation dynamics,⁷ productivity⁸ and drought responses;⁹ yet, little evidence exists for the response of tropical forests to drought.”

We add additional comparisons between our simulation results and evidence emerging from BEF-China (lines 180 -188). We now reference Poorter et al. (2015), who found that species diversity enhances tropical forest carbon storage, as our simulations indicate that functional trait diversity very clearly enhances carbon storage in our simulations (line 269). On line 297 we now cite Liu et al. (2022) who found that species diversity increased drought resistance in forests globally. Our results find similar results but for diversity in plant traits associated with plant water relations. Liu et al. also provide evidence that diversity effects (species diversity) increase in magnitude with increasing drought intensity. We find a similar relationship (Fig. 4) whereby the slope of predicted diversity effect lines increases with increasing drought intensity. We additionally cite Anderegg et al. (2018) who found that diversity in traits associated with water transport buffers variation in ecosystem fluxes during dry periods across temperate and boreal forests.

We hope that these additional references and discussion sufficiently supplant additional comparisons of diversity effects between forests and grasslands.

Figure 2: four clusters to be mentioned in the figure directly, is this referring to ecological strategies on page 24.

89. Sorry for the confusion. Four clusters are now mentioned directly in the figure “Displayed are four clusters identified in the (species x trait) matrix”.

Yes, this is referring to the ecological strategies on page 24.

What is referred to as biomass re-assembly here? And why is it mostly referred to biomass reduction in Figure 3?

90. Community assembly deals with how communities are structured with respect to species, their traits, and their relative abundances or biomass within the community. Under steady state conditions the model simulates an emergent distribution of traits and biomass is apportioned across it, i.e. an assembled community. When conditions change, as is the case for our future change scenarios, communities may or may not re-assemble. What we show here is that re-assembly (change in the biomass distribution across trait-space, change in number of individuals across trait space) does happen, and that changes are not evenly distributed across trait-space. This implies changing relative fitnesses of plant strategies, as mediated by traits, and that the changes are influenced by the climate change scenario and treatments applied. To improve clarity we have relabelled this figure “Change in biomass across trait-space” which is also more consistent with the sister plots in the supplement.

We have added the sentence “Across all scenarios, biomass increases (Fig. S11) are associated with areas where precipitation remained stable or increased (Fig. S2 B, C).”

Regarding the second component of this comment, “why is it mostly referred to biomass reductions in Figure 3”. In the text we refer to both increases (Fig. 3 A, C) and decreases (Fig. 3 B, D). With Figs. S12 and S13 we show that even though biomass across trait-space is decreasing everywhere without increasing CO₂ (Fig. 3 B, D), relatively consistent changes in the relative fitness of strategies become apparent when one considers changes in how the total number of individuals is apportioned across trait space (S12) and changes in how the total biomass of the study area is apportioned across trait-space (S13). Our main point with these results is that, irrespective of climate change scenario, we simulate changes to the way trait space is filled as selection shifts vegetation towards new optima (areas of trait space).

We have added a sentence to this paragraph to add additional context to these findings (line 239): “Such increases in acquisitive strategies can be associated with both vegetation responses to drought-induced mortality events⁴⁴ and increasing CO₂⁴⁵ while increases in the dominance of deciduous species in tropical forests have been documented following long-term precipitation reductions⁴⁶ and drought.⁴⁷”.

We have also modified and added text (lines 319-324) to add additional context to these findings: “Additionally, we simulate changes in the biomass apportionment across trait-space as communities re-assemble, and changes in the functional dominance of strategies. Across the Amazon, large changes have occurred in the functional composition of forests due to climate changes during

the Holocene and Pleistocene.⁴⁴ Our simulations suggest that, over the coming century, these forests will again need to adapt to novel climatic conditions and that the maintenance of trait diversity is crucial to insure such adaptation can happen.”

Please consider using more commas and shorter sentences.

91. Thank you for your comment. Steve strongly dislikes the high number of commas I generally use, I had purposefully removed many of them. More commas have been added and sentences have been shortened throughout the text.

Reviewer #3 (Remarks to the Author):

Review for “Hydraulic-trait diversity increases forest resistance to water deficits”

This manuscript concludes that hydraulic trait diversity increases forest resistance to water deficits. To support this conclusion, the authors conducted a set of dynamic global vegetation model simulations (aDGVM2-Pi) for catastrophic drought and chronic drought response in tropical forests. To interpret the results from model simulations and investigate trait characteristics, the authors used hierarchical cluster analysis (PCA and HCPC as noted in their methods) for 8 traits.

This is an interesting topic. But I have the following concerns:

First, the results and conclusions are solely based on the model simulations. But the observational evidence is almost absent. There is only one observational data point to compare with the time series simulations for the biomass response to catastrophic drought in Fig 1.

92. We are indeed presenting simulation results from a model. Some model is necessary to make future predictions (e.g. all climate change predictions presented by IPCC). We see process based models, constructed using best plant physiological and ecological principles as the best route to make future predictions and identify mechanisms. To respond to your very excellent constructive comment (comment 119) “the authors should take the advantage of the nature of the mechanistic-driven model in explaining these phenomena” we have conducted additional simulation experiments and identified rooting niche differentiation as one mechanism with the potential to explain these phenomena (Fig. S6). To improve confidence that the model can correctly simulate traits of the study region we have also added further benchmarking (Figs. S7, S8). These

figures compare the simulated mean value of specific leaf area (SLA) to a series of products. These published products (Figs. S7 B-G) were created using statistical models (machine learning models) trained using observed trait data, climate data, and satellite derived data. Our model does not use any observed data to generate predictions, they emerge from simulated dynamics. Our simulation results show remarkably similar mean values to a number of these observation-driven products, in one case there is no statistically significant difference in means (Fig. S7 F, S8 (Moreno et al. predictions)). This additional benchmarking increased our confidence that our trait-based model is indeed capable of reasonable predictions of mean SLA for the study region. Additionally, the model has previously been extensively benchmarked against biomass, tree cover, tree height, and the distribution of forest and savanna biomes across the study area. We apologise that this previous benchmarking wasn't appropriately explicated. We have added a sentence (line 123) to highlight the benchmarking previously carried out for the study area.

Our results complement studies that were primarily based on empirical observations (e.g. Liu et al. 2022, Aguirre-Gutiérrez et al. 2022, Oliveira et al. 2021, Schnabel et al. 2021, Anderegg et al. 2018, Isbell et al. 2015), which are also cited. Our modelling study, however, is novel as the model integrates the main mechanisms at work both for the emergence of functional diversity and how it mediates drought responses. The level of detail in process representation is beyond the level that can be captured in observational studies. We think that this presents an important addition to scientific knowledge. Indeed, such a framework is being explicitly called for to advance trait based ecology (Chacón-Labela et al. 2023). However, it is indeed unfortunate that, given the importance of the topic, there have been so few manipulative drought experiments carried out in mature tropical forests. These are however the gold standard against which vegetation models have been tested. Previous modelling attempts (e.g. Powell et al. 2013) to recreate these biomass reductions due to drought have, in general, failed to reproduce observed biomass reductions. ED2 was the only model which produced any notable biomass loss due to drought when precipitation was reduced by 50%. At the TNF site biomass loss was lower than observed while at the CAX site biomass loss was overestimated (observed = 20%, simulated=50%). We are not aware of any major advancement in modelling biomass loss due to drought since Powell et al. (2013).

We have improved the introduction text to highlight these potential deficiencies in previous modelling attempts, elaborate on potential reasons for this lack of drought response, highlight how our methodology addressed these, and add references to empirical studies which support our findings. E.g. (lines 51-55, 66-69, 200-202, 239-242, 251-262, 276-277, 304-306).

Chacon-Labela, J. et al. How to improve scaling from traits to ecosystem processes. *Trends in Ecology & Evolution* 38, 228–237 (2023).

Incidentally, in which year was the AGB reduction observed? 2005?

93. 2004 at the TNF site and 2009 at the CAX site. We have added an “X” to Fig. 1 A, B to indicate the year in which the AGB reductions were observed at each site. The “X” also indicates the % reduction in AGB observed.

The rest figures are purely based on model predictions.

94. Yes, with ample reference to empirical evidence and theoretical predictions. We have now added additional benchmarking (Figs. S7, S8). The model has been extensively benchmarked for the study area in Langan et al. (2017). In Langan et al. (2017) we compared simulated biomass, tree cover, tree height, savanna and forest biome distribution, to various satellite derived products and vegetation maps. To the best of our knowledge, with respect to previous modelling attempts to reproduce observed distributions of biomass, tree cover, tree height, and biome distributions for this approximate study area, our benchmarking is the best reported in the literature. Benchmarking drought responses added further confidence in our ability to reproduce observations. To further support the model predictions, we have added additional model evaluation. We have added comparisons between community mean specific leaf area (SLA) simulated by aDGVM2 and seven published maps of SLA

Fig. Response Letter SLA-MAP: simulated spatial distribution of specific leaf area (SLA) from aDGVM2 (A) compared to published maps. Statistical/machine learning maps: (B) van Bodegom et al. (year), (C) Butler et al. (year), (D) Boonman et al. (year), (E) Mandani et al. (year), (F) Moreno et al. (year), (G) Schiller et al. (year). Optimisation approach (H) Dong et al. (2023). Data provided as supplement to Dong et al. (2023).

distributions for the region (Fig. Response Letter SLA-MAP, SLA-Density). Our simulated SLA is an emergent property of the model whereby natural selection drives the process of community assembly. Remarkably, comparing our simulated results to these various products reveals that our simulated SLA is closest to that of Moreno et al. (2018) with a mean difference not significantly different from zero (95% CI: -0.07 - 0.28 SLAs) (Fig. Response Letter SLA-Density). The model also very closely matched the Mandani et al. (2018) data with a mean between between -0.76 & -0.44 SLAs. The difference between mean simulated and product data (Bodegom, Boonman, Butler, Schiller) was between -4.55 & 3.89 SLAs, which, given the spread of SLA within these maps (ca. 6 - 45), is not large. Our simulated SLA differed the most with those of Dong et al.'s (2023) optimised SLA with a difference in the means of between -12.07 & -11.50 SLAs. Further comparisons at larger spatial scales will be necessary to elucidate differences between aDGVM2 simulations, site data, and statistical/optimised SLA products/predictions.

We have added the two figures here as Figs. S7 and S8 in the supplement and modified the sentence which read “These simulations revealed the sensitivity of modelled biomass to climate change (Figs. 3, S3).” to “These simulations showed that natural selection driven community assembly within the model produces mean SLA values across the study area which closely match a number of published SLA maps (Figs. S7, S8) and revealed the sensitivity of modelled biomass to climate change (Figs. 3, S9, S10, S11, S13).”.

Figure Resonse Letter SLA-Density: density plot of simulated SLA for aDGVM2 compared to published maps. Inset displays a 95% confidence interval of the difference between simulated SLA and published products.

Even if in Fig 1, the model simulations have multiple replicates, and their trajectories are widespread: from almost no drought response to dramatic drought response.

95. Simulating variability in the response of plant communities is a fundamental part of the model. The important result is the simulated average which is the

standard metric used to present vegetation model results. We have explained this in response to reviewer comment 74 above.

I am not sure whether the mean agreeing with the observation is a coincidence, and the simulations at the site CAX seems to be much worse than the simulations at site TNF.

96. Thank you for your comment. We have replicated all simulations since initial submission. The mean response is essentially identical to the values presented at initial submission. These replicated mean responses also very closely match observations. A single value matching observations could be considered a coincidence, it is however extremely unlikely that the mean response agreeing with the observations, twice, is a coincidence. There is certainly more variability about the mean at the CAX site. The difference between the simulated mean response at the CAX site and the observed response differs by ca. 2%, we fail to see how this is “much worse”. This is an excellent match. This statement is also true for the TNF site. Of all vegetation models tested at this site for drought responses, our simulated mean response is closest to the observed, we have added additional details on previous “best results” in our response 92.

In addition, what’s the accuracy of the simulated aboveground biomass (I am not asking the accuracy of biomass change)?

97. Thank you for your comment. We have addressed simulated biomass vs observed biomass at these sites in the comments to a different reviewer (comment 73). The model has been extensively benchmarked for the study area in Langan et al. (2017). In Langan et al. (2017) we compared simulated biomass, tree cover, tree height, savanna and forest biome distribution, to various satellite derived products and vegetation maps. To the best of our knowledge, with respect to previous modelling attempts to simulate biomass for this approximate study area, our biomass benchmarking is the best reported in the literature. We have included a table above (response no. 73) detailing biomass estimates from field surveys and satellite derived estimates.

We attach below a figure showing simulated mean drought responses as in Fig. 1 A, B. In this figure we have removed simulation results where simulated biomass falls outside of the biomass values at each site detailed in response no. 73. This figure shows that a) the mean simulated biomass reduction agreeing with observed reductions is not coincidence and 2) the result is robust when simulation runs with biomass outside of the bounds for each site are removed.

Figure 2 response: Diversity-drought dynamics - Simulated drought Drought experiments for TNF (A) and CAX (B) where simulations with above ground biomass values outside of observed ranges were

Catastrophic drought should be some intense water shortage that happens suddenly. Not sure why the plants do not respond in the beginning years, but the lagged effects last for years. I am a little bit confused whether Fig 1 shows the short-term effect or the lagged effect of the catastrophic drought. Need to define what the catastrophic drought is.

98. Thank you for your comments. Fig.1 has been adjusted. We now indicate the start and end of the simulated drought treatments. We hope it is now clear that simulated vegetation begins to respond to drought at both sites by the end of year one.

Your comment on sudden drought is correct. For the drought treatment, daily precipitation was reduced by 50% for the duration of the simulated drought experiment. As such it is both intense and sudden. Fig. 1 shows the short-term effect of drought. We hope the improvements to Fig. 1 make this clear.

To improve clarity, each instance of “catastrophic” and “chronic” in the manuscript is now written together with its associated precipitation change, i.e. “sudden catastrophic drought” and “chronic precipitation reductions”. Line 125 now reads “We first use the model to mimic drought experiments run at Tapajos (TNF)²² and Caxiuaana (CAX)²³ national forest sites (Fig. S3) by simulating

catastrophic drought (i.e. a 50% reduction in daily precipitation for between 4-7 years).”

Second, the model is a black box.

99. It is indeed challenging to describe process-based models like this in a short paper. We have extensively revised the manuscript, added many clarifications to the main text, provided a full model description, and added many cross references to the model description throughout the main text to improve the description of the model. The code has been provided for review. Our efforts to describe the model follow the highest scientific standards for such models. While ecological dynamics within aDGVM2 are more complex than in other DGVMs or land surface models, the code base of such models, and climate models, is equally complex. These have nevertheless become standard research tools to project climate change and the effects of climate change on the vegetated land surface.

Even if after reading the method parts, including the “aDGVM2-Pi model description”, I was not sufficiently informed, not to mention to be convinced by the presented results in the main figures.

100. The model description has been extensively revised. We very much hope that any reader is now sufficiently informed to evaluate the model and results.

The model has been extensively benchmarked for this study area in a previous publication. This benchmarking included tree height, biomass, canopy cover, and biome distributions. In this submission, the model reproduced the percentage reduction in biomass at each experimental drought more accurately than any other published model. We have added a plot demonstrating that the percentage reduction in biomass at the drought sites remains unchanged when only simulations where the biomass is within the observed range are used. We have now added comparisons of simulated specific leaf area (SLA) and published maps of SLA (Figs. S7, S8). These comparisons show clearly that our simulated SLA matches a number of these maps well. We have conducted additional simulations where we identify below ground niche differentiation as one potential mechanism with the potential to explain simulated diversity effects (Fig. S6).

The authors did not provide the information about the exact hydraulic mechanisms considered in the DGVM, but only referred all these mysterious to a previous paper.

101. Apologies, this was an oversight. We chose not to provide an extensive model description that was cited and available freely online; however, we realise now that this omission may increase the burden for many readers to understand our methodology. This was a recurring critique across reviewer responses.

The full details of the plant hydraulics implementation and exact mechanisms can be found in the now less than mysterious, sections “Plant Hydraulic Strategies” and “Water transport” in the appendix. In the main text, we briefly summarise these mechanisms as “First, we include in our model advances in disciplinary understanding of plant hydraulics (see “Plant Hydraulic Strategies” in Appendix)^{24,25}. This allows the action of natural selection across an individual’s root, stem, and leaf traits which simultaneously affect hydraulic performance through the soil-plant-atmosphere continuum (Fig. A2). Trade-offs between maximum xylem and leaf conductivity and cavitation²⁵ allow alternative strategies to emerge in the model. Phenological traits, i.e. whether an individual is evergreen or deciduous, determine whether individuals endure or avoid water shortages during the dry season. Below-ground traits determine the amount of carbon allocated to roots and plant rooting depth. These traits affect water availability of individuals in the model and thereby competition for water (Fig. A2).” and provide cross-references to the relevant sections and figures in the appendix.

We have added cross-references to the relevant equations used in the model in the following text “These strategies were: (1) drought tolerant strategies exhibiting very negative watermatric potentials at which 50% of conductivity is lost due to cavitation (P_50). This adaptive trait influences derived traits (Figs. S1, A2) via coded empirical trade-offs (see “Plant Hydraulic Strategies” in Appendix) and results in low specific leaf area (SLA, Eqns. A47-49), high leaf longevity (Fig. A3, Eq. A72), low stem and leaf conductivity (Eqns. A50, A52), and high wood density (Eq. A46), and greater resistance to cavitation (Eqns. A68, A69).”

In fig 1, there is also a need to explain Rao’s quadratic entropy, which might be a standard metric in ecology but not a common metric in the broad readers for Nature Communications.

102. A definition is now provided in Fig. 1’s legend as follows “Rao’s quadratic entropy (“the mean functional dissimilarity between two randomly selected individuals³¹”) was used as a measure of functional diversity in (C) and (D).

Also, is the simulated species richness accurate?

103. Trait based models are not trying to simulate species richness, but rather explore how diversity in functional traits may affect ecosystem function. The “species” concept in our model is arbitrary and serves to: 1) allow reproductive isolation (see “Introduction and modelling concepts” in Appendix) and Scheiter et al. (2013) where we explore the consequences of reproductive isolation on trait distributions in emergent communities in detail. 2) Serve as a convenient method to allow us to add distinct sets of traits to the model in a controlled fashion and examine the effects of functional trait diversity.

In Fig 2, How is the arrow related to the dots?

104. Arrows point in the direction of increasing values for each trait. The small symbols indicate the position of a particular species in trait-space. The large symbols indicate the centres of each cluster in trait space. A symbol being close to the end of an arrow for a particular trait indicates that this species has a high value of that particular trait. Additional text has been added to the legend of a figures 2 and 3, as follows “Arrows point toward: a higher allocation of carbon to grow roots (A-Root), leaves (A-Leaf) and stems (A-Stem) vs deferred growth via higher allocation to storage (A-Store), less negative water potential at 50% loss of conductance (P 50), an evergreen leaf phenology (Eg) (as opposed to deciduous), light-triggered leaf flush (Phenology) (as opposed to water-triggered), increased maximum rooting depth (Root-D).”

Why don't the arrows point to the center of the cluster?

105. Convention, vectors generally start at the origin. We fail to see how arrows pointing towards the centre would be beneficial? We have clarified what the arrows indicate in the above response (no. 104).

L79, “the second axis revealed hydraulic-strategies which closely match observed strategies across Amazonia”, please explicitly show how closely they match.

106. We have explicitly compared our simulated SLA to all known SLA products (response 94). There is insufficient information to explicitly compare the traits of emergent strategies and observed strategies in a quantitative manner. However, we have added additional references to the text which describe plant strategies and axes of trait variation which are qualitatively similar to what is emerging from the model.

While we do not have enough information about the distribution of hydraulic traits/strategies across the Amazon to quantitatively evaluate the all traits, we have now included a comparison of model results to multiple products which estimate/predict SLA for the study area (Figs. S7, S8) based on statistical methods in conjunction with observed data, or optimality based predictions. SLA distributions also reflect phenology (raingreen, evergreen, deciduous) and the phenological type is strongly related to hydraulic traits (Eamus and Prior 2001). Additionally, “Our emergent rooting depth patterns (Fig. S1.4a) are consistent with rooting depth predictions (Kleidon & Heimann, 1998; Schenk & Jackson, 2002) and the areas of forest reliant on water from deep soil layers” as described in Langan et al. (2017).

L83-85: do not understand this sentence. “Here, coordinated...”

107. When traits co-vary in consistent ways this is commonly referred to as coordination. We have added a concise definition and reference to line 142.

L94: What is the hydraulic-strategy niche differentiation? Too vague.

108. We have added additional text (lines 182 - 202) on niche differentiation and conducted an additional experiment to identify mechanisms as per one of your comments.

Why is the drought resilience attributed to the richness of diversity but not the hydraulic traits of certain species, such as isohydricity? At least need to exclude these confounding effects other than diversity.

109. We have conducted an additional experiment to identify potential mechanisms responsible for simulated diversity effects. It was not clear which lines of text you are referring to, however, we have revised the text to make it clear that diversity in hydraulic traits (functional trait diversity) is responsible for simulated diversity effects.

We have not yet devised appropriate experiments to test the individual effects of all traits. We are working on it and have added additional relevant responses to reviewer 1 (response 49).

Isohydricity and anisohydricity are not yet included in the model. We are not aware of any vegetation model which includes competition between multiple

individuals that has successfully included traits which would allow this behaviour in plants to emerge. Future development will add additional traits that will allow individuals to keep stomata open for longer as potential decreases or close them sooner with decreasing potential. The model development associated with the addition of these traits and subsequent testing is beyond the scope of the current manuscript.

Our intent with old Fig. S2, S7 was indeed to show the relationship between drought resistance and the diversity of hydraulic trait strategies. These analyses were clearly confusing, as such we have removed them. To be consistent with our common theme of functional trait diversity we also reanalysed future change simulations to isolate the effect of functional trait diversity (new Fig. 4).

L101: According to the description, the CO₂ effect is only included atmospheric forcing from the MPI model's RCP simulations. The DGVM model (which simulates the land processes) is decoupled with those atmospheric forcings. In reality, there should be land-atmosphere interactions/feedbacks. I am not sure if the DGVM simulation has a changing CO₂ concentration.

110. No published trait-based model has been coupled to an Earth System model. Such coupling with aDGVM2 is very much beyond the scope, and not the aim, of the current manuscript. In the present manuscript changes in vegetation do not feed back to change atmospheric forcings, as such we cannot examine dynamic feedbacks between the land surface and the atmosphere. Our approach is common when modelling climate impact on vegetation, coupled simulations with advanced land surface schemes are rare. Drüke et al. (2023) present the only coupled advanced land surface-climate model study we are aware of. The model presented by Druke et al. is not trait based.

The CO₂ concentrations are indeed changing inline with the respective RCP scenario or held constant at 1990 levels. A figure has been added to clarify (Fig. S2 G). A table has also been added to clarify experimental setup (Tab. S2)

We have now added a definition of the CO₂ fertilisation effect (line 209), described the effects of increasing CO₂ on plant physiology (lines 209-211, eqns. A4, A6, A14, A15 in the full model description added as an appendix), added cross-references to the equations in the model which govern these responses, and justified our approach citing the most up to date review the effect of increasing CO₂ on vegetation. See response to comment no. 124 below for revised text.

Drüke, M., Sakschewski, B., von Bloh, W. *et al.* Fire may prevent future Amazon forest recovery after large-scale deforestation. *Commun Earth Environ* 4, 248 (2023). <https://doi.org/10.1038/s43247-023-00911-5>

If CO₂ is only changing in atmospheric forcing from RCP simulations and the DGVM does not consider CO₂ changing, then the experiment only considers the radiative effect of CO₂ (mostly adverse effect: too hot in tropical regions). Only if the DGVM has a changing CO₂ can it consider the CO₂ physiological effect due to gas exchange (which will affect RuBisCo activity). But the question is, how does the DGVM model parameterize the CO₂ physiological effect on water and carbon cycling?

111. Apologies for this not being clear in the initial submission. The model has changing CO₂ concentrations (Fig. S2 G) and we have now added a full model description as an appendix which details how CO₂ affects plant physiology. We have added additional text to briefly describe these in the main text and added references to the relevant sections of the model description (see previous response 101 and lines 207-212).

Fig 3. The colors and clusters make no sense to me. What's the connection between fig 2 and fig 3?

112. Thank you for your constructive comment. The traits used are the same in Fig. 2 and Fig. 3. Fig. 2 shows clusters in trait-space with the aim of identifying trade off axes and emergent strategies. Commonly future changes in biomass are depicted showing a spatial map (e.g. Fig. S11). The purpose of Fig. 3 was to show how changes in biomass are distributed across trait-space. Fig. 3 (and Figs. S12, S13) shows that climate change induces changes across trait-space that are not evenly distributed. As such it depicts the traits of the winners (areas of increase) and losers (areas of decrease) of climate change.

Can you explain Dim vs PC?

113. It's the same thing, just different labelling conventions in different R packages. Dim has been change to PC in the revised manuscript.

Why does P_50 follow dim2 in fig2 while it follows PC1 in fig 3?

114. For the two sites in Fig. 2, variation in P_50 is less important than at the continental scale where P_50 aligns along the first principal component axis.

Could you also define what re-assembly is?

115. Community assembly deals with how communities are structured with respect to species, their traits, and their relative abundances or biomass within the community. Under steady state conditions the model simulates an emergent

distribution of traits and biomass is apportioned across it, i.e. an assembled community. When conditions change, as is the case for our future change scenarios, communities may or may not reassemble. What we show here is that reassembly (change in the biomass distribution across trait-space, change in number of individuals across trait space) does happen, that it is uneven implying changing fitness as mediated by traits, and that the changes are influenced by the climate change scenario and treatments applied.

L124: CO2 fertilization on what? This is another arbitrary argument.

116. We have now added a definition of the CO2 fertilisation effect, described the effects of increasing CO2 on plant physiology, added cross-references to the equations in the model which govern these responses, and justified our approach citing the most up to date review of CO2 fertilisation effects on vegetation.

The adjusted text is as follows “As atmospheric CO2 increases, the CO2 fertilisation effect, i.e. increased uptake of carbon by plants via photosynthesis with increasing CO2, increases leaf scale photosynthesis and water use efficiency³⁵ via changes to the ratio of photosynthesis to photorespiration (Eqns. A4, A6) and reduced stomatal conductance (Eqns. A14, A15). However, best-evidence suggests that confidence in the magnitude of CO2 fertilisation of growth is low³⁵. Therefore, to explore potential bounds on vegetation responses with respect to CO2 fertilisation, we ran simulations both with and without a CO2 impact on plants.”

L139: “...the biomass differences between full and constrained diversity are influenced by the magnitude of reductions in precipitation and climate change scenario (Figs. 4, S9), and that differences were higher in simulations run without CO2 fertilized plant growth (Figs. 4, S9, Tab. S3).” This is the only explanation for Fig.4. I wonder can we believe these simulations?

117. Fig 4 has been replaced. We have added extensive references and sections of text to support the results presented in this manuscript. We list these in response 71.

That's intrinsic mechanisms?

118. This was a good and valid question. We have now run additional simulations and identified rooting niche differentiation as one mechanism responsible for simulated diversity effects (Fig. S6).

Since this is a modeling study, the authors should take the advantage of the nature of the mechanistic-driven model in explaining these phenomena but should not making hand-waving implications/conclusions.

119. We have responded to this comment in response 118.

In summary, considering the issues I raised above, I do not recommend publishing this manuscript in Nature Communications.

Dear reviewer, in the above responses we have clearly addressed all issues you raised. Kind regards.

-----FIN-----

Dear reviewers,

Thank you for taking the time to review our manuscript and for providing extensive constructive commentary. We have substantially revised the manuscript to address comments.

The main changes are:

In response to comments by reviewer 1 (response 1) we now specify our main aims at the end of the introduction (lines 127-154). In response to comments from reviewer 2 related to the apparent unsystematic nature of our research design (responses 40, 41, 42, 43, 44, 45, 46, 47, 48) we follow their recommendation (response 47) and have (1) added text towards the end of the introduction (lines 127-154) which clearly details the scope of the research questions and highlights the exploratory nature of experimental design, and (2) extensively rewritten the manuscript to structure the summary paragraph, results and discussion, and conclusions sections to mirror the order of the research questions list.

Additional changes include:

1. The analysis for Fig. 1 C, D has been changed in response to a comment by reviewer 4 (response 64) to show uncertainty around presented curves. We have conducted a Bayesian analysis instead of spline regression to address this. This analysis allows us to improve the quantification of diversity effects at the site level.
2. Fig. 2 has been revised to address comments by reviewer 1 (response 35) and reviewer 4 (response 71). We now include additional panels (C, D) which aid interpretation of the emergent trade-off axes and the positioning of plant strategies across each axis.

3. A section heading “Key axes of trait variation mediating diversity effects” has been added and the text substantially revised in response to comments from reviewer 1, to bring the text inline with the additional panels in Fig. 2, and to increase the focus on the axes of trait variation rather than positioning of plant strategies along these.
4. The summary paragraph has been rewritten to generally follow recommendations of reviewer 2, to present aims and results in the same order of our research questions, and to bring it inline with journal specifications.

We believe the revisions conducted in response to your comments have significantly improved the clarity of text and presentation of results in the manuscript. We feel that the novelty and significance of our findings, along with the improved clarity of text and presentation, make this manuscript suitable and accessible for the interdisciplinary readership of the current journal.

Detailed responses to each comment are provided below.

REVIEWER COMMENTS

Reviewer #1 (Remarks to the Author):

Dear authors and Editor,

The manuscript "Hydraulic-trait diversity increases forest resistance to water deficits" had significant improvement. It is an interesting and important effort to include hydraulic traits in models, and it can bring a lot of advances in predicting vegetation response to climate change.

However, there are still points to be clearer and written more directly. I know the model is very complex, and it is a challenge to say things understandably and to be concise at the same time.

Response 1: Dear reviewer, thank you yet again for your constructive and detailed commentary. Below please find detailed responses to each of the points you have raised. We have taken all of your comments into account. We particularly focussed on those regarding clarity and directness. The manuscript text has been extensively revised and reordered to improve clarity.

I miss the aim of the work (the hypothesis to test or the question to answer) to be more emphasised in the introduction. You mention the problems, but not exactly what you want to reach with your work. Is it just a method improvement, or there are ecological questions?

Thank you for your comment. Our main aims were not stated clearly enough in the previous submission. We aim to answer an ecological question: to what extent does functional trait diversity increase tropical forest resistance to drought? Answering this question was one of the main motivations for building the presented model.

At the end of the introduction (lines 127-154) we now specify our main aims as follows:

“Global biodiversity is being lost at an alarming rate. Yet, the extent to which functional trait diversity loss may reduce the resistance of tropical forests to anticipated changes in climate remains unknown. In this simulation study, we aim to fill this knowledge gap by exploring how functional trait diversity mediates tropical forest responses to two types of drought: catastrophic (sudden and severe reductions in precipitation) and chronic (long-term climate change-induced precipitation reductions). The study focuses on tropical forests across the broad Amazonian region and examines responses up to the end of the 21st century.

Given the limited understanding of diversity-mediated resistance of tropical forest ecosystems, the underlying mechanisms, and the axes of trait variation potentially involved, our approach was intentionally exploratory.^{33, 34} Using an iterative experimental framework, we integrate site- and continental-scale analyses, where findings from one scale inform the next. By conducting a series of exploratory sensitivity analyses, we aim to identify key axes of trait variation, uncover mechanisms underlying drought resistance, and assess the magnitude of diversity mediated drought resistance. Our aim with this approach was to answer the following key research questions (RQs):

1. Site-scale analysis

1.1 Does functional trait diversity mediate biomass responses to catastrophic drought at the site scale, and what is the magnitude of this effect?

1.2 What key axes of trait variation mediate diversity effects during catastrophic drought, and what mechanisms are associated with these axes?

2. Continental-scale analyses

2.1 Do critical axes of trait variation at the continental scale align with site-scale axes, and are there indications of future shifts in the dominance of plant strategies?

2.2 Does functional trait diversity mediate tropical forest biomass responses to chronic climate change-induced precipitation reductions at the continental scale, and what is the magnitude of this effect?

These analyses aim to clarify the role of functional trait diversity in enhancing the resistance of tropical forest ecosystems to climate change.”

It is a big challenge to add and represent the diversity of traits and trade-offs that yet are not fully understood. One point that is not clear is how the trait data presented in Table A3 was determined. The authors refer to the Genetic Optimization Algorithm but do not explain it. Maybe I missed something. Strangely, the P50 range is the same for trees and grasses. I wonder what the references were to create this range of values, for all traits.

Response 3: Thank you for your comments. We respond to these individually below:

- The cross-over algorithm used in aDGVM2 is based on the genetic optimisation algorithm DEoptim. This is described in the Appendix in section “A3.10.2” and calculated as in equation A75. A crossref to an explanation of the genetic optimisation algorithm in the appendix (A3.10.2) and the figure depicting this (Fig. A6) has been added to the model description in the supplement (S1.1). The basic premise is that swapping out trait values between seeds creates potentially new combinations of trait values. If these new combinations perform well relative to other plants in a community (i.e. they are fitter) they can increase in relative abundance. This process allows simulated vegetation to continuously explore trait-space (the range of potential trait combinations) which may perform better under current or future conditions.
- P50 same for trees and grasses & references: This was the most parsimonious solution at the time. Parsimonious means here that trait differences between plant types are only implemented if there is very clear evidence that such differences generally apply. With our methodology our aim is that differences in the trait values of trees, grasses, evergreen and deciduous plants, should be an emergent outcome of simulated dynamics.
- The trait values of individuals parameterise themselves as an emergent outcome of the modelled process of community assembly. The min and max ranges were primarily derived from literature sources. Some ranges, e.g. allocation traits, would ideally have min and max values close to zero and one. We found that restricting some of these ranges to be closer to observed ranges increased the speed with which the trait and biomass values of communities stabilised. References for the ranges of all traits have been added to Tab. A3.

So savannas were removed from the model, were the grass traits removed as well?

Response 4: For the site level simulations presented in Fig. 1 only trees were simulated. For the continental scale simulations the standard model initialisation protocol as defined in Langan et al. (2017) was used: “The model was initialized with 50% trees and 50% grasses to avoid making prior assumptions about initial vegetation; following initialization, this ratio changes based on the relative success of these plant types.”. Even in a forest, grasses can occur in the model in the understorey or after disturbance (in the model and in reality). This has now been clarified in the supplement for the site level simulations, S.1 as “For all site level simulations only trees were simulated”, and continental scale simulations, S3.1 as “Simulations were run for 1000 years using the standard model parameterisation. This parameterisation initialises the model with 50% trees and 50% grasses to avoid making prior assumptions about initial vegetation distributions²⁶”

There are many hydraulic traits published for Amazonia so far, including Tapajos and Caixuanã, and some root depth data. I was wondering if it would be possible to change these parameters with some real data.

Response 5: In theory we could run simulations where the model is initialised using observed trait values. Such simulations are beyond the scope of the current manuscript and not the focus of our modelling approach. A main focus of our approach is that trait values should emerge from model dynamics and correspond closely to observed data. Emergent trait values corresponding well with observations increases our confidence in the model specifications and internal dynamics. We have demonstrated this for SLA (Figs. S7, S8). In Langan et al. (2017) we additionally showed emergent rooting depth patterns which were broadly consistent with those of Nepstad et al. (2004). A further issue with designing a model which requires observed data for initialisation is that, in geographical areas where trait data is lacking, which encompasses most of the tropics, the model cannot be reasonably initialised.

However, we have already submitted simulation data for two trait-based model intercomparison projects (MIPs). Both projects use observed site data (for example daily precipitation, trait data). Some models in these MIPs use site derived trait data to initialise their models, often only to define the range. Our simulations do not, but the emergent means from our simulations match observations well. One MIP investigates simulated tropical forests under ambient and elevated CO₂ and compares these to observations (conference proceedings here: <https://doi.org/10.5194/egusphere-egu24-22186>). The second MIP investigates drought responses in trait based vegetation models at additional sites across Amazonia (conference proceedings here:

<https://pure.iiasa.ac.at/id/eprint/19063/>). We are awaiting manuscript preparation by both MIP project leaders in Brazil and the U.S.A. As such, we believe these contributions, once published, will adequately address your suggestion above, just not within the current manuscript.

Another point about the traits aspect that I did not get very well: the strategies categories were directly used in the model and the traits that compose these categories as indirect, right?

Response 5: We are not sure we fully understand what is being asked here. To assist us in the identification of simulated emergent plant strategies we used a cluster analysis (Fig. 2). The units in the cluster analysis were the traits and trait values of individual species. Each cluster displayed different values for each of the traits. These strategies aligned along two axes of variation and were then interpreted in comparison to broad plant strategies observed across the Amazon and globally. Each of the traits presented in Fig. 2 are adaptive traits and the values of traits passed on from an adult tree to offspring individuals can change through time (over generations). For some of the traits discussed in the text these are derived/indirect traits, e.g. P50 defines the value of SLA. However, all of the traits presented in Fig. 2 are direct traits, i.e. these are traits which are passed from parent individuals to seeds.

We have tried to clarify this by adding text to the legend of Fig. 2. The following text was added “. All traits are adaptive, see A3.11, Tab. A3. Additionally, in response to some of your other comments (Response 6, Response 7, Response 8). We now follow your suggestion in Response 8. We no longer perform a cluster analysis in Fig. 3, instead we colour the different leaf phenologies as you suggest. In addition to the PCAs for both sites, we also present the main axes of trait variation (conservative vs productive and hydraulic traits) as separate panels. We believe this makes it more clear that our focus is on the axes of variation while also showing that leaf phenologies differentiate along each axis in a more intuitive way.

But then, there is a part that they focus on root traits. Why not try other traits (i.e. P50)? If you are focusing on hydraulic traits maybe you could use only them. It depends on your question.

Response 6: In our previous response no. 49 we have discussed difficulties posed by conducting simulations where P50 is directly manipulated. “We have not yet devised an appropriate experiment to properly test other traits (above ground traits) as changing, for example, P50, would also change specific leaf area, which would change leaf area index and thus affect photosynthesis and light competition. Likewise, changing P50 would change wood density which

would affect the heights and diameters of stems (Eqns. A23, A24, A50, A51) and thus alter light competition, carbon gains, and sapwood conductivity.”

In this manuscript we test root traits at the site-scale and leaf phenological traits at the continental scale. These traits were associated with what we call the hydraulic axis of trait variation, as was P50. Testing rooting traits at the site-scale was labour intensive and would not have been possible at the continental scale. Similarly, we tested leaf phenological traits at the continental scale as, due to the binary nature of these traits, the number of simulations required was computationally tractable.

Our justification for choosing leaf and root traits for further testing was related to evidence supporting their contribution to niche differentiation, the main mechanism proposed for diversity effects, and computational tractability. However, our continental-scale simulation results show that reducing variability in traits associated with leaf phenology, also reduces variability in P50 and maximum rooting depth. In Langan et al. (2017)(Fig. 3) we have already shown that changing the variability in maximum rooting depth (soil depth) influences the probability of evergreen and deciduous strategies co-existing, the probability of observing a phenological strategy which responds to light conditions, mean P50 values, and trait diversity. Thus, given the emergent covariance structures between traits in the model, we expect it to be really difficult to tease apart the individual contributions of particular traits at the continental scale. However, we expect that manipulation of P50 would also result in diversity reductions in leaf phenology and rooting depth. Given the relative consistency of estimated diversity effects at the site- and continental scale we expect that a systematic manipulation of P50 would also yield similar effect sizes for diversity induced drought resistance.

We have now expanded our justification for our choice of tested traits in the manuscript as follows (lines 265-282):

“Niche differentiation, a well established ecological principal, promotes species co-existence and increases the resilience of more diverse communities to perturbations.⁴⁵ In this study, differentiation of plant strategies across the hydraulic axis of trait variation is the most likely candidate to explain the simulated lower biomass loss in more functionally diverse communities. Catastrophic drought, which causes sudden and severe changes to the water

balance, interacts with hydraulic traits which govern plant water availability. In the model, hydraulic traits mediate plant water balance by influencing temporal and spatial access to water. Specifically, leaf phenological traits regulate the timing of leaf flush or abscission, determining the timing of peak water demand for plant strategies (see A3.9), while variability in these traits between plant strategies can reduce the temporal overlap of their peak water demands. Root traits affect spatial dynamics by determining a plants rooting profile and access to soil water; variability in root traits is thought to be able to facilitate the resistance to water stress⁴². Additionally, P50, which defines xylem resistance to cavitation and transport efficiency, covaries in a consistent manner with both leaf phenological and root traits. These findings address research question 1.2; they point towards leaf phenological and rooting traits as being the primary candidates responsible for simulated drought resistance. Variation in these traits can promote temporal and spatial niche separation of water use, potentially reducing peak ecosystem water stress and associated mortality.”

And (lines 329-332):

“To further investigate mechanisms underlying diversity effects, we focussed on root and leaf phenological traits. These traits were chosen because they: 1) align along the hydraulic-trait axis (Fig. 2), 2) have empirical support for their influence in promoting spatial and temporal niche differentiation, and 3) could be experimentally manipulated at appropriate spatial scales.

Regarding the traits trade-offs and the traits within each group, there are some contrasting data from Tapajos linking higher embolism resistance with lower root depth (Brum et al. 2018). Hydrological niche segregation defines forest structure and drought tolerance strategies in a seasonal Amazon forest - 10.1111/1365-2745.13022. This result seems to contradict the trade-offs represented in the model. Maybe it would be good to show in a diagram how traits co-evolve in the model to make it easy to understand.

Response 6: Thank you for your comments. Based on this comment, some of your previous comments, and comments from reviewer 4, we have now changed Figure 2. We no longer use clustering. Instead, we colour species based on their leaf phenologies (Fig. 2 A, B). In order to display how traits co-evolve we add panels C and D. Panel C displays density plots for TNF and CAX for the conservative vs productive axis (PC1) with colours corresponding to the leaf phenologies in A and B. Panel D displays density plots for TNF and CAX for

the hydraulic trait axis (PC2) with colours corresponding to the leaf phenologies in A and B. Below panels C and D we have now added arrows and a text description of the main trade-offs to better show how traits co-evolve.

Thank you for pointing out the study by Brum et al. (2018), we had not seen this. Oliviera et al. (2021), who use some of the Brum data, point out two important axes of tropical forest variation, i.e. the avoidance-resistance trade-off axis and the slow-safe vs fast-risky axis. The slow-safe vs fast-risky axis is implicit in our model due the empirically defined trade off between xylem resistance to cavitation and xylem efficiency. Our results presented in Fig. 2 and Fig. 3 indicate that the avoidance-resistance trade off axis emerges from our model along the “Hydraulic Trait Axis”. Fig. 2 D now presents this in a way which makes this more clear.

Thus, while the results presented in Brum et al. (2018), Fig. 3a, do appear to contradict ours, we show that this important trade off axis is adequately emerging from our models dynamics. The main difference between our results and that of Brum is that this axis of variation is occupied by strategies which differ to those presented by Brum et al. The plant strategies emerging from our simulations are more in line with those presented by Laughlin et al. (2023) for this climate space. We now discuss this in the text as follows (lines 228-241):

“Brum et al.⁴² also highlight the importance of the avoidance-resistance tradeoff axis in tropical forests. However, for the TNF site, the authors associate different strategies with avoidance and resistance. They found that avoiding strategies had deeper roots, allowing greater access to water, and higher P50, while tolerating strategies had shallower roots, exposing them to more regular water deficits, and more negative P50. However, they acknowledge that the generality of these findings require further studies. In a recent global synthesis, Laughlin et al.⁴³ found no universal trade-offs between rooting depth and xylem vulnerability. While the authors did not identify a universal trade-off, they demonstrated that the probability of occurrence of different combinations of rooting depth and P50 depends on specific abiotic factors such as aridity, precipitation seasonality, and water table depth. Our simulations reflect patterns expected for the climatic conditions of TNF (humid, deep water table, and a prolonged dry period⁴²). Within this climatic context, aDGVM2 captures the two dominant patterns in occurrence probability predicted by Laughlin et al.:

vulnerable strategies with high P50 and shallow roots, and resistant strategies with low P50 and deep roots.

In order to better understand the results presented by Brum et al. we examined their data. In the manuscript, the authors present a statistically significant relationship between P50 and the stable isotope ($\delta^{18}\text{O}$), a proxy for rooting depth, where P50 is predicted to be more negative with lower $\delta^{18}\text{O}$ (shallower soil). Using their data we re-produced their Fig. 3a (see Response Fig. 1 below). The plotted regression coefficients and fit (Response Fig. 1 left panel, red line) differ slightly from the published figures. This is likely due to the average value of $\delta^{18}\text{O}$ we calculate for *Erismia uncinatum* (-4.83) differing to that of Brum et al. (ca. -5.25).

The species *Protium apiculatum* was removed from the analysis (Fig. 3a) of Brum et al. as it was evaluated as an outlier based on Cook's distance. However, the values of $\delta^{18}\text{O}$ for this species are included in other analyses, the methods used to determine P50 and P88 appear appropriate, and P50 and P88 for this species were both used to calculate the safety margin for this species in the analyses presented in Fig. 3 c, d. This suggests that the values of $\delta^{18}\text{O}$, P50, and P88 for this species are not due to errors in calculations/measurement.

We repeated the analysis performed by Brum et al. (2018) but did not remove *Protium apiculatum* from the data. The inclusion of this species resulted in a much poorer model fit for both P50 and P88 (Response Fig. 1). Here, red lines and text show the fit without *Protium apiculatum* while blue lines and text show the fit when *Protium apiculatum* is included. This result makes the generality of the trade-off presented by Brum et al. less clear. Thus, we believe that the further evaluation of the validity of this trade-off will require additional collection of such high quality data across tropical forests. Brum et al. also suggest that further studies are required to clarify the generality of relationships found; we have acknowledged this in the revised text (see above).

The main additional trait associated with resistant, deep rooting strategies in our model is related to leaf phenology and trees which flush their leaves based on light conditions. However, this leaf phenological habit, with leaf flushing happening during the dry season, has been observed at TNF (Weirdt et al., 2012, Restrepo-Coupe, et al., preprint), and has been shown to occur across many Amazonian forests (Lopes et al., 2016). In our simulations, this strategy

generally roots to a greater depth and exhibits more negative p50 values. From an evolutionary perspective this makes sense. Here, deep rooting can improve a plants water status by allowing access to a larger volume of soil while more cavitation resistant xylem improves a plants ability to endure water shortages should they occur.

Laughlin et al., 2023, Rooting depth and xylem vulnerability are independent woody plant traits jointly selected by aridity, seasonality, and water table depth, doi: 10.1111/nph.19276

Lopes, A. P. et al., 2016, Leaf flush drives dry season green-up of the Central Amazon, <https://doi.org/10.1016/j.rse.2016.05.009>

Oliveira, R. S. et al. Linking plant hydraulics and the fast–slow continuum to understand resilience to drought in tropical ecosystems. *New Phytologist* 230, 904–923 (2021).

Restrepo-Coupe et al., preprint, Contrasting Leaf Phenologies at Two Highly Seasonal Tropical Forests, <http://dx.doi.org/10.2139/ssrn.4835444>

Weirdt et al., 2012, Seasonal leaf dynamics for tropical evergreen forests in a process-based global ecosystem model, doi:10.5194/gmd-5-1091-2012

It is difficult to represent traits in functional groups, as traits are multidimensional. Even the deciduous and evergreen groups have been challenging this segregation into functional groups. But I understand that it makes it easier to include in the model. Of course, I know it is impossible to represent all this multifunctionality in models, but it needs to be discussed.

Response 7: Thank you for your comment. We agree. In reality, there are also more or less continuous transitions. However, being deciduous or evergreen has been shown by many studies to be a main trait axis in the tropics because many other traits values are correlated with phenology in the tropics (Eamus & Prior 2001). In the previous submission the weighting between the axes of trait variation identified and the cluster analysis aimed at rough identification of functional groups (plant strategies) was incorrectly focussed more towards the functional groups. We have used the opportunity to revise the manuscript to address this and now focus more on the axes of variation which emerge, rather than the clusters along these axes.

Below we list a number of examples of changes to the text which demonstrate the focus on the emergent axes of trait variation:

- Line 134: “Given the limited understanding of diversity-mediated resistance of tropical forest ecosystems, the underlying mechanisms, and the axes of trait variation potentially involved, our approach was intentionally exploratory.^{33, 34}”
- Line 137: “By conducting a series of exploratory sensitivity analyses, we aim to identify key axes of trait variation, uncover mechanisms underlying drought resistance, and assess the magnitude of diversity mediated drought resistance”
- Research question 1.2 (line 145): “What key axes of trait variation mediate diversity effects during catastrophic drought, and what mechanisms are associated with these axes?”
- Research question 2.1 (line 148): “Do critical axes of trait variation at the continental scale align with site-scale axes, and are there indications of future shifts in the dominance of plant strategies?”
- New section heading for site-scale analyses (Research question 1.2)(line 183) “Key axes of trait variation mediating diversity effects” and text (line 184) “T ▫ investigate potential axes of trait variation and identify

potential trait compositions that impart diverse communities a higher resistance to drought, we performed a principal component analysis (PCA) and cluster analysis.”

- Line 266: “In this study, differentiation of plant strategies across the hydraulic axis of trait variation is the most likely candidate to explain the simulated lower biomass loss in more functionally diverse communities.
- New section heading for continental-scale analyses (Research question 2.1)(line 398): “Critical axes of trait variation and climate change-induced shifts in functional dominance” plus text (lines 403-408).

Eamus, D. and Prior, L., 2001, *Ecophysiology of trees of seasonally dry tropics: Comparisons among phenologies*. *Advances in Ecological Research*, DOI: 10.1016/s0065-2504(01)32012-3

In the cluster analysis, the main traits driving the groups were also included. I wonder if the clustering result would be the same removing the Phenology and the Eg and using them as colours and symbols and seeing how they relate with the traits.

Response 8: The clusters in this figure are secondary to the hydraulic axis of trait variation. We have restructured the manuscript considerably to re-weight our focus on the axes of variation, rather than the clusters (see also Response 7). The aim of the cluster analysis was to highlight that plant strategies (functional groups) could be separated into broad groups. In the revised manuscript we have removed the cluster analysis and have followed your suggestion to colour points based on leaf phenology. We believe the new Fig. 2 is now more intuitive and provides more obvious justification for our choice to manipulate leaf phenology at the continental scale. Additionally, the PCA at the continental scale, Fig. 3, also showed that the hydraulic trait axis emerged, this time as the dominant axis. Based on Fig. 2 and Fig. 3, our follow up simulation experiments at the continental scale were designed to test variability along the hydraulic axis; the cluster analysis was not influential here as the hydraulic axis of trait variation, which included evergreenness/deciduousness, Phenology(light/water), P50, and rooting depth, was clear at both spatial scales. We clarify our focus on the axes of variation, rather than the clusters, in informing subsequent simulation experiments at a number of points in the manuscript, e.g. see Response 7, lines 403-408, 419-420, 423-425.

One point that there was some confusion before were the terminologies used in the text. The points raised before are clearer now, but there are new terms, such as niche. When you talk about niche segregation, **are you referring to the niche segregation caused by different root depths?** It gets confusing because sometimes you say root niche differentiation and others just niche differentiation, and hydraulic-strategy niche. So, there are many terms here with different meanings. It is good to keep consistency.

Response 9: Thank you for your comment. We have now extensively revised the text to clarify which niches we mean. We clarify this in the text at two points as follows:

(lines 265-284) “Niche differentiation, a well established ecological principal, promotes species co-existence and increases the resilience of more diverse communities to perturbations.⁴⁴ In this study, differentiation of plant strategies across the hydraulic axis of trait variation is the most likely candidate to explain the simulated lower biomass loss in more functionally diverse communities. Catastrophic drought, which causes sudden and severe changes to the water balance, interacts with hydraulic traits which govern plant water availability. In the model, hydraulic traits mediate plant water balance by influencing temporal and spatial access to water. Specifically, leaf phenological traits regulate the timing of leaf flush or abscission, determining the timing of peak water demand for plant strategies (see A3.9), while variability in these traits between plant strategies can reduce the temporal overlap of their peak water demands. Root traits affect spatial dynamics by determining a plants rooting profile and access to soil water; variability in root traits is thought to be able to facilitate the resistance to water stress.⁴² Additionally, P50, which defines xylem resistance to cavitation and transport efficiency, covaries in a consistent manner with both leaf phenological and root traits. These findings address research question 1.2; they point towards leaf phenological and rooting traits as being the primary candidates responsible for simulated drought resistance. Variation in these traits can promote temporal and spatial niche separation of water use, potentially reducing peak ecosystem water stress and associated mortality. These results suggest that the further investigation of leaf phenological and rooting traits could provide deeper insights into the mechanisms by which diversity mediates drought resistance.

(lines 299-304) ”However, we found that stability in transpiration increased with functional diversity (Fig. S4) via reductions in the standard deviation. This result links diversity mediated drought responses with stability in transpiration (research question 1.2) and provides further evidence in support of temporal (mediated by leaf phenology) or spatial (mediated by differences in root distributions) niche differentiation in water use as being the main mechanism underpinning simulated diversity effects.

The use of the terms experiment and simulation also confuses the reader. Maybe use only simulations?

Response 10: Thank you for your comment. The editor previously requested that we write simulation experiment. We have gone through the text and checked that “simulation experiment” is always used when we are referring to experiments conducted using the vegetation model. This resulted in two text changes, see below.

Changes:

- Fig. 1 legend
- Fig. 2 legend

In the conclusion, the first paragraph is that Amazon is changing to be a source of carbon, but again, this needs to be aligned with the main goal of the work. The title is related to the result of the hydraulic traits diversity being linked to forest resistance, and then this conclusion seems to be secondary in the text. Thus, I believe you need to decide on what to focus on and the main story to tell.

Response 11: Thank you for your comment. Yes, you are entirely correct. This posed a major dilemma for us in constructing the manuscript as biomass responses and our ability to test whether diversity mediates them are linked. Without a response to catastrophic or chronic drought, testing diversity effects would not have been possible. To align this first paragraph with the main goal of our work, but also highlight the significance of our simulated carbon dynamics, we have revised this first paragraph as follows:

(lines 590-601) “The focus of the present study was to explore interactions between functional trait diversity in tropical forests and their responses to catastrophic and chronic drought. At the site scale our simulations revealed

drought responses in line with observations ^{16, 17} (Fig. 1), while at the continental scale we found that the model predicted that the carbon sequestration functioning of the Amazon was broadly in line with observed and predicted trajectories ^{4, 56}(Fig. S8). These results provided the modeling framework necessary to pursue our main aim. However, they are also noteworthy in their own right as they suggest that improvements in modeling catastrophic drought mortality could also enhance the simulation of chronic drought impacts. While significant progress is being made,⁶³ DGVMs and Earth System models (ESMs), our primary tools for predicting future vegetation carbon dynamics and climate change, still require further refinement to capture physiologically and ecologically mediated future dynamics,⁶⁴ particularly those related to catastrophic and chronic drought.”

Minor comments – Main Text

In the introduction, lines 44 -46, maybe add a sentence with the link between biodiversity with the diversity of functional traits, and resistance, to explain the sentence about the Amazon. This link will help you to introduce the next paragraphs where you talk about traits and ecosystem functioning.

Response 12: Thank you for the suggestion. We have altered this sentence as follows: “However, substantial contradictory evidence from theoretical ecology and experimental ecology⁶⁻¹¹ suggests that high-diversity ecosystems, such as the Amazon, should be amongst the most resistant ecosystems on the planet, with high species richness and functional trait diversity enhancing ecosystem resilience^{12-14??}”

Line 81 - Gleason et al 2016 (Weak trade-off between xylem safety and xylem-specific hydraulic efficiency across the world's woody plant species - 10.1111/nph.13646). Found this trade-off can be very weak.

Response 13: Yes, we are aware of this paper. While the authors found the trade-off to be weak, what they did clearly show is that the deviations from this trade-off axis were towards both lower efficiency and lower safety. No species existed above the trade-off line, i.e. no species were both efficient and safe. As such, this trade off axis appeared to represent a physiological constraint line. These two traits are clearly not consistently linked as there is evidence they can deviate independently of each other, however, Gleason were themselves

uncertain what type of selection regime would result in both low efficiency and low safety. There doesn't appear to be any particular fitness benefit but rather a relaxation of selection strength. In a follow up study by Liu et al. (2021) (Gleason was the second author here) the authors explored how associations between deviations from the trade-off axis and climate variables. Paraphrasing one of the main results, the authors find that in strongly seasonal environments with low amounts of precipitation during the growth season, selection strength for co-optimisation of efficiency and safety is higher (i.e. closer to the trade off axis). Thus these authors expand on the study by Gleason by showing, as is pervasive in ecology, that the validity of the safety-efficiency is context dependent and strongly influenced by climate.

In our model we chose to retain this trade off because: 1) trade offs are essential to allow co-existence and thus examination of diversity effects, 2) while the trade-off appears weak, no other candidate axes which could be applied within a hydraulically enabled vegetation model have since been published, 3) the selection strength that this trade-off generates in the model produces reasonably accurate distributions of SLA, 4) the selection strength that this trade-off generates in the model promotes reasonably accurate simulated distributions of evergreen and deciduous vegetation, and 5) is clearly important for emergent axes of trait variation (Fig. 2 A, B, D), 6) evidence suggests (Liu et al., 2021) that the weak alignment of species traits along the efficiency-safety trade off axis is significantly improved when climatic context is considered.

Liu, H., Ye, Q., Gleason, S.M., He, P. and Yin, D. (2021), Weak tradeoff between xylem hydraulic efficiency and safety: climatic seasonality matters. *New Phytol*, 229: 1440-1452. <https://doi.org/10.1111/nph.16940>

Line 101 – leaf area is not a trait? It is important to make clear what are you calling a trait and what is a phenotype. I think phenotype is the whole combination of traits, and maybe this term is confusing.

Response 14: Thank you for your query. Leaf area is certainly a trait, however, as it is a product of the outcome of the interaction between the adaptive traits (P50 defines SLA, allocation to leaf), the abiotic environment, and competition between individuals, we consider it a phenotypic trait.

Line 149 – remove dramatic?

Response 15: Thank you, this has been done. more dramatic” replaced by “higher”

170-174 – Maybe you don` t need to show the equations here, just mention the details in the appendix.

Response 16: Thank you, we have used the opportunity to remove the equations.

Line 185-186- which niche segregation is you referring to here?

Response 17: This section of text has been extensively revised to address this comment and your previous comment (Response 9). Revised text has been added to Response 9.

Line 185 – I` m not sure what line of evidence you are talking about.

Response 18: This sentence has been removed and the section extensively revised (see text to Response 9).

Lines 210-211 – asynchrony and niche differentiation of what? Maybe better to refer to the trait that was evaluated.

Response 19: Thank you for your comment. We have revised the text and now specifically refer to the traits. The revised text is:

“Our analysis revealed that simulated mean transpiration did not increase with functional diversity (results not shown). However, we found that stability in transpiration increased with functional diversity (Fig. S4) via reductions in the standard deviation. This result links diversity mediated drought responses with stability in transpiration (research question 1.2) and provides further evidence in support of temporal (mediated by leaf phenology) or spatial (mediated by differences in root distributions) niche differentiation in water use as being the main mechanism underpinning simulated diversity effects.

Figure 2 – add what each colour means. 96 species?

Response 20: Thank you. We have now added 96 species to the legend. Colours are now defined by leaf phenology, as per one of your suggestions.

Lines 251-252 - Maybe add here that the response was different across sites - further on you say it was different and in some places it reduced.

Response 21: Thank you for your suggestion, we have done this in the preceding sentence. The sentence now reads: “Across all future change scenarios there was considerable variability in biomass changes across the study area; biomass increases (Fig. S7) are associated with areas where precipitation remained stable or increased (Fig. S2 B, C) while biomass decreases were associated with areas where precipitation was reduced.

Lines 257 – which scenario is this?

Response 22: We have addressed this comment below by following your subsequent suggestion.

258 – Which scenario do you mean? Maybe point to the letter of Figure 3. On the right side, there is no increase in biomass. Maybe on the left side, but with increased CO₂ you can see more increase in biomass.

Response 23: Thank you. The letters are in all the correct places in the subsequent sentences. The second half of this sentence is repetitive. We have now shortened the sentence to: “Biomass changes across trait-space revealed future shifts in the functional dominance of strategies (Fig. 3).

Also, you added productive and conservative here and did not mention it before. Better keep with the same terms.

Response 24: Productive and conservative are first mentioned and discussed on line 167. This change in terminology was performed to address one of your original comments. We have gone through the manuscript to ensure this terminology remains consistent throughout.

Lines 264-265 – is this right? Seem contradictory.

Response 24: If you are referring to the inconsistent terminology, i.e. we wrote acquisitive rather than productive, we have changed acquisitive to productive here to keep terminology consistent. Additionally, the text has been extensively revised and discussion of biomass changes and changes across trait-space have been separated. In response to reviewer 2's "less is more" comment, which we agree with, we have removed the previous Fig. 3 and Fig. S13 from the manuscript entirely. Previous Fig. S12 (Change in the number of individuals across trait space) has replaced Fig. 3 in the main text. This has been done to reduce general complexity, decrease our focus on biomass, but also because the depiction of the change in the number of individuals across trait space provides more intuitive evidence for our follow up simulations where variability in leaf phenology is reduced. The new text for this section (lines 409-418) is:

“Future change simulations revealed shifts in the functional dominance of plant strategies across trait space (Fig. 3). Changes in the frequency of individuals across trait space showed that, while there were marginal differences between simulations where CO₂ varied according to RCP scenarios (Fig. 3 A, C) and simulations where CO₂ was held constant at ambient levels (Fig. 3 B, D), all scenarios showed increases in frequency in areas of trait-space associated with productive strategies and deciduous water triggered strategies (Deciduous_water). Such increases in productive strategies can be associated with both vegetation responses to drought-induced mortality events⁶⁰ and increasing CO₂⁶¹ while increases in the dominance of deciduous species in tropical forests have been documented following long-term precipitation reductions⁶² and drought.⁶³”

Lines 279 – add the peak first and after say it is decreasing.

Response 25: Thank you, this has been done.

Lines 287 – only sites used in - which sites? The two sites or the whole Amazon?

Response 26: Based on the less is more suggestion by reviewer 2, the subsetting analysis and figure has also been removed to simplify the manuscript. Reviewer 2 appeared satisfied with our previous response to their comment and the analysis comparing our entire study area to the work of Brienen et al and Hubau et al remains in the manuscript supplement.

Lines 313-314 – this sentence is confusing

Response 27: This sentence has been revised as follows:

“These results highlight the linkages between above- below-ground traits. Specifically, while variability in below-ground traits and resulting niche differentiation (Fig. S6), along with variability in P50,^{9, 10} can buffer the impact of drought on tropical forests, our findings show that reducing variability in leaf phenological traits causes a reduction in diversity associated with rooting depth and P50.

Lines 316-317 – Isn't this assumed? The strategies are linked with a group of functional traits, if you remove the strategies, you remove that group of traits too.

Response 28: The covariance structure of the traits is an emergent property of simulated dynamics. Given this emergent covariance structure it was likely that removing diversity associated with one trait would influence diversity in other traits, it was not however a priori defined. What is assumed are certain relationships between traits, i.e. a trade-off between hydraulic safety and efficiency or specific leaf area and leaf longevity. What we show is that when diversity in leaf phenology is removed, this causes a reduction in the diversity of P50 and rooting depth. With this we attempt to reinforce again the importance of a holistic view of plant trait strategies. Recent empirical evidence highlights the strong linkages between above and below ground trait-coordination (trait covariance). These simulation results support this view as they clearly show the linkages between above and below ground traits.

Lines 325 - 326 – the highest when the drought was the most intense? (the worst drought scenario)?

Response 29: Yes. Displayed in Old Fig. S16 (currently S13) are differences between different precipitation loss intervals, they aren't really scenarios. Precipitation loss was divided into different bins to examine whether the amount of loss affected the difference between full diversity simulations and those where leaf phenologies were removed. What these results show, which is the

same general response displayed in Fig. 4, is that the difference in biomass between full and low trait diversity increases with increasing drought.

Lines 329 – 334 - A very long sentence and it is confusing.

Response 30: Thank you. We have broken this in to two sentences and improved clarity as follows:

“However, the effects of functional trait diversity on vegetation’s response to precipitation changes could not be fully isolated by this analysis. This limitation was due to differences in initial biomass (Fig. S13), differences in the spatial extent and intensity of future precipitation and temperature changes (Fig. S2 B, C, E, F), and differences in the spatial extent and magnitudes of reductions in functional trait diversity caused by the removal of plant strategies (Fig. S11 B, C).”

Lines 335 – How did you isolate the functional diversity effect? Don’t need to say but add a reference to the methods section.

Response 31: This has been done. The sentence now reads “Therefore, to answer research question 2.2, we statistically isolated the effect of functional trait diversity on the climate change-associated reduction in biomass in 2100 (Fig. 4, see S3.3, Tab. S6).”

Supplementary - Methods

Lines 118 – 119 - On line 118 you say 96 times, and then on line 119 you 96 species. Is it 96 simulations and 96 species? 96 species each one of the 96 times.

Response 32: Thank you for spotting this misplacement of text. The text related to 96 species was supposed to be in the section “aDGVM2 constrained diversity simulations”. The text has now been moved.

Lines 128 and 129 - You mention the most abundant species in the simulations. Was this after the model ran? Were the initial abundances equal? Maybe add info about the initial species dominance/abundance.

Response 33: Yes, this was after the model ran. For each species number treatment, the model was initialised with an approximately even abundance of each species. The following sentence has been added to S2.3 to clarify: “For each set of simulations with different numbers of species, the abundance of each species in the initial population was approximately equal.”

Lines 153 – 155 – does this refer to the Figure 2? I got confused.

Response 34: This refers to functional diversity (Rao’s Q), displayed in Fig. 1 C, D. However, the same traits are presented in Fig. 2. We have now clarified this in the text as: “Relationships between functional diversity and biomass change calculated using these traits are presented in Fig. 1 C, D. Principal component analyses using these traits are presented in Fig. 2.”

Lines 163-165. It is confusing. Is it four or two clusters? Maybe remove the water and light tigger, plus the phenology to check if the groups still will be the same. Looks like a cycling thing. It is also confusing whether you use 3 or 4 clusters.

Response 35: Apologies for the confusion. We follow your advice here and suggestions by reviewer four. We have now removed the cluster analysis from the manuscript. Your suggestion works just as well as the cluster analysis. We now colour the different trait combinations (species) in the principal component analysis (Fig. 2 A, B) based on phenology type. We additionally now provide density plots for each principal component analysis which display how these phenology types are distributed across the conservative vs productive (PC1) and hydraulic trait axes (PC2) of variation.

This change in analysis allowed us to refocus the importance of our results on the emergent axes of variation, rather than the clusters, but also demonstrate how different leaf phenologies are distributed across these axes which provides support for our subsequent manipulation of diversity in leaf phenology at the continental scale.

Was the section on line 310 done for all steps (CAX and Tapajos and the whole of South America)? If so, maybe on methods move it up and say it was done for both.

Response 36: No. It wasn't necessary to do this for the CAX and TNF sites as the species trait pools had already been defined for each site. This is described in sections "aDGVM2 constrained diversity simulations" and "Calculating functional trait diversity from site-level drought simulations". For these sites functional diversity was calculated separately for each.

At the continental scale we wished to calculate functional diversity for the entire area at once. This required the use of clustering to assign individuals from different simulated grid cells to particular areas of trait-space. This approach ensured that the maximum trait diversity was calculated using data from all sites, and that the per site functional diversity could then be interpreted relative to all other sites.

The simulations of experimental diversity were also done for both spatial analyses. Maybe do the same with the methods sections that apply to both spatial scales.

Response 37: We're not certain what is meant here. If you are referring to the ordering of the methods section in the supplement then this won't work as the methods applied at each scale were different (Response 36). If you are suggesting that we do the site-scale diversity experiment at the continental scale, then we should point out that this is not computationally feasible. The methods used at the site level to test the sensitivity of simulated biomass to drought at varying levels of diversity were different to those at the continental scale. The exploration of site-level responses allowed us to refine the sensitivity analyses to conduct at the continental scale. At the site level we identified the suite of hydraulic traits as being important. These traits could then be manipulated at the continental scale. This was an important step which enabled the testing of the effects of functional trait diversity at this larger scale as the site scale sensitivity analysis would not have been possible at the continental scale.

The sensitivity analysis conducted for each site took 1728 separate simulations and took ca. four days to run (on a large computer cluster). There were ca. 2000 sites simulated at the continental scale following removal of savanna sites. Each

one was simulated for RCP4.5 and 8.5 both with and without increasing CO2 (4 levels). Thus, the site-level simulation protocol, applied at the continental scale, would have taken many years to run (approximately 87 years) (4 days x 4 levels x 2000 sites = 32000 days).

Another point, you mention the whole analysis was for South America and you show the results in graphs. But you always just refer to this part as Amazonian. Other forests in South America are not in the Amazonian region. So maybe use the term of the biome tropical rainforests or just analyse the Amazonian region.

Response 38: Thank you for your comment. You are correct, some of the areas of tropical forest included in our study area fall outside the Amazonian region. The main focal study area is centered around the Amazonian region. We would prefer to retain the current formulation used in the text because the Amazon rainforest is globally recognised which, we believe, will allow readers to better identify the broad study area. We would also prefer not to remove the additional tropical forest areas from our study as we believe that our results also provide valuable insights into the potential future responses of these forests. We have added additional text to S1.2 to specify that some of our study areas fall outside the boundaries of the Amazonian region. The text is as follows:

“ Our study region is focused on tropical forests in South America. The majority of the study area is centered on the Amazon rainforest. Some areas of tropical forest included in the analysis fall outside the strict boundaries of the Amazon biome. For simplicity, and because of the global recognition of the Amazon as a key tropical forest, we refer to the study region collectively as Amazonian forests throughout the text.

However, if our choice presents a major barrier to publication we can also re-do all of our analyses to remove the tropical forests that fall outside the boundaries of the Amazonian region.

Reviewer #1 (Remarks on code availability):

I'm sorry but I do not work with models, so I don't think I'm the best person to review the code.

#####

Reviewer #2 (Remarks to the Author):

The authors have done an enormous job revising the manuscript. In the first submission, the manuscript was so unclear that I found the study near impossible to assess. In this iteration, I could better follow what the authors have done, albeit still with considerable efforts.

Response 39: Thank you for the effort you have again taken to review our manuscript and provide constructive feedback.

My main concern is that the research design appears unsystematic, and not clearly motivated and explained.

Response 40: Thank you for your constructive criticism. Clearly, we had not sufficiently explained and motivated our research design. In light of your comments we have significantly revised the manuscript. We hope that the revision clarifies our motivation and the systematic nature of our research design sufficiently to allow reconsideration of your recommendation to reject the manuscript. We detail these revisions below.

The focus of the study appears to be "to evaluate whether diversity can buffer the carbon sequestration functioning of Amazonian forests under warming and drying climate"

Response 41: Yes, this is the main focus of the study. In the abstract we clarify the focus of the study as:

Lines 12-14: "In this simulation study, we fill this knowledge gap by identifying key axes of trait variation and quantifying the extent to which functional trait diversity increases tropical forests' resistance to drought."

We follow your suggestions and now clarify the focus and scope of the study, as well as the justification for our research design in the final paragraphs of the introduction as:

Lines 127-154: "Global biodiversity is being lost at an alarming rate. Yet, the extent to which functional trait diversity loss may reduce the resistance of

tropical forests to anticipated changes in climate remains unknown. In this simulation study, we aim to fill this knowledge gap by exploring how functional trait diversity mediates tropical forest responses to two types of drought: catastrophic (sudden and severe reductions in precipitation) and chronic (long-term climate change-induced precipitation reductions). The study focuses on tropical forests across the broad Amazonian region and examines responses up to the end of the 21st century.

Given the limited understanding of diversity-mediated resistance of tropical forest ecosystems, the underlying mechanisms, and the axes of trait variation potentially involved, our approach was intentionally exploratory.^{29, 30} Using an iterative experimental framework, we integrate site- and continental-scale analyses, where findings from one scale inform the next. By conducting a series of exploratory sensitivity analyses, we aim to identify key axes of trait variation, uncover mechanisms underlying drought resistance, and assess the magnitude of diversity mediated drought resistance. Our aim with this approach was to answer the following key research questions:

1. Site-scale analyses
 - 1.1. Does functional trait diversity mediate biomass responses to catastrophic drought at the site scale, and what is the magnitude of this effect?
 - 1.2. What key axes of trait variation mediate diversity effects during catastrophic drought, and what mechanisms are associated with these axes?
2. Continental-scale analyses
 - 2.1. Do critical axes of trait variation at the continental scale align with site-scale axes, and are there indications of future shifts in the dominance of plant strategies?
 - 2.2. Does functional trait diversity mediate tropical forest biomass responses to chronic climate change-induced precipitation reductions at the continental scale, and what is the magnitude of this effect?

These analyses aim to clarify the role of functional trait diversity in enhancing the resistance of tropical forest ecosystems to climate change.”

The study starts with introducing the 'catastrophic' and the 'chronic' drought experiments, but readers will soon find out that the two cases are in fact not comparable.

Response 42: We now introduced these two types of drought, short term catastrophic drought (a sudden and severe reduction to the water balance) and chronic long-term climate change associated precipitation reductions between lines 129-132.

We previously acknowledged that these are different types of disturbance in our previous response letter. In Response 32 we stated that: "Although sudden severe droughts and long-term gradual reductions in precipitation represent changes to the same resource, they are different types of perturbation which are often referred to as pulse and press perturbations.". If we are misinterpreting your comment regarding comparability, we would appreciate further clarification. However, these two cases represent changes to the same resource, which, we believe, makes them quite readily comparable.

Demonstrating a realistic drought response was a requisite first step to allow the examination of whether functional trait diversity mediates drought responses. Importantly, benchmarking simulated responses to catastrophic drought against observations provided evidence the model was producing a reasonable response, in contrast to previously published literature. Future change scenarios have been run for the Amazon region using many vegetation and Earth System models. We are not aware of any published studies which have demonstrated reductions in biomass during catastrophic drought which are comparable to observations. This pervasive lack of responses to sudden changes in the water balance (catastrophic drought) in these models raises doubts about their ability to realistically simulate the effects of slower water balance changes (chronic climate change-induced precipitation reductions).

Our results show that aDGVM2 simulates a mean amount of biomass loss during catastrophic drought which aligns well with observations. We therefore expect that aDGVM2 may also produce more appropriate simulated responses to future climate change-induced precipitation reductions. In response to your previous comment (Response 72) on the validity and robustness of the model simulations with respect to the net carbon sink of the Amazon having peaked in the 1990s, we have demonstrated that the model produces results broadly in line

with those of Hubau et al. (2020). We believe the previously presented results (Figs. S9, S10) provide sufficient evidence that the model is indeed capturing the response of forests to changes in the water balance which are unlikely to have been captured by previous modelling attempts. Given that your previous query (Old Response 72) was not raised again, we presume you were satisfied with the presented results.

Additionally, in Fig. S7 we display simulated future biomass changes across the study area which correspond well with areas where future precipitation is reduced (Fig. S2 B, C). These results support the model's ability to respond to slower water balance changes, complement its demonstrated response to drought, and provide novel projections of future change across the study region.

We believe we have provided sufficient evidence that demonstrates that a model which responds in an appropriate fashion to catastrophic drought, also responds to chronic drought in a way that has not been captured by previous modelling attempts. Thus, we believe that both types of drought, since they are both changes to the water balance are comparable.

The experiments are for **different sites**, different time periods, and even different ways of manipulating diversity (“initiation with different number of species” in Fig.1; “mean values of the deepest rooting strategy at the CAX site” in Fig S6-catastrophic; “removal of Functional diversity based on Rao’s quadratic entropy” in Fig S14; ”removal of ecological strategies” in Fig S15... etc.). As such, the experimental setup appears ad hoc and disorganised.

Response 43: Apologies for the ad-hoc appearance. Owing to the limited understanding of potential mechanisms and traits underlying diversity mediated drought resistance, our study utilised an exploratory approach. We now clarify this in the introduction (lines 134-136, and Response 41 above).

The experimental setup is iterative, with each step building on prior evidence to address our main aims and bridge spatial scales. We also clarify this in the introduction (lines 136-137, and Response 41 above).

Based on your comment and suggestions in Response 47 we have: 1) added a text at the end of the introduction which clearly details the scope and the research questions (see Response 41), 2) restructured the entire manuscript to

mirror the order of the research question list, 3) highlighted the iterative nature of our experimental design as we move from site- to continental-scale simulations and provided justification for our follow up simulations at each step.

Examples of (3) include:

- Lines 278-280: “These findings address research question 1.2; they point towards leaf phenological and rooting traits as being the primary candidates responsible for simulated drought resistance.
- Lines 282-282: “These results suggest that the further investigation of leaf phenological and rooting traits could provide deeper insights into the mechanisms by which diversity mediates drought resistance.
- Lines 299-304: “ However, we found that stability in transpiration increased with functional diversity (Fig. S4) via reductions in the standard deviation. This result links diversity mediated drought responses with stability in transpiration (research question 1.2) and provides further evidence in support of temporal (mediated by leaf phenology) or spatial (mediated by differences in root distributions) niche differentiation in water use as being the main mechanism underpinning simulated diversity effects.
- Lines 329-339: “To further investigate mechanisms underlying diversity effects, we focussed on root and leaf phenological traits. These traits were chosen because they: 1) align along the hydraulic-trait axis (Fig. 2), 2) have empirical support for their influence in promoting spatial and temporal niche differentiation, and 3) could be experimentally manipulated at appropriate spatial scales. At the site scale we tested the role of root niche differentiation in mediating diversity effects. This scale allowed us to use replicate simulations to identify and re-run cases where all plant strategies were present with relatively even abundances (see S2.7 for details), a setup supported by portfolio and evenness effect theory.⁴⁹ Testing this at the continental scale was not computationally feasible.
- Lines 429-436: “While the site-scale analysis confirmed the role of root trait variability in mediating responses to catastrophic drought (Fig. S6)), it also highlighted the potential importance of variability in leaf phenological traits for regulating the timing of peak water demand among plant strategies. Thus, at the continental scale, we aimed to test whether variability in traits associated with leaf phenology mediate tropical forest responses to chronic drought (research question 2.2). This experimental focus was supported by continental-scale climate change simulations,

which predicted that future conditions would induce changes in leaf phenology and favour water-triggered plant strategies (Fig. 3).

We hope that the above clarification, and the extensive text revisions sufficiently resolve reservations related to our research design.

Regarding Sites: The sites where catastrophic drought was simulated are contained within the larger study region (Fig. S3).

Regarding Time periods: In Tables S1 and S5 the time period for each experiment is detailed. Every simulation experiment was run for 1000 simulation years up to the year which corresponds to 2100. All drought experiments were forced using RCP4.5 with CO₂ increasing as prescribed by this scenario. The additional application of a period of drought at TNF and CAX is the only difference in forcing during the simulation time period between these drought simulations and the simulations run for the entire study area using RCP4.5 with increasing CO₂.

Regarding different ways of manipulating diversity: in the text above we have provided justification for the different ways in which diversity was manipulated. Our experimental design was exploratory. We used the site-scale sensitivity analysis to investigate axes of variation which may underlie diversity effects and point the way forward for continental-scale analyses where such high resolution sensitivity analyses were not possible.

As reader, I wonder why the authors do not simply systematically simulate combinations of different drought types and diversity reduction types? I.e., keeping everything else constant, and systematically vary combinations of drought type and diversity reduction type?

Response 44: Thank you for your response. At the onset it was not clear what these drought types may be, hence the exploratory nature of our experimental design. The presented site-scale analyses assisted in the identification of drought types. We were not certain how traits would covary in the model to form plant strategies, nor which traits might mediate drought responses. The analyses conducted for Fig. 1 C, D, very much akin to real world experiments designed to

investigate diversity effects, were systematic, pointed towards the presence of diversity effects, and allowed the identification of axes of variation and traits which were likely responsible for diversity effects.

Furthermore, it may seem entirely logical to design an experiment where the diversity level is manipulated and everything else is held constant, however, this ideal experiment is not possible in the real world or in the model. It was possible to design the experiment for Fig. S6 in a fashion where only below ground traits were changed and everything else remained constant. However, as mentioned in previous response 49, we have not yet devised an appropriate set of experiments which would ensure everything else stayed constant. For example, changing P50 would also change SLA, this would affect both the individual and the plot level leaf area index and thus gross photosynthesis and light competition. This would also change wood density, which would affect the heights of stems, and thus alter light competition and gross photosynthesis. As evident from Fig. S12, changing diversity additionally changes the amount of biomass stored across the study area. It thus becomes more difficult to compare like with like. The binning based on biomass presented for Fig. S13 was designed to allow better comparison of like with like and clearly displays differences between low and high diversity while only requiring basic summary statistics. While the ideal experimental set-up you have suggested above is currently not possible with this model, the statistical analysis presented for Fig.4 does exactly what you suggest, it allows us to control for all other factors and isolate the effect size of functional trait diversity.

If the authors have a rationale for their seemingly disorganised research design, it's not made clear in the manuscript.

Response 45: Thank you for your feedback, which we greatly appreciate. We believe our poor presentation of the exploratory nature of our study is responsible for its seemingly disorganised nature. We hope our above responses and the significant revisions to the manuscript now make our research design clear.

The research design might also appear particularly disorganised, because of the lack of a good overview. I previously asked if the authors could provide an overview of their experimental setup flow to make their manuscript more reader-friendly. Table S2 seems to be the addition that's supposed to provide

such an overview, but it still unfortunately still fails to provide enough clarity. For example, the key terms 'catastrophic' and 'chronic' used in the main manuscript do not appear in Table S2 at all (nor anywhere else in the Supplementary material, based on word search). I can also not match the types of diversity reduction experiments between the figures and this table. For example, the experiment for Fig S6 that used "mean values of the deepest rooting strategy at the CAX site" does not seem to appear in Table S2, and "the removal of ecological strategies" used in Fig S15 is also not listed in Table S2. Either the terminology is inconsistent or the table is incomplete? Anyway, it makes it very hard for the reviewers and readers to get a good grasp of the different combinations of diversity reductions and droughts that formed the basis for the conclusions.

Response 46: Apologies for unclear formulations and links which have caused unnecessary difficulty. We have revised the main text, we hope this revision will significantly reduce any difficulty a reviewer or reader may have to grasp the different combinations of diversity manipulation and drought which form the basis for the conclusions. Additionally, in the supplement we have attempted to resolve the above mentioned issues as follows:

- The experiment for Fig. S6 has now been added. The previous table has now been broken into two tables to improve clarity. We hope this separation will make it more reader-friendly as it is clearer which table is associated with site-scale catastrophic drought and continental-scale chronic climate change associated precipitation reductions. Table S1 now details site scale setup, reference to Fig. S6 has been added to this table. Table S5 now details the setup for the continental-scale study area.
- For Tables S1 and S5, an additional column has been added (Figure Reference) which contains cross-references for all figures associated with each row.
- In Table S1 under the "Experiment Type" column, "Catastrophic" has been added to "Drought" to improve consistency of terminology.
- In Table S5 under the "Experiment Type" column, "Chronic" has been added to "Climate Change" to again improve consistency of terminology.
- Additionally, "Catastrophic drought" has been added to the S2.1 section heading. Chronic changes are now mentioned in S1.2 of the supplement.
- In Table S5, the third column has been edited to now include a listing for "the removal of ecological strategies". We now refer to these as plant

strategies throughout the manuscript as recommended in another comment.

- The section headings S3.1 and S3.3 have also been altered to provide consistent terminology with the main text and Table S5.

T□ add to the confusion, it's not clear from the start which research questions the paper is addressing. In the beginning, it seems as if the paper is focused on addressing if and to what extent hydraulic trait diversity reduce biomass loss under catastrophic and chronic droughts. But the Results section also address the role of different types of traits somewhere between presenting the results from the catastrophic and chronic droughts experiments, which adds to the messy impression. I would suggest that the authors (1) add a paragraph towards the end of the introduction, that clearly details the scope and the research questions, and (2) structure of the results section so it mirrors the order of the research questions list, and make sure the research questions are correspondingly addressed in the abstract and conclusions section.

Response 47: Thank you again for the time taken to review our manuscript and the highly constructive nature of your commentary. We have followed your suggestions here and significantly revised the manuscript. We have detailed much of this in our above responses. We believe that these revisions have significantly improved the clarity of the manuscript.

T□ conclude, I think a more systematic approach, perhaps even with fewer experiments, might have delivered more convincing insights. **'Less is more' could apply in this case.** However, while I find a much more systematic and structured approach to experimental setup to be necessary, I can not completely rule out the possibility that it's still a matter of clarity in the presentation. But in the current form, I can not recommend publication for the interdisciplinary readership of Nature Communications. I wish I could be more positive, as the topic investigated is both novel and important.

Response 48: Dear reviewer, thank you for your critical and constructive commentary. We also appreciate your acknowledgement of the novelty and importance of the topic addressed.

Your recommendation to reject is unfortunate as we believe our approach was appropriately systematic for an exploratory study. It is however apparent that we did not structure the manuscript appropriately to make this clear. We have followed your advice and conducted a major restructure of the manuscript. We

believe this restructuring, detailed in our above responses, significantly improves the manuscript. We agree that our results are both novel and important. Comments on the validity of the results presented have been addressed. Your major critique, and reason for rejection, was related to clarity of our experimental setup. We believe this has now been addressed. We would be grateful if you could reassess our manuscript and reconsider your decision to reject.

Some specific comments:

Abstract: Where does the 34 % come from? Based on RCP 8.5, 50% precipitation reduction? Can the text in the Results section be more explicit about how the number is calculated? A lot of context is needed to interpret this number, I'm not sure putting it in the abstract outside that context is a good idea, as it might lead to more misunderstanding than clarity.

Response 49: The 34% comes from Fig. 4. It is the maximum difference between high and low functional trait diversity. Yes, Fig. 4 D, RCP8.5, 50% precipitation reduction, and fixed CO₂.

Quantifying the effect of functional trait diversity on resistance to drought is the main aim of this study. Therefore, we would prefer to leave the numbers which demonstrate this in the abstract.

However, the revised manuscript has been extensively restructured, as per your recommendation (Response 47). The abstract has also been restructured to account for this and to conform with journal specifications (<200 words and no references). In the abstract we now are now more explicit about where these numbers are derived from and provide the range for site- and continental-scale diversity effects (lines 17-19). This section of the abstract now reads:

“We show that higher functional trait diversity reduces site-scale biomass loss during sudden catastrophic drought by 17-32% and continental-scale biomass loss due to chronic climate change-associated precipitation reductions by 10-34%.

L49-55: The authors suggest that the lack of hydraulic traits representation in models as a main reason for the underestimation of diebacks in model simulations, but ref. 8 does not actually say much about possible reasons for low

dieback rates in models at all, and various other reasons are given the other refs?
Perhaps consider reformulating the sentence(s)?

Response 50: It was not our intention to suggest the lack of hydraulic traits in models as the main reason for the underestimation of diebacks.

The text below is reformulated to indicate a separation between the lack of drought responses, and the lack of hydraulic traits.

“In contrast, modelling studies commonly simulate drought induced diebacks which are lower than observed diebacks.^{19, 20} A **further** problem with this previous work is that the models used did not consider how hydraulic traits can mediate plant water status and how diversity in these traits may influence how forests respond to drought.⁸ Therefore, these analyses could not evaluate whether hypotheses from theoretical and experimental ecology, that diversity increases the resistance of ecosystems, apply to hyper-diverse tropical forests.”

L217: Not sure how Fig. S7 supports the statement. Wrong cross-ref?

Response 51: Apologies, this was supposed to reference Fig. S6. We now use the built in cross-ref capabilities of LaTeX to avoid such errors.

L374: “conservation of functional trait diversity” – what does that mean in a practical sense? Elaborate?

Response 52: Thank you for your comment. We have expanded the text to incorporate it. The text now reads:

“Thus, our results quantify the potential benefits of holistic conservation approaches that incorporate the conservation of functional trait diversity,⁷² particularly traits that mediate plant water status.¹⁰ The conservation of diversity in plant functional traits can ensure continued resistance of tropical forests to drought, increase long-term carbon storage, and help mitigate further exacerbation of climate change.”

Conclusions/Discussion section: Discuss limitations and future research directions more?

Response 53: As per journal guidelines (<https://www.nature.com/ncomms/submit/article>) we have aimed at keeping the Conclusions/Discussion section succinct. We have however added additional text throughout the manuscript, and some in the Conclusions/Discussion section, which highlights limitations and points towards future directions.

Additional text:

- Lines 242-243: “We expect that globally, covariance between rooting depth, P50, and leaf phenology, will change based on the specific abiotic context, as found by Laughlin.⁴⁶”
- Lines 282-284: “These results suggest that the further investigation of leaf phenological and rooting traits could provide deeper insights into the mechanisms by which diversity mediates drought resistance.”
- Lines 393-395: “Taken together these results: 1) provide evidence that improvements in the sensitivity of vegetation to catastrophic drought may represent a path toward improved modelling of future carbon dynamics in both DGVMs and Earth System models”
- Lines 595-601: “However, they are also noteworthy in their own right as they suggest that improvements in modeling catastrophic drought mortality could also enhance the simulation of chronic drought impacts. While significant progress is being made⁶⁵ DGVMs and Earth System models (ESMs), our primary tools for predicting future vegetation carbon dynamics and climate change, still require further refinement to capture physiologically and ecologically mediated future dynamics⁶⁶, particularly those related to catastrophic and chronic drought.”
- Lines 625-633: “Thus, our results highlight the potential value of an improved integration of the biosphere in Earth System Models (ESMs). Historically in ESMs, the focus on improving the representation of ecological processes has lagged significantly behind advancements in geochemical processes. Our results, demonstrate a substantial effect size of functional trait diversity which is comparable to the effect of nutrient dynamics on vegetation productivity and carbon storage.⁷¹ Thus, our findings strongly support recent calls to redress this imbalance and prioritise ecological realism in ESMs;⁶⁶ in the face of climate change, such improvements are essential to better understand and predict the feedbacks and interactions between the biosphere and atmosphere.”

Fig. 3: explain the unit 'tonnes per hexagon'.

Response 54: We agree with your less is more statement. We have replaced the old Fig. 3 which showed biomass changes with the figure from the supplement which displayed changes to the number of individuals across trait space (Old Fig. S12). This was done as the identification of frequency changes in trait space is more inline with our research questions. Old Fig. 3 and the supplemental figure (Old Fig. S13) have been removed entirely.

The legend of Fig. 3 has been changed to explain hexagons:

“and each climate change scenario using all trees in the study region. Trait space was divided into hexagons. Colours display an empirical 2D kernel density estimate of the average biomass change per hexagon between 2000 and 2100.”

Fig. 4: are the precipitation reductions occurring on top of precipitation reductions predicted for the different RCP scenarios 1990-2100? (L341 says 2500 mm/year is mean for the area – but for which period of time?)

Response 55: Apologies for the confusion. No additional precipitation reductions are applied on top of the reductions associated with the different RCP scenarios. The lines in Fig. 4 are the predicted mean responses derived from a linear mixed effects model. The approximate mean for the area (L341) is the mean for the study area (Fig. S2 A) for the period 1961-1990. The precipitation reductions are those displayed in Fig. S2 B, □ which correspond to RCP4.5 and 8.5.

Additional clarification has been added to the legend of Fig. 4, cross references have been added to Fig. S2 and the text in section S3.3.

The additional text in the legend in Fig. 4 is:

“Predictions are based on a linear mixed effects model (Tab. S6). Lines are predicted mean responses for the effect of functional diversity on change in above ground biomass at varying levels of precipitation reduction. Shading indicates a 95% confidence interval. The range of values for precipitation reductions was based on the approximate mean reduction for the study area of 30% (Fig. S2 B, C) up to a value of 50% consistent with site-level drought experiments. See S3.3 and Tab. S6 for further details.”

The additional text in section S3.3 is:

“To present plausible predictions of the differences between high and low functional diversity (Fig. 4) we used approximate mean values for annual precipitation in the 1961-1990 time period (2500mm, mean in data 2456mm, Fig. S2 A) and precipitation reductions in 2100 (Fig. S2 B, C) close to the mean reduction in the data (750mm, mean in data 713mm).”

We hope the additional text and cross references make it clearer that the results presented in Fig. 4 are predictions from a linear mixed effects model which was constructed using DGVM simulation output. The aim of this analysis was to isolate the effect site of functional trait diversity at varying levels of precipitation reduction.

Input data files (Dropbox link) returned error message (files not available).

Response 56: Apologies, it is not clear how this error occurred.

Please find a new link below. It has been tested on a different browser and the ZIP file is downloadable without requiring any dropbox login info.

<https://www.dropbox.com/scl/fi/apgngoxzk18mfi3ngxown/InputDataNattComms.zip?rlkey=yu0uoh9a5vgxs31dbahr9hl8x&st=phum41m0&dl=0>

Supplementary Material – Cross-refs only refer to heading names, which is fine in a digital format using the search function, but hard to navigate in a printed format. Maybe consider adding heading numbers and/or page number?

Response 57: Nature guidelines on latex documents specify that heading numbers should not be included. We presume this mainly applies to the main text. The final page numbers may also be unclear. We have now added heading numbers to the Supplement and Appendix to aid navigation. Throughout the manuscript, supplement, and appendix, the cross-refs now use these heading numbers.

Fig. S6: set to the mean values of the deepest rooting strategy at the CAX site (egwt, Fig. S6). Does it refer to Fig. S5? Anyway, not entirely clear what is meant by "the mean values of the deepest rooting strategy". Do I understand it right that S5 A and B show the distribution 'before' removing root niche differentiation. For clarity, would it be possible to simply make a sub-plot of traits after removing root niche differentiation?

Response 58 A: "Do I understand it right that S5 A and B show the distribution 'before' removing root niche differentiation."

No, S5 A and B show the distribution of rooting depths for all 96 species used to initialise diversity experiments shown in Fig. 1 for the TNF and CAX sites. Colours indicate the densities of rooting depth for the different plant strategies. This is stated in S2.6 as "Plotted the maximum rooting depth for each plant strategy (Fig. S5) using the species-trait-pool to examine rooting niches of strategies.". Our intention with Fig. S5 was to display that there was diversity in rooting depths and that the different plant strategies had different mean rooting depths and density distributions.

Response 58 B: "For clarity, would it be possible to simply make a sub-plot of the traits after removing root niche differentiation?"

This can be done for the replicates used for this experiment if required. However, we very much agree with your comment (Response 48) that "less is more". The addition of further plots does not appear in line with this.

To clarify, the legend text of Fig. S5 now reads:

“Root niche differentiation: Density plot of the distribution of rooting depths for plant strategies at Tapajos National Forest site (TNF) (A) and Caxiuana National Forest site (CAX) (B). Plant strategies are, deciduous trees with a leaf phenological trigger which responds to changes in light (Deciduous Light), deciduous trees with a leaf phenological trigger which responds to changes in soil water (Deciduous Water), evergreen trees with a phenological trigger which responds to changes in light (Evergreen Light), and evergreen trees with a leaf phenological trigger which responds to changes in soil water (Evergreen Water). Densities were scaled to a maximum value of one. Colours correspond to the colours of plant strategies in Fig. 2 with densities calculated for each site and plant strategy using the same 96 species.”

Fig. S15: The term ‘ecological strategies’ is not explained in the caption nor in the main manuscript. The term is used only once more in the caption of Fig. 2 (based on word search in the document), but does not provide an explanation for what was done in Fig S15. Elsewhere, the term ‘plant strategy’ is also used. For clarity, maybe stick to one of the two terms if they refer to the same thing (and make sure it is clearly defined).

Response 59: Thank you for flagging the inconsistent terminology. Ecological strategy was previously defined in the supplement between lines S157-161 following Westoby et al. (2002) and between lines S169-171.

To improve consistency, all instances of “ecological strategy” have been changed to “plant strategy”

Westoby et al. (2002) Plant Ecological Strategies: Some Leading Dimensions of Variation Between Species
<https://doi.org/10.1146/annurev.ecolsys.33.010802.150452>

Reviewer #2 (Remarks on code availability):

Dropbox link to input data files returned error message (files not available).

Response 60: Apologies, see response 56.

#####

Reviewer #4 (Remarks to the Author):

General comments:

1. Coming into the middle of the review process, I can say the revised manuscript **reads clearly**. I am not experiencing the confusion that the original reviewers encountered. I feel the issues that led Reviewer 2 to suggest rejection have been addressed in the thorough revision of this manuscript.

Response 61: Thank you very much for your positive feedback. Your feedback on the clarity of the manuscript is also encouraging.

2. The original reviewers also made requests to address issues of model validation and skill. These are crucial, but it is also apparent that such opportunities are limited. The authors' reference to their previous work (ref 26), which mainly included tropical savannah, are appreciated. But this appears to be next-generation ecological modeling work. So, the additional careful documentation of the model structure and theory, governing equations, experimental set-up and permutations explored, all are vital and appreciated for increasing confidence in the model and conclusions. After all, the standard for all peer-reviewed scientific publications is reproducibility, and it appears the original manuscript fell well short of providing adequate detail for any other research to duplicate these simulations and analysis. The current version is much better, and the changes make the paper more relevant as well as robust.

3. This is a model sensitivity study. I think if it were clearly presented as such (in the introduction and the conclusions) it would ameliorate some of the criticism and concern regarding the limited validation of the model. Combined with the careful description of how the traits and their selection are grounded in reasonable theory, it would strengthen the position of the paper.

Response 62: Thank you for your comments. We now state that this is a sensitivity study on line 131 as well as highlighting the exploratory nature of our research design between lines (127-141).

We have now added references for all trait ranges in Tab. S4 and provided a figure displaying the maximum variation in root profile shapes (Fig. A2). We have also substantially revised the manuscript to improve its general readability.

Specific comments (line numbers refer to the mark-up version of the manuscript):

L 114: The text states the model has a fixed probability each year for seed production. I am not familiar with tropical species, but in midlatitudes many tree species have very uneven seed production, with periodic “mast years of very high production as an enhanced survival strategy. Is mast seeding also a tropical plant trait, and if so, would that additional parameter affect the emergent properties of trait-based adaptation as well?

Response 63: Apologies, this is a misunderstanding caused by the poorly structured sentence. Seed production happens every year, in every gridcell. The number of seeds produced by any plant can vary based on changing abiotic and biotic conditions which affect total carbon gain. The fixed probability is in relation to the exchange of trait values.

The sentence has been changed to: “Seed production occurs annually whereby parent individuals belonging to a particular species (species is a generic label applied to individuals at initialisation to allow reproductive isolation between groups) exchange trait values with a fixed probability (see “Reproduction, inheritance, mutation, and cross-over,” Fig. A5 in Appendix).

Fig 1 C-D: Could you also show spread/uncertainty for these curves as well?

Response 64: This has been done and thank you for the suggestion. Adding uncertainty required a change of analysis. Adding spread/uncertainty to a spline regression was cumbersome. We tried to model these relationships using ordinary least square regression, but the assumption of homoskedastic residuals was violated. We now model the effect of functional diversity on both the mean response of biomass changes, as well as its variance, using a Bayesian methodology. Fig 1 C-D now both include uncertainty as the 95% credible intervals for the mean posterior effect of functional trait diversity (see also new tables S1 and S2). We have replaced the old Fig. 1 D (species richness vs functional trait diversity) with the effect of functional on residual standard deviation. We feel that this result, that functional trait diversity also reduces variability in simulated responses, is more relevant to the present study and is inline with empirical effects of diversity. We have added additional text and references to the main text to contextualise these findings.

The additional text in the Fig. 1 legend is as follows: “(C) and (D) show the results of a Bayesian linear regression analysis (see S2.4 and Tabs. S2, S3). Shading indicates the 95% credible interval around mean posterior predictions.”

Fig 2 and associated text: How do these principal components relate to commonly used (and more widely understood) metrics like WUE and LUE?

Such metrics also convolve the species traits you have listed in fairly straightforward ways.

Response 65: Dear Reviewer, we agree that many biogeochemists and modellers may be more familiar with metrics like WUE and LUE. However, we believe that the very ecological focussed terminology currently used will be understood by a much broader audience while also being accessible to biogeochemists and modellers.

Calculating WUE and LUE and relating them to species traits and trait diversity is an excellent idea. This is beyond the scope of the current manuscript but we will certainly investigate in follow up studies.

Eq S2: Why did you not use CDF matching to translate anomalies? It is non-parametric, thus more robust than MSE minimization.

Response 66: We agree that CDF might also be appropriate to derive anomalies. However, we carefully checked the fitted anomalies from the MSE minimization and found that it represents the trends in the anomalies with high agreement. We prefer not to modify our approach because this would require re-doing all model simulations and analyses. We believe that changes in how we derive anomalies would not substantially change our results, the effect size of functional diversity on biomass resistance to water balance changes, as these are quite consistent from the site to continental scale.

Some references to figures in the supplement are misnumbered - both in new text (e.g., L 86) and in the original text (e.g., L 241, 243), which apparently wasn't checked for changes to the figures. Please carefully check that correct figure numbers and panels are referenced throughout the revised manuscript.

Response 67: Thank you for pointing out these inconsistencies. We now reference each figure using LaTeX functionality so figure numbers will be updated automatically.

Supplement L 325: change “used use to “used”.

Response 68: This has been done.

Fig S4: Please clarify - is what's being called “stability of transpiration” here and elsewhere actually calculated solely from precipitation (according to the caption)? That would presume no moisture stress, which seems contrary to the premise of the paper being focussed on drought response.

Response 69: Thank you for your query. Apologies, this was a mistake. The two instances of precipitation in the figure label have been replaced by transpiration. The calculations presented are solely based on transpiration, not precipitation.

Eq A41: I am not familiar with this root distribution function. Please provide a reference for its origin.

Response 70: This root distribution function was not derived from the literature. We developed it such that it fulfils our requirements. Specifically, the function should be flexible enough to describe shallow and deep roots with a high proportion of root biomass in upper or deep soil layers. We have added a figure to the appendix (Fig. A2) which displays the maximum potential variation of plant root distributions in the model.

Fig A4: This figure does a particularly nice job of summarizing the phase-space portrayed in Figs 2 and 3 in the main body of the manuscript, and is an aid in interpreting those figures. I suggest mapping into this figure the specific trait abbreviations listed in Figs 2 & 3, and moving this figure into the main text - either as an additional panel of Fig 2, or as Fig 2 itself and then combining current Figs 2 and 3 into one figure. For the broader journal readership, it would be very valuable to have this in the main text.

Response 71: Thank you for your suggestion. Fig. A4 is only about carbon allocation to the various plant compartments. Adding this figure would assist interpretation of the axis of variation associated with productive vs conservative strategies, but not with the axis of variation associated with hydraulic traits. Figure A2 would assist interpretation of the hydraulic axis. We have tried to combine these two figures in various ways with the existing Fig. 2. We have chosen not to add these additional figures to the main text as no combination of plots produced a combined figure which improved interpretation.

To aid interpretation we now add cross references in the main text to this figure (line 194). To improve this figure in a way similar to what you have suggested above, and based on suggestions by reviewer 1, we have added panels C and D. We believe these panels now provide a more intuitive summary of phase-space (trade off axes) presented in Fig. 1 A, B, which assists interpretation of our results for the broad readership of this journal.

Reviewer 2 asked about coupled land-atmosphere modeling with such a trait-based model. Inclusion of a trait-based model in a DVM would be the “6th generation” of land models the “5th generation” being those including vegetation demographics (e.g., FATES). This is an exciting prospect - would such a scheme be ready for CMIP7, or will be waiting longer to see coupled

climate simulations with trait-based vegetation models? Just asking for your speculation/opinion here, but you could dangle the potential for trait-based modules in Earth system models as a closing remark in the Conclusions.

Response 72: We have no idea how long it will take. We could imagine that such a scheme would not be ready for CMIP7, it would be great if it was.

We have followed your suggestion and dangled the potential for trait-based models (but also any model which responds to drought) in Earth System models between lines 587-593:

“These results provided the modeling framework necessary to pursue our main aim. However, they are also noteworthy in their own right as they suggest that improvements in modeling catastrophic drought mortality could also enhance the simulation of chronic drought impacts. While significant progress is being made,⁶⁵ DGVMs and Earth System models (ESMs), our primary tools for predicting future vegetation carbon dynamics and climate change, still require further refinement to capture physiologically and ecologically mediated future dynamics,⁶⁶ particularly those related to catastrophic and chronic drought.”

Reviewer #4 (Remarks on code availability):

No code posted online. See the authors' code availability statement.

Response 73: Dear Reviewer,

In the code availability statement we write that the model code has been made available for review and reproducibility purposes. The code was uploaded as one of the manuscript items for both the initial submission and the revision (with the described changes made to residual content). We can see and download the code for both submissions online. It's not clear from Reviewer one and Reviewer two's responses whether they were able to view this. If you were not able to view this code this suggests a potential issue with visibility of uploaded items on the website.

Nature's code publication guidelines apply to previously unreported custom code. The model has already gone through review and been published (e.g. Langan et al., 2017). As such, the current manuscript does not use previously unreported custom computer code. However, we support open science and will publish the code (with restriction for one year) upon publication. However, publication of unsupported code to Zenodo (or similar) would likely represent a major barrier to use for end users, we are thus currently working on a GIT based solution and user agreement similar to that of the Max Plank Institute for Biogeochemistry (<https://www.bgc-jena.mpg.de/en/bsi/projects/quincy/software>).

This would adhere to their “fair use” principles, offer a level of user support, and allow users to remain up to date with future releases. We hope to have this workflow ready upon publication.

Data for figures are posted online.

Dear Reviewers,

We would like to express our gratitude for your assistance and constructive commentary.

All comments have been addressed. Individual responses are below.

REVIEWER COMMENTS

Reviewer #1 (Remarks to the Author):

Dear authors,

The manuscript ‘Amazon forest resistance to drought is increased by diversity in hydraulic traits’ is an important and novel work that can bring new insights in climate change impacts on vegetation. Congratulations for all the changes made in the manuscript. It has significantly improved since the first version.

I liked the new analysis, the new figures, and all the re-writing. It now reads a lot easier than previous versions, and it is clear to understand what was done and their results. With all these changes I think I had a much better understanding of the manuscript.

I’m happy with the responses and changes, and although it had a significant improvement, I still have a few last points that the manuscript would benefit before publication.

I would like to state that the points below are from my understanding of the paper, and what I think could be done to give a last improvement to it. I also understand, if for the authors some of these suggestions does not make sense.

Best wishes

Response 1: Dear Reviewer,

I greatly appreciate the effort taken to provide constructive feedback on this and earlier versions of the manuscript. Your assistance has significantly improved the manuscript.

Your specific comments/suggestions are addressed below.

Kind regards,

Liam Langan (on behalf of all authors)

Comments/Suggestions

1. I liked the way you exposed the goals and hypothesis, but I think when you write about the two types of droughts, it is important to make the link with the scale of the effect as well (maybe after you write the hypothesis).

Something like, to test the different drought effects we had to focus on different scale approaches, where the catastrophic drought was focused on local sites and the chronic drought

effect, also related to the changes in climate (long term changes), was tested on the whole forest scale (see Methods SI for details).

Maybe then with this reference you can add an explanation somewhere in the SI why you change the scale when you change the drought type (something like your response 37 to reviewer #2).

I guess this would also address some of the points that the reviewer #2 made, about the changing in methods along the manuscript. It does get confusing as the methods and your approaches change while you describe your results.

I understand it is because the nature of the investigative study, but it also needs to be clear for the reader.

Response 2: Thank you for your comments and suggestions. We had overlooked linking the two types of drought at this point in the manuscript.

Following your suggestions we have added the following text at the point you suggest (lines 128-133)

“Site-scale catastrophic-drought simulations allow comparisons with empirical observations. Demonstrating realistic catastrophic drought responses is requisite for the examination of functional trait diversity-mediated responses and supports the investigation of continental-scale chronic drought responses. Site-scale diversity manipulations inform continental-scale manipulations essential for assessing the broader implications for tropical forests across the Amazonian region to the end of the 21st century.”

2. About the questions, I think you are mainly looking at 2 questions, but at different perspectives (drought type).

So, you mainly want to understand:

- How functional diversity mediates biomass in both drought type/scale? (when I say How I think it already includes the magnitude of the effect).

- What are the key functional traits that mediate these responses, in drought type/scale?

I think if you do something like that it is already clear that you will have two sections and that you will investigate the same questions in each drought type/scale analysis.

Response 3: Thank you for your suggestion. While we appreciate how your suggestion would streamline the presentation of our questions, we feel that the current order of questions better represents the order in which we performed our simulations as we moved from the site- to the continental-scale. Leaving the questions unchanged also fits well with the additional text we have added in response to your previous comment.

3. For the results, I don't think it is necessary to be pointing out the hypothesis every time within brackets.

Response 4: Thank you. We agree. The RQs are also in the heading sections. We have removed all RQs in brackets but retained the instances where the questions are being directly referred to in running text.

I suggest splitting the sections in drought types and remember the reader the scale it was done, but the main points cited in your goals were to test the drought types, and not the scale. From what I understood the scale change was more a method reason.

Response 5: Ultimately our goal was to identify whether diversity affected responses to chronic-drought (climate change) at the continental scale. We feel that demonstrating continental-scale effects is important as it: 1) demonstrates that site-scale effects are relevant at larger spatial scales, this was not necessarily the case; 2) provides results at a scale appropriate to highlight their relevance for Earth System processes.

The site-scale analyses were necessary to allow us to: 1) demonstrate realistic drought responses, 2) initially assess whether we would find any evidence of diversity effects, 3) identify axes of variation and traits potentially responsible for diversity effects which could then be manipulated at the continental-scale.

Example:

Section 1 could be something like: **Catastrophic drought simulations, a site scale analysis**
Then, you could divide the section in two sub-sections (Biomass change mediated by trait diversity and the Key axes of trait variation mediating plant response).

Section 2 could be something like: **Chronic drought and climate change simulations: Amazon analysis.**

Response 6: We would prefer to leave the sections as are for the following reasons.

Our sections are already divided as you suggest, however, at each scale, there is an initial comparison of responses to drought which is not explicitly included in our research questions. At the site-scale the first comparison is with empirical drought responses. At the continental-scale the first comparison is with the analyses and predictions of Brienen and Hubau. Thus sections were divided as follows, which strongly resembles your suggestions. We have adopted your suggested section headings above and changed subsection headings (and RQ headings) as follows:

1. Catastrophic drought: site-scale analyses
 - a. Simulated vs observed biomass responses
 - b. RQ1.1
 - c. RQ1.2
2. Chronic drought and climate-change simulations: continental-scale analyses
 - a. Future trajectories of biomass change
 - b. RQ2.1
 - c. RQ2.2

Then, again, you could divide it in two sub-sections. Here, can you change the order of the results and show figure 4 before figure 3? So, the section about Diversity effect on biomass would come first, and then, you can detail the key traits that have impacted more this change in plant dominance. It will make more sense in the logical order.

Response 7: Thank you for your helpful suggestions. We follow your previous suggestions (response 6) more closely as we feel this better fits the structure of the manuscript and more closely matches the actual order in which analyses were performed.

4. This manuscript is quite dense of information, and I wonder if there are parts that does not reflect to the main results could go to the Supplementary making the manuscript more objective.

Response 8: We could not identify any large sections which could be moved to the supplementary information. Some lines of text discussing the differences in our results, those of Brum et al., and those of Laughlin have been removed altogether as they were unnecessary. One of the final synthesis discussion paragraphs has been removed as it was discussed in sufficient detail in previous sections and not the main focus of the manuscript.

5. Also, I have the impression that the conclusion is very long and has lots of information, looking like more to a discussion. My suggestion is to see what information is new and add to the discussion part/or SI and just leave as conclusion the most important message for the reader.

Response 9: Thank you for your comment. You are correct. Also, Nature Communications does not allow a separate conclusions section. This text being more like a discussion fits with journal guidelines. Following your recommendation we have removed the paragraph on future changes in the abundance of plant strategies across trait space. However, in response to a comment by reviewer two, we have had to slightly extend the initial paragraph which contextualises drought and climate change responses, how representative the site and continental scale drought experiments were in relation to anticipated future changes, and highlight potential deficiencies in the handling of vegetation responses to drought in Earth System Models as these can affect vegetation-climate feedbacks and future precipitation patterns.

6. About the Amazon point, why did not the rest of the Amazon was included then? A large-scale study, you can choose by country, continent, or vegetation type and here you mixed all of them and I was just curious why. And the focus is Amazon forest, and it is a huge forest that goes beyond Brazil and Venezuela. So, I guess maybe add some more explanation on SI.

Response 10: Unfortunately, there is no good answer to this question. I can only apologise for the division of areas based on national boundaries.

During the model development phase I (then a PhD student) chose an area that encompassed a large amount of tropical forest and savanna vegetation as forest savanna boundaries were my initial focus, but the area was small enough that it could be reasonably simulated. aDGVM2 requires quite a bit more RAM and simulation time than other vegetation models. At the time, subsetting areas based on national boundaries represented an easy way to derive a manageable simulation area. In retrospect, dividing by country boundaries, and not re-adjusting the simulation area to the full extent of the Amazon forests, were mistakes.

I fully recognise the vastness of the Amazon forest that extends into many countries. Future simulations will recognise the full extent of these forests.

I have added text to the supplement (L1027-1030) to state briefly why the area was subset in this way and that future simulations will take account of the full extent of the Amazon forest that extends beyond the currently subset national boundaries.

7. On the SI, figures SI 1 and A3, I did not see any information about the allocation in storage. It would be good to add this detail.

Response 11: Fig. A3 depicts the hydraulics module, allocation to storage does not directly influence plant hydraulics.

S1 is a concept diagram. In this diagram we present only the adaptive traits which directly influence plant hydraulics. Allocation to storage does not directly affect plant hydraulics. Figure A5 contains information about allocation to storage as does the text in sections A3.8 and A3.9 of the appendix.

#####

Reviewer #2 (Remarks to the Author):

The authors have done a great job revising the manuscript! I only have a few comments.

Response 12: Dear Reviewer,

Thank you for the positive feedback!

The term 'functional diversity' is central to this paper, but there seems to be different ways of calculating it and it's not always clear which calculation the authors refer to when the term is used.

- "Functional diversity" and "Functional trait diversity" – Please use consistently or add at first use that they have the same meaning.

Response 13: Thank you. We have clarified (line 50) that functional trait diversity and functional diversity are synonymous before the first use of functional diversity.

- Information about what goes into Functional diversity is quite essential, and should probably be in the main manuscript. E.g., consider mentioning the eight traits from L1183 in the Supplementary in the main (text and/or relevant figure caption).

Response 14: In all figures which display functional diversity the number of traits used for the calculation and reference to these traits have been added.

In the main text we make it more clear that different traits were used in Figs. 1 and 4 by adding 8T and 2T respectively after RaoQ and clarifying what this means in the captions.

The Fig. 1 caption references the 8 traits displayed and described in Fig. 2 and the caption.

The Fig. 4 caption mentions both traits (P50 and Root-D) used to calculate functional diversity.

The Fig. S4 caption now mentions that functional diversity was calculated using 8 traits and references the 8 traits displayed and described in Fig. 2 and the caption.

The caption of Fig. S11 already mentioned the two traits used to calculate functional diversity.

- So, how many different ways of calculating functional diversity are there in this paper? At L1314 Suppl.: "Functional diversity was calculated as Rao's...": Confusing. So is the 'functional diversity' here calculated from a different set of traits than the eight traits listed at L1183? At L1366 Suppl.: "Two hydraulically associated traits were used to calculate functional diversity...". If the term 'functional diversity' is used to refer to different calculations, it should be made more clear, for example by use of subscripts (such as $FD_{8traits}$, $FD_{hydraulic}$ etc).

Response 15: Thank you. This has been done, see response 14.

Also, the term 'hydraulic trait' is not clearly defined anywhere in the main text. The 'water-related traits' in abstract does not really explain it. If hydraulic traits simply refer to the two traits "plants maximum rooting depth, and the xylem water potential at which 50% of conductance is lost", it should be stated clearly early in the main manuscript, given how central the term is for this paper.

Response 16: Thank you. We have now defined what we mean by hydraulic traits on L229-231 as:

"Thus, along this axis, classically hydraulic and phenological traits together mediate plant water status; we, therefore, refer to these traits collectively as hydraulic traits."

Abstract "We show that higher functional trait diversity reduces site-scale biomass loss during sudden catastrophic drought by 17-32% and continental-scale biomass loss due to chronic climate change-associated precipitation reductions by 10-34%." I am of the view that numbers given in an abstract should be possible to understand roughly without having read the entire paper. In this case, the way the numbers given may be interpreted in many ways, for example, a straightforward way to interpret these numbers would be to think that they refer to an uncertainty range relevant for real-world droughts. This is, however, not at all the case.

Response 17: Thank you for your comments.

Regarding the potential misinterpretation in relation to real-world drought, we now begin the sentence with "Simulations reveal" rather than "We show that" to avoid this potential interpretation.

Catastrophic and chronic droughts are also terms that leave it up to the imagination of the reader. Without a better understanding of the context and background, these numbers are quite meaningless, and may spread in news and social media to cause more confusion than clarity. I would suggest either replacing the quantitative reporting with a qualitative one (for example your interesting finding that FD effect on ecosystem resistance is stronger under more severe drought conditions), or that you also provide the necessary background (e.g., ...by 34 % under severe chronic drought (i.e., 50 % precipitation reduction over X years)...).

Response 18: We agree for the most part with your comments 17 and 18. We had not properly considered the news and social media context. We have rewritten the abstract to follow both of your suggestions, i.e. providing the necessary background, and adding the qualitative reporting.

This text now reads:

“Simulations reveal that higher functional trait diversity reduces site-scale biomass loss during sudden catastrophic drought, i.e. a 50% precipitation reduction for four and seven years, by 17% and 32%, and continental-scale biomass loss due to severe chronic climate change-associated precipitation reductions, i.e. RCP8.5, constant CO₂ (380 ppm), and a 50% precipitation reduction over 100 years, by 34%. Additionally, we find that functional trait diversity-mediated biomass resistance is stronger under more severe drought conditions.”

You may also want to explain how the 30-50% precipitation experiments compare to historical observations or projections of precipitation levels for the Amazon, for helping the reader understand how comparable these theoretical experiments are to the real-world, having in mind the broad readership of Nature Communications.

Response 19: Thank you for another valuable suggestion. This will also provide a better understanding of the context and reduce potential misinterpretations (similar to response 18).

We have adjusted the first paragraph of the closing section now called “Cross scale synthesis: drought, climate change, and functional diversity” to account for your comment. We also use the opportunity to highlight uncertainty in future precipitation changes, how precipitation patterns can change with vegetation change, and that ESMs generally fail to capture vegetation responses to drought. This text is as follows:

”At the site scale our simulations reveal drought responses in line with observations^{16, 17} (Fig.1), however, such extreme multiyear events are unlikely. At the continental scale the model predicts that the carbon sequestration functioning of the Amazon is broadly in line with observed and predicted trajectories^{4, 56}(Fig. S8). However, the chosen climate forcing data predicts large precipitation reductions.⁶³ The 50% precipitation reduction (Fig. 4) demonstrates the effect of functional diversity under extreme drought. Such a reduction is not predicted for Northern South America under CMIP6 SSP5-8.5.^{64, 65} For SSP5-8.5, CMIP6 models predict conflicting precipitation changes (median -14.7%, 5-95% quantile range -31.9% to 4.9%). However, Brazilian ESM experiments demonstrated that the interactive effects of Amazon rainforest cover change (savannization) and climate change resulted in a 44% precipitation reproduction across the Amazon basin showing that vegetation changes can have profound cascading effects on climate. ESMs generally fail to capture vegetation responses to drought.⁶⁶ Thus, while significant progress is being made,⁶⁷ DGVMs and ESMs, our primary tools for predicting future vegetation carbon dynamics and climate change, still require further refinement to capture physiologically and ecologically mediated future dynamics,⁶⁸ particularly those related to catastrophic and chronic drought.”

Other than that, only minor comments:

- Fig. 1: Consider making the cross a bit more prominent (e.g. add legend). Caption says ”Horizontal bars indicate the start and end of simulated and real-world drought treatments.” – does this refer to the same horizontal bar annotated as ”Simulated drought” in the A and B subplots? If so, consider revising the annotation to e.g. simply ”drought treatment” or

”simulated and real-world drought” for clarity. Also, is functional diversity only calculated for the model results, i.e., in C, real-world functional diversity can not be shown, is that right?

Response 20: Thank you for your comments.

The cross has been made more prominent and a legend has been added.

“Simulated drought” has been changed to “Drought Treatment”.

Yes, functional diversity is only calculated for the model results, we do not have information at this point on real-world functional diversity in these forests.

- L299 'results not shown': Better to add support of claims to the Supplementary? Actually, Nature Communications actively discourage this type of statements (see <https://www.nature.com/ncomms/submit/how-to-submit>): "We request that authors avoid "data not shown" statements and instead include data necessary to evaluate the claims of the paper as Supplementary Information."

Response 21: Thank you, results to support this claim have been added to Fig. S4 as additional panels. We now include additional panels which show the relationships between functional diversity, normalised mean transpiration (panel B), and the normalised standard deviation (panel C). The text has also been modified to reference these panels in the main text.

- Fig. S1: part of the text annotations are up-and-down, better to flip them the right way to make it easier to read.

Response 22: This has been done.

- Supplementary. Consider adding brief explanations and motivations for the choice of measures and methods (e.g., RaoQ) along with relevant references. Even if the choice may seem obvious to you, it might not be for readers, especially if you target readers also outside the immediate ecologist community.

Response 23: This has been done. The reason was that it can take relative abundances into account. A brief explanation and reason for choosing RaoQ has been added (L984-986, L1141-1143).

Reviewer #4 (Remarks to the Author):

General comments:

The new organization, with clearly stated research questions and the results section structured primarily around the questions (one section and principal figure per question), is a vast improvement in terms of clarity. The new structure of the text, showing clearly how each question has been addressed, is easy to read and follow. This distillation of the work (“less is more”) greatly clarifies the results and makes the paper stronger. When one has done so much work, it is difficult to leave so much “on the cutting room floor”, so to speak, but it is necessary to convey your story clearly. It also makes the case much more strongly that this model platform is useful for such explorations of trait diversity.

The care now taken to describe and justify the steps of the research and the decisions made, in the main paper, supplement and appendix, is also a great improvement, making the study and conclusions more sound.

I suggest only minor revisions are necessary before the paper is published (see specific comments below). I do not need to see the revised manuscript again.

Response 24: Dear Reviewer,

Sincere thanks for your constructive feedback which helped us improve our manuscript.

Specific comments:

L 67: Start a new paragraph with “We pursue...”

Response 25: this has been done.

L136, 140, 187, 369, 376, 412, 429-445, 590-644, perhaps more: Please use present or present-perfect tense for your own work and results, not past tense. Throughout the paper, present and past tense is mixed when describing your work and results, which is quite jarring.

Response 26: Apologies for the mixing of tenses. This has been correct at the points you suggest. The main text has been checked to ensure consistency of tense usage. The number of changes to the main text at different positions are listed below. Additional changes to the supplement are not listed.

- L70: 1
- L136-143: 2
- L163: 1
- L169-182: 5
- L184-314: 5
- L315-440: 36
- L441-549: 30

L161: Use the model name “aDGVM2” instead of “the model”, as this is the first instance in this section.

Response 27: this has been done.

L222: The strategies do not “avoid” the drought – the drought occurs regardless. Better to say that the strategies moderate or reduce the consequences of drought.

Response 28: “avoid” has been replaced by “reduce the consequences of”.

L226-227: I believe the terms in parentheses are reversed from what they should be.

Response 29: Apologies, you are correct. This order has been corrected.

Figure 2 and associated text: It is now clear that there are only 2 species in the Deciduous-Light category, and they arise only at one site. Should they even be included in the analysis? It seems no significant conclusions can be drawn from such a small sample

regarding where they sit in PC-space - this point should at least be mentioned. On the other hand, we see clear separation in PC-space for the Deciduous-Water category.

Response 30: While the Deciduous_Light category only consists of two species that emerge at only one site (and did not emerge in previously submitted simulation results), we see no need to remove these data points. Their presence does not affect our analysis or conclusions. We have added the following note to the figure caption to acknowledge that no significant conclusions can be drawn from the two data points associated with Deciduous_Light plant strategies.

Fig. 2 caption: “Note: significant conclusions cannot be drawn from the two Deciduous_Light species.”

L404-406: This statement is somewhat incomplete, and thus inaccurate. PC1 and PC2 have now exchanged places – this should be stated clearly. However, these two axes remain the two main sources of variance, explaining 49-60% depending on the location and scale. That can be emphasized as well - currently there seems to be attention given only to the hydraulic trait axis.

Response 31: Thank you for your comment. This has been done (L417-421).

Figure 3: The explained variance stated on each axis is identical (to the 3 significant digits shown) for each of the four cases. This seems improbable – is there perhaps an error in the plotting script? There are clear (albeit usually small) differences in the PDFs.

Response 32: Dear reviewer, precipitation and temperature anomalies were only applied for the last 100 simulation years (Tab. S5), we have clarified this in the supplement L1032 and L1038.

CO₂ treatments also only diverge from the simulation year corresponding to year 2000 concentrations (Fig. S2 G). As such, the simulation output for each climate change scenario was identical in the year 2000 (the random seeds in the model were also fixed to ensure this), hence the identical numbers for explained variance. All simulations being identical in the year 2000 is the reason for the identical variances. From the year 2000 onwards, simulation forcing differs, which leads to the differences in the frequency of individuals across trait space in 2100.

Figure S1: I still find this (an even more) useful conceptual diagram for the model framework – it is a shame it cannot fit into the main paper.

Response 33: We too find this concept diagram useful but feel it is better situated in the supplement.

Figure S4: For clarity and to help make the author’s point, could you add the y=0 line to the plot, and perhaps also a vertical line showing the x-value (they are very close) where the curves cross this line? This is a key threshold in the study.

Response 34: This has been done. Panels displaying the individual relationships between functional diversity, mean transpiration, and the standard deviation in transpiration have also been added. This was requested by reviewer 2.

Figure S6: The abbreviations for the curves are not intuitive compared to the terminology used elsewhere. I presume “Red” means “reduction” but in the caption and elsewhere, you say diversity (“Div”) is "removed". Please choose consistent labeling.

Response 35: Thank you for your comments. The abbreviations have now been removed from Fig. S6 and labels are consistent with legend text and main text.

Figure S12: Likewise, the curve labels are quite cryptic. Please expand as in the Fig. S11 caption, or map them clearly to what is shown in Fig. S11.

Response 36: Thank you. This has been done. The labels for Figs. S12 and S13 now match those of Fig. S11.

Figure A2: The figure is not “below” – the first five words can be deleted.

Response 37: Apologies, this has been done.